# Action repetition biases choice in context-dependent decision-making
**Ben J. Wagner** [1,2] ✉, **H. Benedikt Wolf**[1] **& Stefan J. Kiebel** [1,3]

Humans are prone to decision biases, which make behavior seemingly irrational. An important cause for decision biases is that the context in which decisions are made can later influence which choices humans prefer in new situations. Current computational models (e.g. relative value learning or range normalization) often require extensive environmental knowledge to explain these biases. Here, we tested the hypothesis that decision biases are mainly driven by a tendency to repeat context-specific actions. We implemented a series of nine value-based decision-making tasks on n = 351 male and female participants and reanalyzed six previously published datasets (n = 350 participants). We found that higher within-context repetition of an option was associated with biased choices including higher subjective valuation and lower uncertainty for repeated actions. Next, we used a hierarchical Bayesian reinforcement learning model based on two basic principles, learning by reward and action repetition and tested it on all datasets. Our results show that the combination of these two basic principles is sufficient to explain biased choices in stable environments. We demonstrate via extensive model comparison that our model outperforms all tested alternatives (implementations of value normalization and a goal centric account). These results provide insights into decision biases during value-based decision-making and suggest a parsimonious mechanism for understanding habit-like choice tendencies.

Unlike an unconstrained rational agent, human information processing is limited by memory capacity and processing bandwidth[1]. Contextualized processing might be one adaptive solution to such limitations as it constrains the stream of information, i.e., what memories or action plans are accessed[2]. Evidence for the role of context in human cognition includes classical and instrumental conditioning[3], the interplay of context and memory in extinction learning[4,5], the role of context in associative-, structure- or representation learning[6–8], the context-specific interplay of habitual and goal-directed behavior[9–12] or contextual inference in motor learning[13,14]. Experimentally, strong effects of context have also been shown in value-based decision-making tasks where participants first learn to make choices between rewarding options and subsequently show counter-intuitive decision biases[15–19]. Specifically, in the first phase of such an experiment, participants learn the value of different options in exclusive choice contexts. For example, participants must make repeated choices between two options, A and B (in context one) and options C and D (in context two), but never between options across contexts (e.g., between A and C). After this learning phase, participants enter a so-called transfer phase. In this phase participants make choices between two options across contexts, i.e. choices between new

combinations of options such as A and C, or between B and D. It is in this transfer phase that participants often make choices that appear surprisingly counter-intuitive or even irrational, especially when viewed through the lens of expected utility theory, which assumes that humans behave as if they maximize subjective utility. For instance, humans often display a strong and systematic choice preference when none should exist, such as when both A and C should have the same expected value[15,18–20]. In some cases, participants even prefer losses over gains[16].

In recent years, various mechanisms and computational models have been proposed to explain these counterintuitive preferences. Some of these models were inspired by the discovery of normalization of neuronal signals, such as contrast gain control in the retina[21] or auditory cortex[22]. In consequence, it has been suggested that normalization might be a key principle that also applies to how the brain stores and retrieves value representations[23,24]. Further research investigated which specific form of normalization explains human choices best[17,18] or whether humans balance between internal and external reward signals, given an internal goal to obtain the maximum outcome[19,25]. One potential issue with these approaches is that they, in principle, require specific statistical knowledge about the

[1]Chair of Cognitive Computational Neuroscience, Faculty of Psychology, TU Dresden, Germany. [2]Department of Psychiatry and Psychotherapy, Faculty of Medicine, University of Tübingen, Tübingen, Germany. [3]Centre for Tactile Internet with Human-in-the-Loop, TU Dresden, Germany. ✉e-mail: ben.jonathan.wagner@gmail.com; ben_jonathan.wagner@tu-dresden.de; ben.wagner@tuebingen.mpg.de

environment. For example, for range normalization[18] or the weighting of intrinsic and extrinsic reward signals[19], participants are assumed to update value estimates with respect to the outcomes of all alternatives. However, such knowledge may or may not be available in real-world decision-making.

We propose a computationally simpler account of these biased preferences, based on standard reinforcement learning (RL) and a tendency to repeat previous context-specific choices. In this framework, consistent with Thorndike's *Law of Exercise*[26], we suggest that merely making a choice can bias the likelihood of selecting that option in the same context again. Several experimental and modeling studies present evidence that preference formation may be shaped by both reward-based and repetition-driven mechanisms, i.e., a repetition bias[10–12,27–35], and that making and repeating a choice might be related to its valuation[36–38]. Theoretically, this repetition bias may arise from cognitive resource constraints, as perseveration-like behavior can serve as an adaptive strategy to reduce cognitive effort in complex decision environments[1,32,35].

To comprehensively test such a mechanism, we collected nine new datasets ($n = 351$ participants) and conducted a reanalysis of six datasets from published studies ($n = 350$ participants). These include tasks involving probabilistic and continuous rewards, losses, and gains, as well as tasks with as few as two or as many as 66 across context comparisons, and tasks where feedback is provided either for all alternatives or only for the selected option. Using formal model comparison based on hierarchical Bayesian modeling, we compare this model to many previously proposed alternatives: relative value learning[15], divisive (DIV) normalization[23,39–41], variants of range normalization[18] and a model that weights external and internal reward signals[19]. We hypothesized that combining reward-based learning with action repetition would be sufficient to explain decision biases previously attributed to value normalization.

## Methods
### Participants & preregistration
Participants were recruited via the online platform *Prolific* (www.prolific.com). We only considered participants who completed the experiment (male = 175, female = 176; mean(age) [sd] = 32.32 [8.24] years). In addition, we excluded two participants after we looked at the data: one for taking a much longer-than-anticipated break between learning and transfer, and one due to a data-collection error, resulting in 349 (male = 174, female = 175) participants for analysis. All participants were provided written informed consent prior to task participation via the *REDCap* platform. Inclusion criteria were fluent English and age within the range of 18–49 years. Gender was self-reported via questionnaire. We did not collect data on race or ethnicity. The study procedure was approved by the Ethics Committee of TU Dresden (ID: 578122019) and was not preregistered.

### Sample size
We did not run a formal a priori power analysis. Instead, we decided on a minimum sample size by benchmarking against the two most closely-related studies available when we designed the work: Klein et al.[15] and Bavard et al.[16] reported robust effects of their normalization models in comparison to standard RL with $N = 21$–$40$ participants per experiment.

### Tasks
All tasks consisted of a learning stage (60–80 trials) and a subsequent transfer stage (20–36 trials; see Table 1). The learning phase of each task consisted of two choice contexts. A choice context was defined by the available options (represented by abstract stimuli). Participants encountered two options per choice context in tasks with probabilistic reward feedback (probabilistic tasks: p1[p.1.1, p1.2, p1.3], p2, p3, p4.1, p4.2) and three options per context in tasks with Gaussian reward feedback (Gaussian tasks: g1, g2). In these probabilistic tasks, each option was associated with a reward probability, e.g., 0.7 for winning 0.1€ or nothing. In Gaussian tasks, the outcome for each option was sampled from a normal distribution with an option-specific mean, e.g., 70 points, and a standard deviation of 6 (see Table 1 for details). Participants received either partial feedback, i.e., they

were visually informed about the outcome of the chosen option, or full feedback, i.e., they were informed about the outcome of the chosen option and all alternatives. The presentation of the different contexts was fully interleaved. For a detailed overview of all features for all tasks, see Fig. 1 and Table 1.

### Probabilistic tasks
Task p1 (including variants p1.1, p1.2, and p1.3) and task p2 aimed to replicate the task structure of previous studies. Specifically, p1.1, p1.2, and p1.3 were variants of the task used by Klein et al.[15], with small variations to the reward probabilities for each option (see Table 1 for an overview). In these tasks (p1.1–p1.3), participants received full feedback. Task p2 shared the same underlying parametrization as p1.1 but provided participants with partial feedback. The common feature across these tasks and the Klein et al.[15] tasks is that the difference in reward probabilities for options in the first context is different from the difference in reward probabilities for options in the second context. For example, during learning, participants encountered a low contrast (LC) context with options $A_{LC}$ (70% reward probability) and $B_{LC}$ (50%), resulting in a 20% difference. Additionally, they faced a high contrast (HC) context with options $C_{HC}$ (70%) and $D_{HC}$ (20%), where the reward probability difference was larger at 50%. By using different task parametrizations and providing either full feedback (p1) or partial feedback (p2), we aimed to replicate the main findings from Klein and colleagues[15] and test the robustness of these effects.

In task p3, the LC context was encountered more frequently than the HC context during the learning phase (50 times vs 30 times), allowing us to explicitly test the hypothesis that the number of repetitions influences preferences for choices in the transfer phase.

In tasks p4.1 and p4.2, both contexts in the learning phase had the same difference of reward probabilities (0.2). Participants encountered a low gain (LG) context with options $A_{LG}(0.60)$ and $B_{LG}(0.40)$, and a high gain (HG) context with options $C_{HG}(0.76)$ and $D_{HG}(0.56)$, meaning the best options in each context differed in expected value. For the learning phase, presentation of contexts was fully interleaved following a pseudorandomized sequence. During the transfer phase, our focus was on testing specific predictions by different models, whether participants would prefer option A (0.60) or C (0.76). For example, according to state-dependent normalization (mean centering), participants should show indifference between these options (Please note that this equality in relative value would not hold if we were to normalize by the sum). In task p4.1, early on in the learning phase, we showed counterfactual feedback to participants in the HG context. We hypothesized that this would make the best option in the HG context, $C_{HG}(0.76)$, harder to learn and thus less frequently chosen. In task p4.2, this early counterfactual feedback in the HG context was totally absent, making the task easier, and we expected that participants would choose the best option in the HG context more often. For all probabilistic tasks, reward sequences were pseudorandomized under the rules described below (except the manually altered change in the first trials of p4.1 and p4.2), and all participants experienced the same sequence (see Supplementary Table 4 for exact sequences). Crucially, sequences for the learning phase were designed such that the sampled reward probabilities closely matched the true reward probability of each option, measured in blocks of 10 to 12 trials. For instance, if an option had an expected value of 0.7, approximately 7 out of every 10 to 12 trials had to be rewarded. Additionally, the reward sequence for the best options, $A_{LC}(0.7)$ and $C_{HC}(0.7)$, in both contexts was kept identical for the last four trials in the learning phase (except in task p1.2). This was done to control for simple forms of perseveration or forgetting.

### Transfer-phase
Design of the transfer phase followed prior studies. For probabilistic tasks, we followed Klein et al.[15] focusing on the first (binomial tests and logistic regression) or first two transfer trials (computational modeling). Within these tasks, reward feedback was provided during transfer. For the first transfer trial, reward feedback was equal for the best HC and LC options (e.g., $A_{LC}(0.7)$ and $C_{HC}(0.7)$) to minimize differential value-updating (here,

**Table 1 | Overview of task parametrizations, i.e., reward schedules, number of trials, and transfer comparisons for all new and reanalyzed value-based decision tasks**

| New datasets | | | | | |
|---|---|---|---|---|---|
| **Probabilistic tasks** | | | | | |
| **Task (nParticipants)** | **Reward probability % (LC context)** | **reward probability % (HC context)** | **Feedback** | **nTrials learning** | **nTrials transfer (distinct comparisons)** |
| p1.1 ($n = 36$) | A: 0.7 B: 0.5 | C: 0.7 D: 0.2 | full | 30 per context | 10 cross-context (2 comparisons: A vs C and B vs D)/ first two for main analysis |
| p1.2 ($n = 30$) | A:0.73 B: 0.5 | C:0.73 D: 0.2 | full | 30 per context | 10 cross-context (2 comparisons: A vs C and B vs D)/ first two for main analysis |
| p1.3 ($n = 29$) | A: 0.6 B:0.4 | C 0.6 D: 0.1 | full | 30 per context | 10 cross-context (2 comparisons: A vs C and B vs D)/ first two for main analysis |
| p2 ($n = 49$) | A: 0.7 B: 0.5 | C: 0.7 D: 0.2 | partial | 30 per context | 10 cross-context (2 comparisons: A vs C and B vs D)/ first two for main analysis |
| p3 ($n = 49$) | A: 0.7 B: 0.5 | C: 0.7 D: 0.4 | full | 50 LC context 30 HC context | 10 cross-context (2 comparisons: A vs C and B vs D)/ first two for main analysis |
| p4.1 ($n = 30$) | A: 0.6 B: 0.4 | C: 0.76 D:0.56 | full | 30 per context | 10 cross-context (2 comparisons: A vs C and B vs D)/ first two for main analysis |
| p4.2 ($n = 29$) | A: 0.6 B: 0.4 | C: 0.76 D: 0.56 | full | 30 per context | 10 cross-context (2 comparisons: A vs C and B vs D)/ first two for main analysis |
| Gaussian tasks | | | | | |
| **Task (nParticipants)** | **Reward magnitude (LC Context)** | **Reward magnitude (HC Context)** | **Feedback** | **nTrials learning** | **nTrials transfer** |
| g1 $n = 50$ | A: 70 B: 50 C: 40 | D: 70 E: 40 F:10 | full | 40 per context | 4 cross-context trials (9 comparisons) |
| g2 $n = 48$ | A: 70 B: 50 C: 40 | D: 70 E: 40 F:10 | partial | 40 per context | 4 cross-context trials (9 comparisons) |
| Existing datasets | | | | | |
| **Task** | **Reward probability % LC context** | **Reward probability % HC context** | **Feedback** | **nTrials learning** | **nTrials transfer (distinct comparisons)** |
| Klein et al.[15] Exp. 2 ($n = 24$) | A: 0.7 B: 0.5 | C:0.7 D: 0.2 | partial | 30 per context | 30 cross-context (1 comparison: A vs C), first two for main analysis |
| Klein et al.[15] Exp. 3 ($n = 21$) | A:0.8 B:0.6 | C: 0.8 D: 0.6 | full | 30 per context | 30 cross-context (1 comparison: A vs C)/ first two for main analysis |
| Bavard et al.[16] Exp. 1 ($i = 20$) | context 1: A: 1€ (0.75) B: 1€ (0.25) | context 2: C: 0.1€ (0.75) D: 0.1€ (0.25) | context 3 E: -1€ (0.25) F: -1€ (0.75) | context 4 G: −0.1€ (0.25) H: −0.1€ (0.75) | partial/full | 40 per context | 4 cross-context (28 comparisons) |
| Bavard et al.[16] Exp 2 ($n = 40$) | A: 1(0.75) B: 1(0.25) | C: 0.1(0.75) D: 0.1(0.25) | E:-1(0.25) F:-1(0.75) | G:−0.1(0.25) H:−0.1(0.75) | full | 40 per context | 4 cross-context (28 comparisons) |
| Task | mean (reward) Context 1 | mean (reward) Context 2 | mean (reward) Context 3 | mean (reward) Context 4 | | |
| Bavard and Palminteri.[18] Exp. 3a ($n = 100$) | A : 86 B :50 C :14 | D :86 E :50 F :14 | G :50 H :32 I :14 | J :50 K :11 L :14 | full (stim D and J blocked in 90 trials) | 45 per context | 2 cross-context (66 comparisons) |
| Bavard and Palminteri.[18] Exp. 3b ($n = 100$) | A :86 B :50 C :14 | D :86 E :50 F :14 | G :50 H :32 I :14 | J :50 K :32 L :14 | full (stim D and J blocked in 135 trials) | 45 per context | 2 cross-context (66 comparisons) |

Note for tasks p1-p4.2 and the two data sets from Klein et al.[15], we only used the first two transfer trials for standard analysis and computational modeling, as these tasks included reward feedback during the transfer phase.

reward was delivered to both options). All subsequent transfer trials (starting with the second) followed the same pseudo-random schedule used during learning. Full feedback in the transfer phase was shown in tasks that had full feedback during learning; partial-feedback tasks continued to reveal only the chosen option's outcome. For the Gaussian tasks, we followed the design of Bavard et al.[16] and Bavard and Palminteri.[18] Here, no feedback during transfer was provided. For standard analysis, we again focused on the first cross-context comparison of the high-reward options, while for computational modeling, we included all trials and cross-context comparisons. For an overview of task dynamics during learning and transfer, see Table 1.

**Gaussian tasks**

Tasks using choice contexts with different reward magnitudes and different reward ranges provide a unique opportunity to test the predictions of normalization models, such as DIV normalization[23,39,40,42] and variants of range normalization[18]. In tasks g1 (full feedback) and g2 (partial feedback), each choice was associated with a different reward magnitude. Reward magnitudes were drawn from a Gaussian distribution with different means and a standard deviation of six. Participants encountered three options per context. In the LC context, the means of the reward magnitudes were $A_{LC}(70)$, $B_{LC}(60)$, and $C_{LC}(50)$ points, while in the HC context, they were $D_{HC}(70)$, $E_{HC}(40)$, and $F_{HC}(10)$ points. During learning and transfer, both contexts yielded different background colors and unique tree icons in the corners. This was an exploratory design decision with the intention to boost participants' awareness of the different contexts and cross-context decisions during the transfer phase. According to standard range normalization, participants should show indifference between $A_{LC}(70)$ and $D_{HC}(70)$, as both would normalize to 1

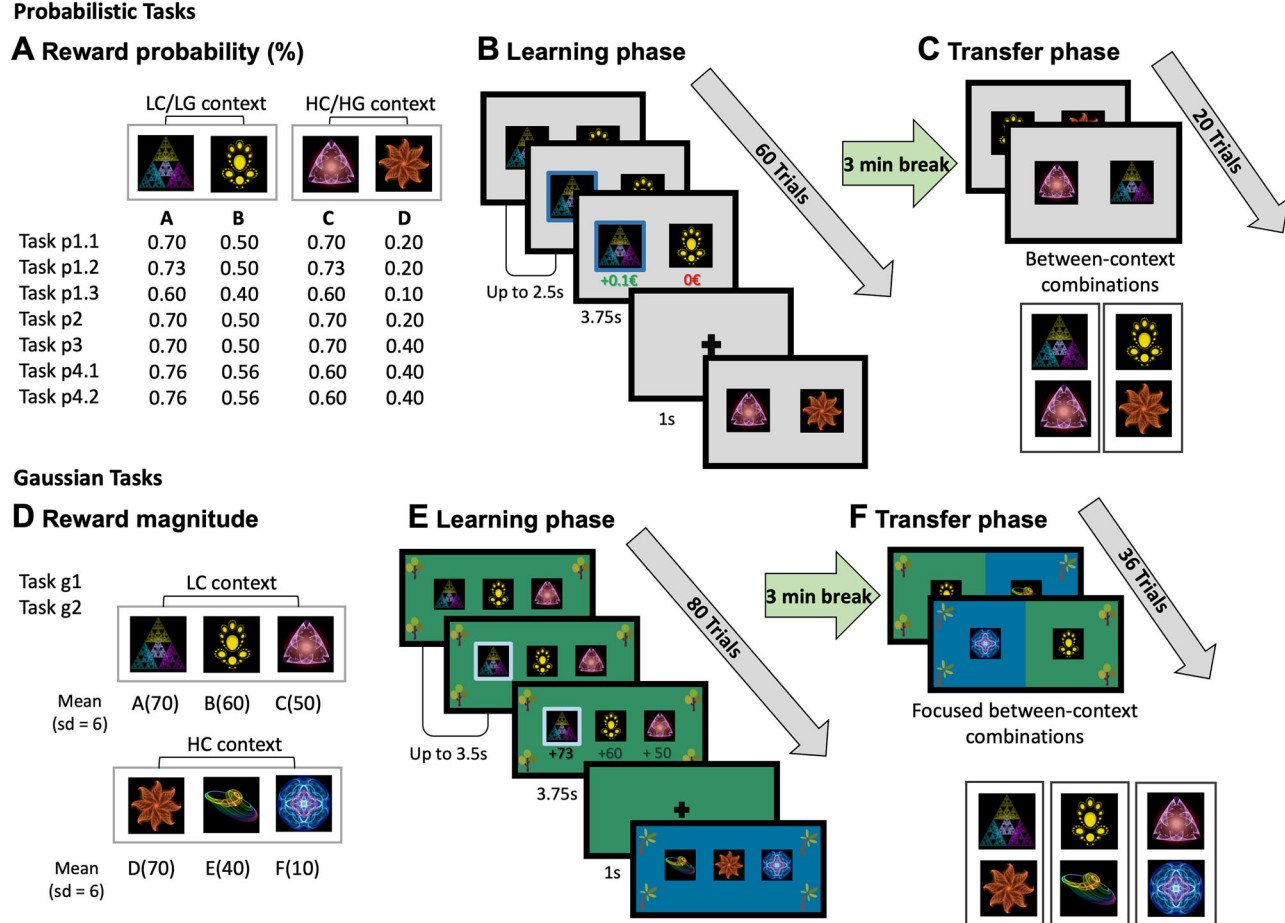

**Fig. 1 | Overview of value-based decision-making tasks (acquired data).** Top row: Probabilistic Tasks. **A** Reward probabilities for all four options (**A**–**D**), encountered in both the low contrast (LC) and high contrast (HC) choice contexts in tasks p1.1 to p3, and in the low gain (LG) and high gain (HG) choice contexts in tasks p4.1 and p4.2. **B** Learning phase for probabilistic tasks. Participants are presented with pairs of options (abstract fractal stimuli) within both LC/LG and HC/HG contexts and must select one option per trial. Across multiple trials, they learn to choose the better option through trial-and-error. In tasks p1.1–p1.3 and p3–p4.2, full feedback was provided for both the chosen and unchosen options. In task p2, only partial feedback was provided for the chosen option. Trials are presented in a pseudorandomized order, ensuring that the actual reward probabilities are maintained across every 10–12 trials (for exact sequences, see Supplementary Table 4). For analysis purposes, tasks p1.1 to p1.3 were pooled into a single dataset (task p1) due to their shared features and the absence of significant differences in results when analyzed individually. **C** In the transfer phase, options from both LC/LG and HC/HG choice contexts are tested against each other to evaluate whether participants show systematic biases, such as a preference for the $C_{HC}(0.70)$ option over the $A_{LC}(0.70)$ option. For all probabilistic tasks introduced in this study, we tested two cross-

context comparisons during transfer ($A_{LC}$ vs $C_{HC}$ and $B_{LC}$ vs $D_{HC}$) **D** Reward magnitudes for Gaussian tasks. In these tasks, trial-wise reward magnitudes for each option are sampled from a Gaussian distribution with a standard deviation of six. Each choice context consists of three options. In the LC context, reward distributions for different options do overlap, making it more challenging to identify the best option. **E** Learning phase for Gaussian tasks, equivalent to **B**. Participants make repeated choices between options within the LC and HC choice contexts. To help distinguish between contexts, visual cues are provided: distinct background colors (blue or green) and different tree types (evergreen or broadleaf trees) displayed in the four corners of the screen. **F** In the transfer phase, equivalent to **C**, participants' preferences across contexts, options from both the LC and HC choice contexts are tested against each other (4 trials for each of 9 cross-context comparisons). This cross-context presentation is indicated by combined visual cues, including a mix of background colors and tree types. In all tasks, participants' cumulative monetary reward was calculated by summing their trial outcomes: €0.10 for each correct choice in probabilistic tasks, or an amount proportional to the points earned in Gaussian tasks (0 to 100 points multiplied by €0.01). Participants received their cumulative reward as a bonus payment upon completing each task.

when applying range normalization. One purpose of using these tasks was to test whether this theoretical prediction holds or if alternative models, especially the proposed repetition model, explain the preferences in the transfer phase better. Rewards during the learning phase were sampled from a normal distribution on a trial-by-trial basis as in Bavard and Palmineri[18], resulting in unique reward sequences for each participant due to sampling noise. No reward feedback was provided in the transfer phase for Gaussian tasks.

**Bonus payment**

For all tasks, performance bonuses were computed based on individual task outcomes. Participants could earn an additional 0.1€ for each reward collected in all probabilistic tasks and 0.01 * reward magnitude in points for

Gaussian tasks. Bonus reimbursement was paid via the *Prolific* internal payment system after task performance.

**Overall procedure for new tasks**

Nine groups of male and female human participants ($n = 351$) performed nine variants (p1 [p1.1, p.1.2, p1.3], p2, p3, p4.1, p4.2, g1, g2) of value-based decision-making tasks. Each group completed a single task (see task details above). After reading the instructions, participants completed a short training session (10 trials) and then answered four questions, in the form of a quiz, to ensure they understood the task. In case of an error, participants were allowed to retake the quiz until all questions were answered correctly.

Participants made their choices using a computer mouse or touchpad by pressing the left mouse button or touchpad. Keyboard-only input was not

allowed, as we assumed participants would be more focused if action selection required using a cursor to indicate their choice, rather than simply pressing left or right keys. In the main task, participants completed a 60 to 80-trial learning phase (depending on the task version, see above), followed by a mandatory three-minute break. Once a counter indicated the end of the break, participants received an on-screen text message informing them that they could resume making choices among all previously encountered options by pressing the space key. No additional instructions were provided regarding the recombination of options across choice contexts. In the transfer phase, participants made choices between options from different choice contexts. After completing the transfer phase, participants were asked to rate their belief about the true reward probability of each option using a 0-100 visual analog scale (except for task p1).

## Binomial tests & correlations

As a first step, we performed standard analyses based on linear inference statistics. To this end, we first show descriptive plots of the average rate of choosing the best option during learning and choice preference during the first two transfer trials (TT). To test whether participants show systematic biases in the first TT, we conducted exact binomial tests (two-sided) to determine whether preferences between the two high-reward options in the first transfer trial (e.g., $A_{LC}$ [0.7] vs. $C_{HC}$ [0.7]) significantly deviated from chance levels, indicating indifference.

We then focused our analysis on linear relationships between relative choice frequency during learning and relative choice frequency (choice preference) during transfer.

To do this, we counted the choices on the group level for each option during the learning phase and during the transfer phase and normalized these two counts with respect to the total number of presentations of a specific option pair. For example, if option $A_{LC}$ was chosen 400 times and option $C_{HC}$ 600 times, during learning, the relative choice frequency of $A_{LC}$ would be 0.4 and of $C_{HC}$ 0.6 (relative frequency learning). Likewise, if $A_{LC}$ was chosen 25 times and $C_{HC}$ was chosen 75 times when directly compared in the first transfer trial, the relative choice frequency would be for $A_{LC}$ 0.25 and for $C_{HC}$ 0.75 (relative frequency transfer). Subsequently, we analyzed the resulting task-specific correlation coefficients and posterior credible intervals between the relative frequency scores during learning and transfer. To further control for potential confounding effects of expected value on the observed correlations, we implemented additional selection criteria for our pairwise comparisons. First, we selected only those pairs where both options had equal expected values (objective expected value by experimental design), such as $A_{LC}(0.7)$ and $C_{HC}(0.7)$. Second, to account for possible effects of state-dependent normalization, assuming that participants learn values on a relative scale, we included only pairwise combinations with equal relative value. Additionally, we excluded the worst options (14-point bandits) from Bavard and Palminteri[18], as these options were selected in <10% of trials within a three-option context on average. However, we also computed all correlations with these options included, and the results did not substantially change. All correlations were computed as Bayesian correlations via *brms* (v2.20-2.22)[43] and an uninformative prior over 0. We drew 4 MCMC chains with 4000 iterations each. We then report the posterior mean of the correlation coefficient and its 95% posterior credible interval.

We quantified evidence for non-zero correlations with Bayes factors (BF$_{10}$) by fitting a bivariate model with a free residual correlation ($\rho$) and comparing it to a null model with $\rho$ fixed to 0 via bridge sampling. Data normality was assessed using the Shapiro-Wilk test. Based on test results, we report either Pearson or Spearman correlation coefficients.

## Frequency and valuation

For the correlation analysis between normalized choice frequency during learning or learning and transfer and valuation ratings, we determined the relative choice frequency for each option at the individual participant level. For each option, we calculated a normalized frequency score, representing how often it was chosen relative to all other options encountered during

learning (e.g., for tasks p1: how often did a participant choose option A in relation to options B, C and D). We then determined the differences in these normalized choice frequencies between pairs of options and correlated these differences with the differences in their individual valuation ratings. To control for reward effects, we focused on options with equal expected values (absolute and relative), as described previously. To analyze the relationship between choice frequency and uncertainty, we calculated the relative choice frequency for each option and the standard deviation of valuation ratings at the group level. We then tested for a linear relationship between relative choice frequency and the standard deviation of the value distribution. We further examined a correlation of relative choice frequency and absolute distance from the true reward probability on the participant level (probabilistic tasks) or reward magnitude (Gaussian tasks).

## Mixed-effects Bayesian logistic regressions

To test whether an effect of choice frequency remains once individual variability is taken into account, we fitted a Bayesian generalized linear mixed model with a Bernoulli likelihood and logit link (Eq.1):

$$Choice_{i,j}^{HC/HG} = Bernoulli\left(logit^{-1}\left(\beta_0 + \beta_1 FreqLearn_{i,j} + u_{0,j} + u_{1,j} FreqLearn_{i,j}\right)\right)$$
(1)

In this formulation, *i* indexes first-transfer trials, *j* indexes participants, *FreqLearn$_{i,j}$* is the relative frequency with which one option was chosen during the learning phase, $\beta_0$ and $\beta_1$ are group-level intercept and slope capturing the average baseline preference and the mean effect of frequency (repetition), and $u_{0,j}$ and $u_{1,j}$ are participant-specific random intercepts and slopes. A posterior estimate of $\beta_1$ that is credibly positive would demonstrate that greater learning-phase repetition increases the likelihood that a specific option is selected in the transfer phase. All models were fitted via the *brms* package (v2.20-2.22)[43], which interfaces Stan (2024, Stan Development Team) for full-Bayesian inference. Weakly informative priors were used (Normal (0, 1) for fixed effects; Exponential (1) for standard deviations). Models were fitted per four chains × 4000 iterations (2000 warm-up). For every data set, we fitted the full model above and a null model containing only group and random level intercepts. The relative evidences of these models were quantified with Bayes factors computed via marginal-likelihood estimation in *brms* (v2.20-2.22)[43]. Values > 10 are considered strong evidence for the frequency effect.

## Previously published datasets

We reanalyzed six datasets from three previous studies[15,16,18]. The selection of datasets was made prior to conducting any analyses (see Table 2 for details).

## RL models

Multiple computational models have been proposed to explain decision biases, i.e., irrational choice preferences, in value-based decision-making[15,16,18,19,42]. In the present study, we systematically compare these models against each other. Specifically, we analyzed our data using six different RL models: relative value learning[15], DIV normalization[23,39,41], variants of range normalization[18] and a model that weights external and internal reward signals[25]. As a control model, we used an absolute value learning (ABS) model[44] (see Supplementary Table 1 for an overview on comparisons). In the ABS model, which is equivalent to standard Q-learning, Q values (Eq. 2) within a context are updated in a stepwise procedure through a prediction error term (Eq. 3):

$$Q_{a,t}^{(c_t)} = Q_{a,t-1}^{(c_t)} + \eta * \delta_{a,t}^{(c_t)}$$
for a = {chosen, unchosen}
(2)

$$\delta_{a,t}^{(c_t)} = r_{a,t}^{(c_t)} - Q_{a,t-1}^{(c_t)}$$
for a = {chosen, unchosen}
(3)

## Table 2 | Reanalysis of previously published datasets

| Paper | Tasks | Reason for reanalysis |
|---|---|---|
| Klein et al.[15] | Exp2 ($n = 24$), Exp3 ($n = 21$) | Experiments 2 and 3 support the author's main findings, i.e., these tasks provide evidence for relative value learning. We replicated their task design in our tasks p1 and p2. |
| Bavard et al.[16] | Exp1 ($n = 20$), Exp2 ($n = 40$) | This study investigated choices between losses and gains and gains with different magnitudes in the transfer phase. Further this dataset is of higher complexity due to its four choice contexts and 28 across-transfer comparisons in contrast to only one transfer comparison in Exp.2 and Exp.3 by Klein et al.[15] and two transfer comparisons in our tasks p1 and p2. The observed apparent preference for losses is modeled by state-dependent normalization. |
| Bavard & Palminteri.[18] | Exp3a ($n = 100$), Exp3b ($n = 100$) | This study established evidence that preferences in the transfer phase are best explained by scaled range normalization (denoted as RANGEω). In addition, Bavard and Palminteri[18] blocked the best option in a subset of contexts during learning. This enforces repetition of the second-best option. Our Gaussian tasks (g1 and g2) are similar to their experiments 3a and 3b in terms of reward feedback, but of lower complexity (4 transfer comparisons in our datasets vs. 66 transfer comparisons in their dataset) |

where $c$ is the context in trial $t$. $\eta$ is a placeholder for the learning-rate (for different parametrizations see below) and $r_a$ is the reward for the chosen or unchosen option in trial $t$. $\delta_{a,t}$ is the corresponding prediction error, i.e. the difference between the received reward $r_{a,t}$ and the previously estimated Q-value for that option $Q_{a,t-1}^{(c_t)}$. In the case of partial feedback, only the Q-value for the chosen option is updated, thus $a$ = chosen. In the case of full feedback, $Q$ values for both the chosen and unchosen options are updated and $a$ = {chosen, unchosen}. Across all probabilistic tasks and the data from Klein et al.[15] participants encountered one unchosen option per context. In Gaussian tasks and data from Bavard and Palminteri participants encountered two unchosen options per context.

### Learning rules

Before presenting the remaining models and finalizing the ABS model, we introduce two learning mechanism variants, which were applied across all models.

In the first model variant, we used a trial-wise learning rate that decays as a function of context occurrences, and in the second model variant, we used different constant learning rates for chosen and unchosen options. Model variant 1 is motivated by RL research and suggests that when reward probabilities are stationary (no fluctuations in the environment), learning rates are assumed to decay over time, i.e., participants go gradually from a phase of initial exploration to exploitation[44]. A constant learning rate throughout the entire experiment would not adequately capture this shift. We tested several decay functions through comparative analyses based on simulated data (details not reported). Based on these tests, we selected a hyperbolic decay function, where the learning rate decreases as a function of the number of times a context has occurred (Eq. 4).

$$\eta_t^{(c)} = \frac{1}{1 + d^{(c)} * ci_t^{(c)}} \quad (4)$$

Here $d^{(c)}$ is the context-specific decay parameter that weights the number of times a specific context has occurred, and $ci_t^{(c)}$ is a context-specific counter that increases every time the corresponding context $c$ is experienced. $\eta_t^{(c)}$ is the corresponding context and trial-specific learning rate.

For the second model variant, following prior studies, we implemented different constant learning rates for chosen ($\eta_{chosen}$) and unchosen ($\eta_{unchosen}$) options[15,16,18]. Note, we do not systematically compare these two model variants, as this would go beyond the scope of this study. Rather, we implemented both learning rule variants to show that our main results do not change when other learning mechanisms, especially those that were used in prominent studies on state-dependent normalization are applied[16,18,45].

### Repetition model

The proposed repetition model (REP) consists of two components: (i) standard RL (ABS; see Eq. 2 and Eq. 3 above) with (ii) reward-independent

learning of a bias to repeat previous choices[10,11,29,30,33]. For this second component, we let repeated choices influence the repetition bias (REPbias) of an option depending on which option is chosen:

$$REPbias_{a=chosen,t}^{(c_t)} = REPbias_{a=chosen,t-1}^{(c_t)} + \eta_{repetition} * \left(1 - REPbias_{a=chosen,t-1}^{(c_t)}\right) \quad (5)$$

$$REPbias_{a=unchosen,t}^{(c_t)} = REPbias_{a=unchosen,t-1}^{(c_t)} + \eta_{repetition} * \left(0 - REPbias_{a=unchosen,t-1}^{(c_t)}\right) \quad (6)$$

i.e., after each choice, the REPbias within the current context increases for the chosen option and decreases for the unchosen options. The trial-specific change in an option's REPbias is governed by a participant-specific learning rate parameter, $\eta_{repetition}$. This model is theoretically related to Miller et al. and others[17,31] but differs in two significant ways: Unlike Miller et al.[31], we do not employ a weighting parameter (arbiter) to balance the influence of $Q$ values and REPbiases. Further, following[30,33], the REP model allows the REPbias of each option to both increase and decrease, providing a more flexible adjustment based on the choice history of each participant. This contrasts with models that may only allow for the strengthening of biases over time.

### DIV normalization

The idea behind DIV normalization applied to value-based decision making is that rewards are normalized with respect to other rewards within a context[23,24,39,40]:

$$\delta_{a,t}^{(c_t)} = \frac{r_{a,t}^{(c_t)}}{\sum r^{(c_t)}} - Q_{a,t-1}^{(c_t)} \quad (7)$$

for a = {chosen, unchosen}

particularly both the chosen and unchosen (if full feedback is provided) rewards $r_{a,t}$ within the current context $c_t$ at display are normalized relative to the sum $\sum_{c_t} r^{(c_t)}$ of alternative rewards within the trial-specific prediction error term $\delta_{a,t}^{(c_t)}$. Please note that Q-values are updated as usual according to Eq. 2, but with the modified prediction error term (Eq. 7).

### Range normalization

Range normalization (RANGE) assumes that option values are rescaled as a function of the maximum and the minimum rewards presented in a

context[17]:

$$\delta_{a,t}^{(c_t)} = \left( \frac{r_{a,t} - \min_x \left( r_{x,t}^{(c_t)} \right)}{\max_x \left( r_{x,t}^{(c_t)} \right) - \min_x \left( r_{x,t}^{(c_t)} \right)} \right) - Q_{a,t-1}^{(c_t)} \tag{8}$$
$$\text{for a} = \{\text{chosen, unchosen}\}$$

here, the chosen and unchosen (in case of full feedback) reward $r_{a,t}$ is normalized as a function of the minimum (worst) and maximum (best) reward in the current context $c_t$ (Eq. 8). This normalized reward within the prediction error term is then used to update the Q-values for the corresponding options (Eq. 2).

### Scaled range normalization
Bavard and Palminteri[18] proposed an extension to range normalization by introducing an ω exponent, enhancing the flexibility of the normalization process:

$$\delta_{a,t}^{(c_t)} = \left( \frac{r_{a,t} - \min_x \left( r_{x,t}^{(c_t)} \right)}{\max_x \left( r_{x,t}^{(c_t)} \right) - \min_x \left( r_{x,t}^{(c_t)} \right)} \right)^{\omega} - Q_{a,t-1}^{(c_t)} \tag{9}$$
$$\text{for a} = \{\text{chosen, unchosen}\}$$

In this scaled range normalization (*RANGE ω*) model, the ω parameter allows the model to skew value estimates either concavely or convexly (Eq. 9). This skewing was argued to better account for the non-linearity in choice preferences observed in the datasets analyzed by Bavard and Palminteri[18].

### Intrinsic enhanced reward model (IER)
Molinaro and Collins[19] proposed a goal-centric RL model that integrates an intrinsic reward mechanism. This binary internal signal (0 or 1) corresponds to the goal of selecting the best option in each trial. The model weights this intrinsic signal with external reward feedback, allowing Q-values to be updated as a weighted combination of both outcomes. The underlying idea is that participants prefer options during the transfer phase that are more consistent with their internal goal of choosing the best option:

$$ir_{a=chosen,t}^{(c_t)} = \begin{cases} 0, & \text{if } r_{chosen,t}^{(c_t)} \neq \max_x \left( r_{x,t}^{(c_t)} \right) \text{ or} \left( r_{unchosen,t}^{(c_t)} \text{ is known and } r_{chosen,t}^{(c_t)} < r_{unchosen,t}^{(c_t)} \right) \\ 1, & \text{otherwise} \end{cases} \tag{10}$$

$$ir_{a=unchosen,t}^{(c_t)}$$
$$= \begin{cases} 0, & \text{if } r_{unchosen,t}^{(c_t)} \neq \max_x \left( r_{x,t}^{(c_t)} \right) \text{ or} \left( r_{unchosen,t}^{(c_t)} \text{ is known and } r_{chosen,t}^{(c_t)} > r_{unchosen,t}^{(c_t)} \right) \\ 1, & \text{otherwise} \end{cases} \tag{11}$$

In detail, the trial-wise intrinsic reward signal $ir_{a,t}^{(c_t)}$ for an option is set to 0 if (i) the chosen reward does not equal the maximum obtainable reward in that context or if (ii) the chosen reward turns out to be smaller than the unchosen reward (Eq. 10 and Eq. 11). Otherwise, $ir_{a,t}^{(c_t)}$ is set to 1. The intrinsic reward signal in that context is then weighted relative to the extrinsic reward $r_{a,t}$:

$$ier_{a,t}^{(c_t)} = \omega * ir_{a,t}^{(c_t)} + (1-\omega) * r_{a,t}^{(c_t)} \tag{12}$$
$$\text{for a} = \{\text{chosen, unchosen}\}$$

where ω is a weighting parameter (Eq. 12). Subsequently, Q values are updated according to Eq. 2, but using prediction errors $\delta_{a,t}^{(c_t)}$ based on the

intrinsically enhanced reward $ier_{a,t}^{(c_t)}$ signal (Eq. 13):

$$\delta_{a,t}^{(c_t)} = ier_{a,t}^{(c_t)} - Q_{a,t-1}^{(c_t)} \tag{13}$$
$$\text{for a} = \{\text{chosen, unchosen}\}$$

### Relative model (REL)
In the relative value learning (REL) model proposed by Klein et al.[15], participants are assumed to directly learn a $Q_{difference}$ value for each context $c$:

$$Q_{difference,t}^{(c_t)} = Q_{difference,t-1}^{(c_t)} + \eta * \delta_{difference,t}^{(c_t)} \tag{14}$$

$$\delta_{difference,t}^{(c_t)} = \left( r_{option1,t}^{(c_t)} - r_{option2,t}^{(c_t)} \right) - Q_{difference,t-1}^{(c_t)} \tag{15}$$

where reward information for both options within the current context is used to compute prediction error and update the $Q_{difference}$ estimates. This means that this version of the REL model can only apply to choice contexts with two options.

### Action selection
Action selection for each model was computed using a softmax decision rule[44], which weights Q-values (Eq.16; ABS, DIV, RANGE, RANGEω and IER models), $Q_{difference}$ values (Eq.17; REL model) or Q-values and REPbiases in the case of the REP model (Eq.18), to model the probability of selecting the chosen option on trial $t$.

$$p\left( chosen_t = a | c_t \right) = \frac{\exp\left( \beta * Q_{a,t}^{(c_t)} \right)}{\sum_{a'} \exp\left( \beta * Q_{a',t}^{(c_t)} \right)} \tag{16}$$

$$p\left( option1 | c_t \right)$$
$$= \frac{\exp\left( \beta * (0.5 + Q_{difference,t}^{(c_t)}) \right)}{\left( \exp\left( \beta * (0.5 + Q_{difference,t}^{(c_t)}) \right) + \exp\left( \beta * \left( (0.5 - Q_{difference,t}^{(c_t)}) \right) \right) \right)} \tag{17}$$

$$p\left( chosen_t = a | c_t \right) = \frac{\exp\left( \beta * (Q_{a,t}^{(c_t)} + REPbias_{a,t}^{(c_t)}) \right)}{\sum_{a'} \exp\left( \beta * (Q_{a',t}^{(c_t)} + REPbias_{a',t}^{(c_t)}) \right)} \tag{18}$$

All three softmax functions are parameterized by the inverse temperature $\beta$. When $\beta = 0$, choices are random, and as $\beta$ increases, choices become more dependent on learned Q-values.

For probabilistic tasks, following Klein and colleagues[15] and as in the standard analysis (see above), we focused our analysis on the first two transfer trials for each choice context. Previous studies have similarly focused on the initial decisions in the transfer phase or included only two or four decisions per transfer pair[15,16,18]. For Gaussian tasks, we included all four transfer trials for each context. For our probabilistic tasks, learning mechanisms were applied to the learning and transfer phases, as feedback was provided during transfer. The repetition-bias mechanism was applied during learning and transfer for all tasks. Note that the results did not differ from the case when the repetition bias was only applied to the learning phase (see Supplementary Table 3).

### Modeling partial feedback
In the partial feedback condition, reward information for alternative (unchosen) options, such as their outcomes (rewards) or the range of possible alternative rewards, was unavailable to participants. This limitation presents a challenge for accurately updating Q values and applying normalization or weighting in the DIV, RANGE, RANGEω, and IER models.

Past studies have addressed this issue in different ways. Some researchers treated the learned and stored $Q$ values as valid proxies for the unobserved rewards. These proxy values were then used as inputs for normalizing within the prediction error terms[16,45]. Others opted for computational simplicity by allowing models to retain full knowledge of contextual information, such as the minimum and maximum possible rewards, even though participants did not receive complete feedback[19]. For comparing these models in our study, we addressed the challenge of partial feedback for these models by implementing a memory system for the DIV, REL, RANGE, RANGEω, and IER models. When participants received partial feedback, the most recent known reward information for each option was retrieved to inform their computations. Additionally, as proposed by Molinaro and Collins[19], we calculated the IER signal exclusively for the chosen option when full feedback was unavailable. Note that the REP model computations are in all conditions, including partial feedback, only based on information that is always available to the agent.

### Hierarchical Bayesian inference

Models were fit to all trials from all participants using a hierarchical Bayesian modeling approach. Parameter inference was performed using Markov Chain Monte Carlo sampling as implemented in the JAGS software package (Plummer, 2003) (Version 4.3) in combination with R (Version 4.3) and the *R2Jags* (0.7.1.1) package. For group-level means, we used uniform priors defined over numerically plausible parameter ranges ($[−5, 5]$ for the learning- ($\eta$) and baseline learning-rates $b^{(c)}$ in standard-normal space; $[0, 5]$ for $\beta$ and $[0,3]$ for the hyperbolic decay $d^{(c)}$. Group-level standard deviations were likewise modeled via uniform priors over a plausible parameter range. For full model code and parametrization, see the code availability section. We initially ran two chains with a varying burn-in period of at least 20,000 samples and thinning of two. Chain convergence was assessed via the Gelman-Rubinstein convergence diagnostic $\hat{R}$ and sampling was continued until $1 \leq \hat{R} \leq 1.05$ for all group-level and individual-subject parameters. In some cases, learning-rate parameters did exceed that threshold; however, all parameters had a $\hat{R} \leq 1.1$. 15.000 additional samples were retained for further analysis.

### Model comparison

The ABS, REP and IER models were compared across all tasks. For tasks with two options per context and equal reward magnitude (p1, p2, p3, p4.1, p4.2 and Exp. 2 and Exp3 from Klein and colleagues[15]), we additionally used the REL model. For tasks with more than two options per context and/or where the range of reward magnitudes differed between contexts or options (tasks g1, g2, and datasets from Bavard and colleagues[16,18]), we additionally applied divisive normalization (DIV)[39,42] and range normalization models (RANGE & RANGE $\omega$)[17,18]. Quantitative model comparison was performed via the Widely Applicable Information Criterion (WAIC) where lower values indicate better fit[46,47].

### Model and parameter recovery

To assess model identifiability, we conducted full model confusion on four representative datasets that differ in their reward-generating process (probabilistic or Gaussian) and model predictions (Task p3, Task p4.1, Task g1 and Exp2 from Bavard et al.[16]). In addition, we performed parameter recovery for the REP model on two representative datasets that differ on the number of cross context comparisons and task dynamics (Task p3 and Exp2 from Bavard et al.[16]). In our procedure, we generated posterior-predictive data during MCMC: at each post-warm-up draw, we simulated a full dataset from the likelihood conditional on that joint parameter draw ($\beta$-weights, context-specific decay rates, repetition learning-rate $\eta$) and therefore yielded 10,000 posterior-predictive datasets for each model. From this posterior predictive pool, we randomly sampled 20 independent datasets per model (subject × trial matrices) for confusion and 10 datasets for parameter recovery. For model recovery, each replicate was then re-analyzed with the entire model battery (ABS, REP, IER, REL for task p3 and p4.1 and ABS,

REP, IER, NORM, RAN, RAN-ω for task g1 and Exp2 from Bavard et al.[16]) using equal uninformative priors, sampler settings and convergence criteria of the main fits. Model recovery was evaluated with WAIC; the model with the lowest score in a replicate was deemed its winner. To assess parameter recovery, we computed correlations between the known parameter values used to generate synthetic datasets and the posterior means obtained from fitting the model to these datasets.

## Results

### Standard analyses

We first perform a descriptive assessment of the choice data during learning and test whether participants show any systematic preference during the beginning of the transfer phase. We then present three key results from our standard analyses across all 15 datasets: (i) there is evidence for a correlation between participants' relative choice frequencies in the learning phase and their choice preferences in the transfer phase; (ii) the relative choice frequency during learning is linked to post-task valuation of options, specifically, options chosen more frequently are overvalued compared to those chosen less frequently; and (iii) the relative choice frequency also relates to participants' certainty about an option's value, with participants being more certain about the value of options they have chosen more often during learning.

### Assessment of preferences

In those tasks, where both available contexts differed primarily by contrast (p1 and p2 or g1 and g2) participants on average chose the best option, i.e. the option with the highest expected value, $C_{HC}(0.7)$ or $D_{HC}(0.7)$) in the high contrast (HC; see Fig. 1A) context more often than the best option ($A_{LC}(0.7)$) in the LC context (see Fig. 2A–C, F, G). Similarly, and as expected, during transfer, most participants preferred the best HC option ($C_{HC}$ in probabilistic tasks p1 and p2 or $D_{HC}$ in Gaussian tasks g1 and g2) over its equally valued counterpart ($A_{LC}$). To quantify those preferences we report the proportion $\hat{p}$ choosing of choosing the target option ($C_{HC}$ in probabilistic tasks; $D_{HC}$ in Gaussian tasks) in the first transfer trial (TT) together with its 95% exact binomial confidence interval; p-values are from two-sided exact binomial tests of $H_0$: $p = 0$ (Task p1(full feedback): $\hat{p}(C_{HC}) = 0.73$ $[0.63−0.82]$; $p < 0.0001$; Task p2(partial feedback): $\hat{p}(C_{HC}) = 0.63$ $[0.48−0.77]$; $p = 0.08$; Task g1(full feedback): $\hat{p}(D_{HC}) = 0.74$ $[0.59−0.85]$; $p < 0.001$; Task g2 (partial feedback): $\hat{p}(D_{HC}) = 0.74$ $[0.60−0.86]$; $p < 0.001$; see Fig. 2H, I, M, N). Strikingly, in task p3 (Fig. 2J), this relationship reverses when the LC context was encountered more often during learning (50 times vs. 30 times) and thus the best LC option was chosen more often than the best HC option ($\hat{p}(C_{HC}) = 0.29$ $[0.16−0.43]$; $p = 0.004$). This shows that the HC effect is marginalized by a higher number of repetitions. Tasks p4.1 and p4.2 included two choice contexts with the same relative value differences but different absolute magnitudes (see Fig. 1). In task p4.1, where participants chose options, during learning, in approximately the same relative choice frequency (green and orange lines in Fig. 2D), we do not find evidence for clear choice preference during transfer even if expected values favor a high gain (HG) option over a low gain (LG) option (Fig. 2K; $\hat{p}(C_{HG}) = 0.5$ $[0.31−0.69]$; $p = 1$;). In contrast, in task p4.2 (Fig. 2E, L), choice preference during transfer is clearly visible ($\hat{p}(C_{HG}) = 0.83$ $[0.64−0.94]$; $p < 0.001$; Fig. 2L), when participants had a clear favorite option during learning, as measured by within-context relative choice frequency (Fig. 2E). We will elaborate on differences between task p4.1 and p4.2 below. In terms of probabilistic tasks, effects were larger when full feedback was provided and diminished in the partial feedback task. This can be seen in Fig. 2A (full feedback) and B (partial feedback) for the average within-context relative choice frequencies during learning, i.e., red ($C_{HC}$) and blue ($A_{LC}$) lines. This difference is numerically more pronounced when full feedback is provided (Fig. 2A). Likewise, the preference for this HC option (red; $C_{HC}$) over the LC option (blue; $A_{HC}$) during transfer is more pronounced in the full- (p1: $p < 0.0001$; CI$[0.63−0.82]$) compared to partial feedback task (p2; $p = 0.08$; CI$[0.48−0.77]$; see Fig. 2H, I; see Table 3 for an overview).

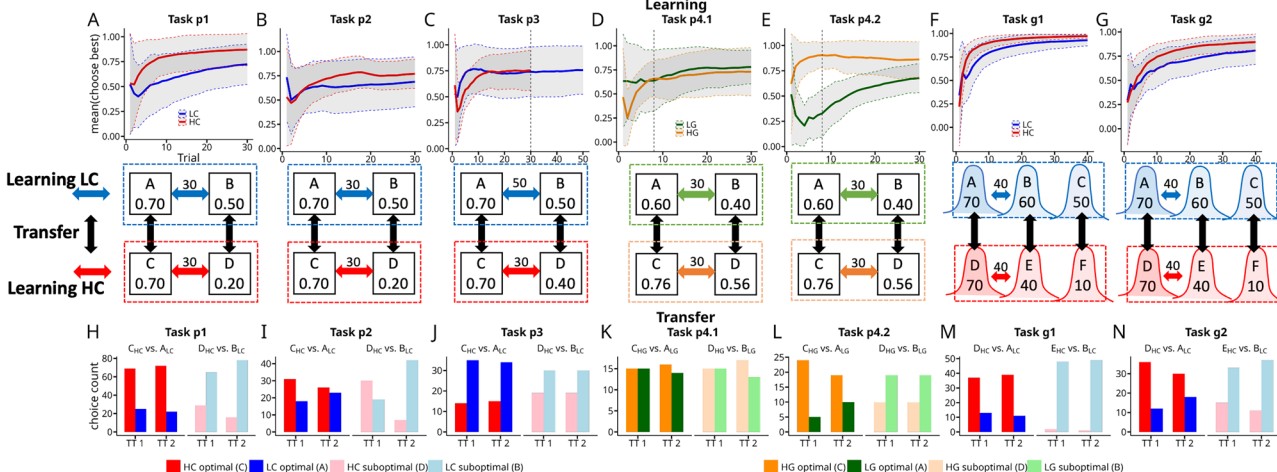

**Fig. 2 | Relative choice frequency (within context) during learning and choice preference during transfer.** Top row (A–G) Relative choice frequency (within context) during learning across all probabilistic and Gaussian tasks. **A** Task p1 (full feedback), relative choice frequency for the best option in both the low contrast (LC; blue) and high contrast (HC; red) contexts. **B** Task p2 (partial feedback), relative choice frequency for the best option in the LC (blue) and HC (red) contexts. **C** Task p3 (full feedback), relative choice frequency for the best option in LC (blue) and HC (red) contexts. **D** Task p4.1 (full feedback), relative choice frequency for the best option in low gain (LG; green) and high gain (HG; orange) contexts. **E** Task p4.2 (full feedback), relative choice frequency for the best option in LG (green) and HG (orange) contexts. **F** Task g1 (full feedback), relative choice frequency for the best option in LC (blue) and HC (red) contexts. **G** Task g2 (partial feedback), relative choice frequency for the best option in LC (blue) and HC (red) contexts. Middle row (A–G) Task parameters and context repetition counts. **A** Task p1 (collapsed p.1 to p.3), reward probabilities and number of repetitions of LC and HC contexts (30 trials for each context during learning). In the LC context, option $A_{LC}$ had a 0.70 reward probability, and option $B_{LC}$ 0.50. In the HC context, option $C_{HC}$ had a 0.70 and $D_{HC}$

a 0.20 reward probability. **B–E** Tasks p2, p3, p4.1, p4.2, reward probabilities and number of repetitions of each context, as in (A). **F** Task g1, reward magnitudes for Gaussian tasks, represented by points. In the LC context, options A, B, and C offered average rewards of 70, 60, and 50 points, respectively. In the HC context, options D, E, and F offered 70, 40, and 10 points. **G** Task g2, analogous to panel F but with partial feedback. Bottom row (H–N) Choice counts for available options within the first and second transfer trial (TT), illustrating participants' choice preferences when options from different contexts are presented together. **H** Task p1, absolute preference in TT1 and TT2 for the best option in the HC context (red, option $C_{HC}$) and the optimal option in the LC context (blue, option $A_{LC}$), as well as for the options with lower reward probabilities ($B_{LC}$ (pink) and $D_{HC}$ (light blue)). **I** Choice counts in TT1 and TT2 in task p2 (partial feedback). **J** Choice counts in TT1 and TT2 in task p3 (full feedback). Here, participants more often chose the best LC option ($A_{LC}$), indicating an effect of repetition during learning. **K** Choice counts in TT1 and TT2 in task p4.1 (full feedback). **L** Choice counts in TT1 and TT2 in task p4.2 (full feedback). **M** Choice counts in TT1 and TT2 in task g1 (full feedback). **N** Choice counts in TT1 and TT2 in task g2 (partial feedback).

**Table 3 | Results of two-sided binomial tests on the first transfer trial**

| Task | Comparison | p̂ (stimulus) | 95% binomial CI | *p* (two-sided) | *n* |
|---|---|---|---|---|---|
| p1(p1.1–p.1.3) | $C_{HC}$ vs $A_{LC}$ | 0.73 ($C_{HC}$) | [0.63–0.82] | *p < 0.0001* | *n* = 94 |
| p2 | $C_{HC}$ vs $A_{LC}$ | 0.63 ($C_{HC}$) | [0.48–0.77] | *p = 0.08* | *n* = 49 |
| p3 | $C_{HC}$ vs $A_{LC}$ | 0.29 ($C_{HC}$) | [0.16–0.43] | *p = 0.004* | *n* = 49 |
| p4.1 | $C_{HG}$ vs $A_{LG}$ | 0.5 ($C_{HG}$) | [0.31–0.6] | *p = 1* | *n* = 30 |
| p4.2 | $C_{HG}$ vs $A_{LG}$ | 0.83 ($C_{HG}$) | [0.64–0.94] | *p < 0.001* | *n* = 29 |
| g1 | $D_{HC}$ vs $A_{LC}$ | 0.74 ($D_{HC}$) | [0.59–0.85] | *p < 0.001* | *n* = 50 |
| g2 | $D_{HC}$ vs $A_{LC}$ | 0.75 ($D_{HC}$) | [0.60–0.86] | *p < 0.001* | *n* = 48 |

For each task, we test whether the proportion of participants choosing stimulus C (tasks p1–p4.2) or stimulus D (tasks g1 and g2) over stimulus A differs from chance (H₀: *p* = 0.5; the two stimuli have either equal absolute values in experiments p1, p2, p3, g1, and g2 or equal relative values in experiments p4.1 and p4.2). For each task, we show p̂, the observed choice proportions and the 95% binomial confidence interval. A *p* value < 0.05 (italic) indicates a significant group-level bias in favor of one stimulus.

**Frequency correlation between learning and transfer**
We next analyzed whether relative choice frequency (across learning contexts: e.g., $A_{LC}$ vs $C_{HC}$) during learning and relative choice preference during transfer (within transfer context: e.g., $A_{LC}$ vs $C_{HC}$)) are significantly correlated. To do this, we computed, for the learning phase, relative choice frequencies for all pairwise combinations of options encountered in the transfer phase (two cross-context comparisons for our probabilistic tasks and nine cross-context comparisons in Gaussian tasks). For example, in task p1 (Fig. 2A, H), we calculated how often participants chose, during learning, option $C_{HC}$ relative to how often they chose option $A_{LC}$ (relative frequency learning; see methods on binomial tests & correlations). Note that by design options $C_{HC}$ and $A_{LC}$ were never paired in the same trial during learning, but one can compute their relative choice frequencies, i.e., how often each option was selected in relation to the other. Analogously, for the transfer phase, we computed

how often participants preferred $C_{HC}$ over $A_{LC}$ during the first two transfer trials when directly compared against each other (relative frequency transfer). We found that relative choice frequency during learning and relative choice preference during transfer was highly correlated across all datasets (pooled data across all probabilistic tasks incl. Klein et al. (2017); *r* = 0.79, posterior credible interval (CI): [0.53, 0.93]; BF₁₀: = 671), Gaussian tasks (ρ = 0.97 [0.94, 0.99] BF₁₀: >1000), both datasets from Bavard et al.[16] (Exp1: *r* = 0.69 [0.46, 0,85] BF₁₀: >1000; Exp2: ρ = 0.81 [0.64, 0.91] BF₁₀: >1000) and two datasets from Bavard et al. (2023; Exp3a: ρ = 0.91 [0.86, 0.95] BF₁₀: >1000; Exp3b: ρ = 0.86 [0.78, 0.91] BF₁₀: >1000). However, computing correlations across all pairwise combinations can be problematic, as these correlations are influenced by expected reward. For example, in tasks p1 and p2, participants might choose option $B_{LC}(0.5)$ more frequently than option $D_{HC}(0.2)$ during learning, solely because $B_{LC}(0.5)$ has a higher expected

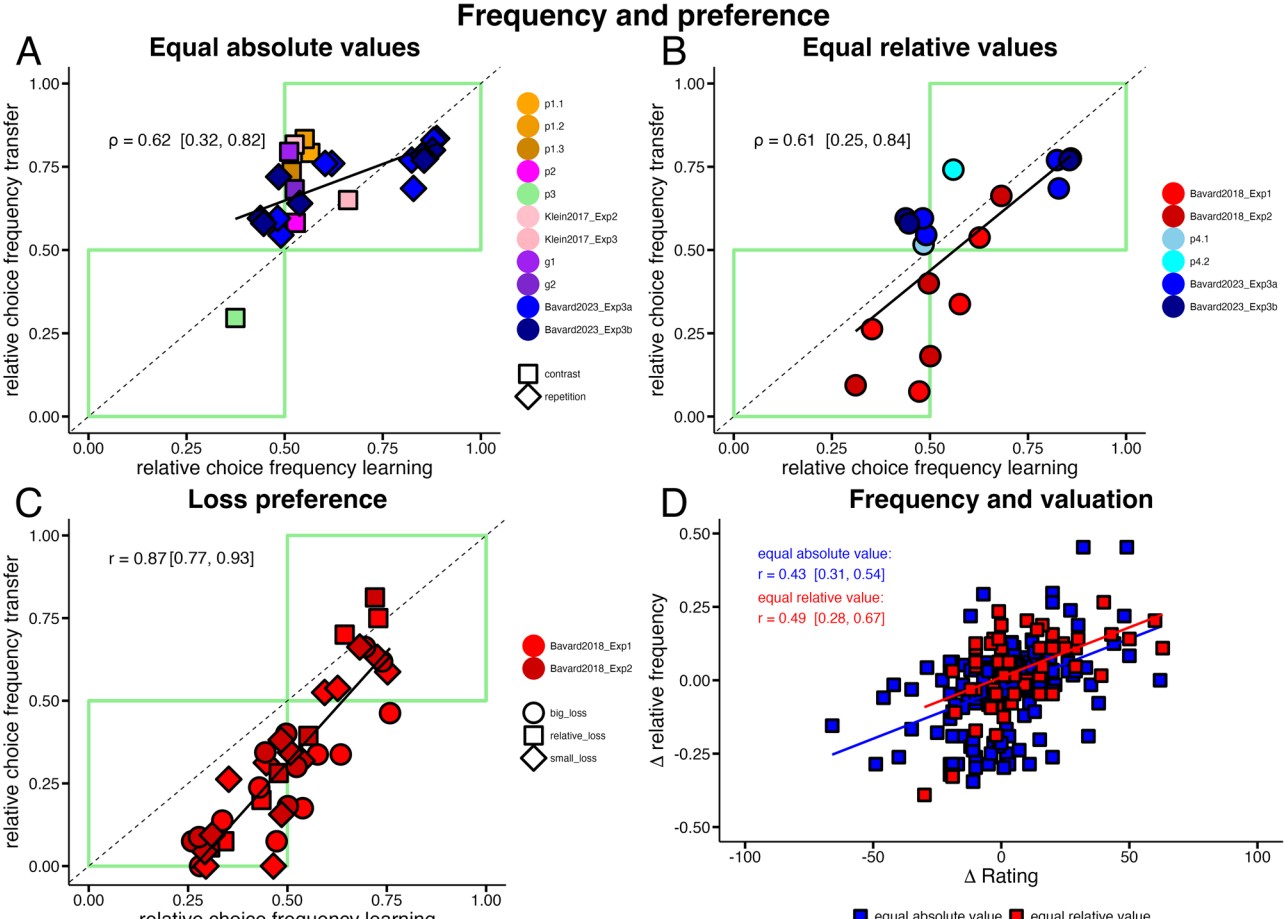

**Fig. 3 | Correlation between learning-phase choice frequency and transfer-phase choice preferences. A–C** Linear relationships between relative choice frequency during the learning phase (x axis; relative frequency learning) and preference during the transfer phase (y axis; relative frequency transfer) for selected pairwise combinations of options participants encountered in the transfer phase. Data points (colored dots; colors indicate different datasets) represent relative choice frequencies, see methods on binomial tests & correlations. Green rectangles: Highlights data points where an above (upper right) or below (lower left) 50% relative choice frequency, during learning, is associated with an above or below 50% relative choice frequency during transfer. **A** Relative choice frequency of option pairs with equal expected values. Nearly all pairwise combinations fall within the upper right green rectangle, indicating that options preferred during the transfer phase (>50%) were chosen more frequently than their counterpart (>50%) during the learning phase. Bayesian correlation analysis showed strong evidence for a correlation of relative choice frequency during learning and transfer ($\rho$ = 0.62, 95% posterior credible interval (CI): [0.32, 0.82], n = 25 comparisons). **B** Relative choice frequencies of option pairs with equal relative values. Results indicate that even when relative values are equal, relative choice frequency during learning is related to relative choice frequency (preference) in transfer ($\rho$ = 0.61, CI: [0.25, 0.84], n = 18 comparisons). **C** Loss preference: option pairs where losses are compared against gains, from Bavard et al.[16] This panel illustrates that preferences for losses or smaller gains (relative losses) during transfer follow the same positive relative frequency learning —relative frequency transfer relationship (r = 0.87, CI: [0.77, 0.93], n = 40 comparisons). Data points are differentiated by shape according to three categories to illustrate specific influences on decision-making: small loss category: Comparisons between an option associated with a small loss (−0.1€) and options associated with a

gain (+0.1€, +1€). X values are relative frequencies for the small loss when normalized with respect to the compared gain option. Relative loss category: Comparisons between an option associated with a small gain (0.1€) and options associated with a big gain (+1€). X-values represent relative choice frequencies for the small gain option when normalized with respect to the compared large gain option. Big loss category: Comparisons between an option associated with a big loss (-1€) and a small or big gain (0.1€, 1€). X values are relative frequencies for the big loss when normalized with respect to the compared small or big gain option (see Bavard et al.[16] for further details). All loss-associated options preferred during the transfer phase (as indicated by their placement within the upper right green rectangle) were chosen more frequently (above 50%) during learning than their higher-valued counterparts. Options outside the green rectangles do not necessarily violate the linear relationship between relative choice frequency during learning and transfer; instead, they likely reflect the dominance of expected value effects over frequency. **D** Relationship between differences in normalized choice frequencies (Δ relative frequency; see methods on frequency and valuation) and post-task option valuation (Δ rating; difference in rating between both options). Data points (participants) are color-coded based on whether the choices involved only equally valued options (blue, $C_{HC}$ vs. $A_{LC}$ in probabilistic tasks or $D_{HC}$ vs. $A_{LC}$ in Gaussian tasks; r = 0.43 CI: [0.31,0.54], n = 196 comparisons) or options with equal relative value (red, $C_{HG}$ vs. $A_{LG}$ from tasks p4.1 an p4.2; r = 0.49, CI: [0.28,0.67], n = 59 comparisons). This plot shows how differences in normalized choice frequency of two options correlate with differences in participants' post-task valuation of the two options. The positive association of differences in normalized frequency and option valuation indicates that valuation is positively related to choice frequency.

value. Similarly, if successfully learned, participants may prefer option $B_{LC}$(0.5) over $D_{HC}$(0.2) during transfer because of the difference in expected value. To address this issue, we aimed to control for such effects by focusing our analysis on three different subsets of the data. First, we selected only those option pairs from all 15 datasets where both options

had equal absolute expected values by design (see Methods: Bbinomial tests & correlations). We still found strong evidence for a correlation between choice frequency during learning and choice preference during transfer ($\rho$ = 0.62, [0.32, 0.82] $BF_{10}$: 109; Fig. 3A). The presence of this correlation indicates that participants' choice preference for an option

**Table 4 | Hierarchical (multi-level) logistic regressions linking learning-phase choice frequency (FreqLearn) to transfer phase choice preference (Choice[HC/HG])**

| Dataset | Task | Term | Estimate | 2.5% | 97.5% | BF₁₀ | Sig95 |
|---|---|---|---|---|---|---|---|
| all pairs | Pooled data | b_Intercept | −2.269 | −2.404 | −2.137 | >1000 | ✓ |
| all pairs | Pooled data | b_Slope | 3.879 | 3.653 | 4.114 | >1000 | ✓ |
| all pairs | Bavard et al.[16] | b_Intercept | −3.249 | −3.623 | −2.848 | >1000 | ✓ |
| all pairs | Bavard et al.[16] | b_Slope | 5.107 | 4.337 | 5.827 | >1000 | ✓ |
| all pairs | Bavard et al.[18] | b_Intercept | −2.176 | −2.316 | −2.037 | >1000 | ✓ |
| all pairs | Bavard et al.[18] | b_Slope | 3.682 | 3.444 | 3.920 | >1000 | ✓ |
| all pairs | Gaussian tasks | b_Intercept | −3.671 | −4.106 | −3.254 | >1000 | ✓ |
| all pairs | Gaussian tasks | b_Slope | 8.220 | 7.325 | 9.130 | >1000 | ✓ |
| all pairs | Probabilistic tasks | b_Intercept | −0.841 | −1.254 | −0.430 | >1000 | ✓ |
| all pairs | Probabilistic tasks | b_Slope | 2.139 | 1.336 | 2.949 | >1000 | ✓ |
| equal abs value | Pooled data | b_Intercept | −0.455 | −0.696 | −0.215 | >1000 | ✓ |
| equal abs value | Pooled data | b_Slope | 2.349 | 1.937 | 2.774 | >1000 | ✓ |
| equal abs value | Bavard et al.[18] | b_Intercept | −0.362 | −0.632 | −0.099 | >1000 | ✓ |
| equal abs value | Bavard et al.[18] | b_Slope | 2.244 | 1.809 | 2.701 | >1000 | ✓ |
| equal abs value | Gaussian tasks | b_Intercept | 0.736 | −0.414 | 1.877 | 0.9711 | |
| equal abs value | Gaussian tasks | b_Slope | 0.695 | −1.254 | 2.658 | 0.9711 | |
| equal abs value | Probabilistic tasks | b_Intercept | −0.670 | −1.470 | 0.136 | 82.9645 | |
| equal abs value | Probabilistic tasks | b_Slope | 2.377 | 0.925 | 3.875 | 82.9645 | ✓ |
| equal rel value | Pooled data | b_Intercept | −0.699 | −0.981 | −0.419 | >1000 | ✓ |
| equal rel value | Pooled data | b_Slope | 1.976 | 1.523 | 2.442 | >1000 | ✓ |
| equal rel value | Bavard et al.[16] | b_Intercept | −2.097 | −2.790 | −1.424 | >1000 | ✓ |
| equal rel value | Bavard et al.[16] | b_Slope | 2.783 | 1.679 | 3.935 | >1000 | ✓ |
| equal rel value | Bavard et al.[18] | b_Intercept | −0.207 | −0.522 | 0.113 | >1000 | |
| equal rel value | Bavard et al.[18] | b_Slope | 1.598 | 1.079 | 2.128 | >1000 | ✓ |
| equal rel value | Probabilistic tasks | b_Intercept | 0.322 | −0.868 | 1.568 | 1.2414 | |
| equal rel value | Probabilistic tasks | b_Slope | 0.795 | −1.030 | 2.653 | 1.2414 | |

Posterior summaries are shown for the fixed-effects intercept (b_Intercept) and the slope of FreqLearn (the normalized frequency during learning). Estimate: Posterior mean of the regression coefficient (on the log-odds scale); 2.5% and 97.5% = lower and upper bounds of the 95% posterior credible interval (CI). BF₁₀ = Bayes factor favoring the model that contains the group and random level slope and intercept over a null model containing only group and random level intercepts. Values ">1000" indicate BF₁₀ ≥1000. A ✓ in the Sig95 column marks coefficients whose 95% CI does not include zero. Choice set indicates the subset of cross-context stimulus pairs that was analyzed ("all pairs", "equal absolute value", "equal relative value"). Study/Task names the experiments: "Pooled data" refers to the joint analysis across comparisons and datasets.

during transfer is related to the relative choice frequency for that option during learning (relative to the compared transfer option with equal expected value).

Second, one might argue that even though the objective expected values of the compared options were equal, preferences could still be influenced by relative values, i.e., implying that humans learn values on a relative scale, as assumed by state-dependent normalization[15,16]. To address this possibility, we selected only those option pairs from all 15 datasets where both options had the same normalized (relative) values (see methods binomial tests & correlations). We found that the relative choice frequency during learning and the relative choice frequency during transfer remained robustly correlated ($\rho = 0.61$, CI: [0.25, 0.84]; BF₁₀: 23 see Fig. 3B). This further indicates that frequency is associated with preference formation. If frequency had no effect on preference, or if participants relied exclusively on state-dependent normalization, we would expect no correlation in this case (see Supplementary Table 2 for an overview). Note that, by design, via controlled trial sequences and, in most tasks, full feedback, those matched pairs had equal absolute- and equal context-normalized values, yielding equal ground-truth expected values (EVs) for both options in each comparison. Therefore, those analyses above assume that participants' internal estimates tracked these EVs. To address potential confounds and subject-level variability, we present hierarchical regressions below and control analyses based on model-derived Q-values in the Supplement (Supplementary Figs. 15–17). Third, to provide evidence that loss preference during transfer, as reported by Bavard

and colleagues[16], might as well be related to choice frequency during learning, we selected all option pairs from their study where participants had to choose between gains and losses of varying magnitudes during the transfer phase. Analysis of these selected data revealed a strong correlation between choice frequency during learning and choice preference during transfer ($r = 0.87$, CI: [0.77, 0.93]; BF₁₀: >1000; see Fig. 3C). Importantly, when an option associated with a loss (small loss, relative loss, or big loss) had a higher choice frequency relative to an option associated with a gain during transfer, we found that such a loss option always had a higher relative choice frequency during learning (all red datapoints above 0.5 (y-axis; relative frequency transfer) fall within the upper right green rectangle).

**Hierarchical mixed-effects regression**

To further quantify the relationship of choice frequency during learning on transfer preferences, we conducted a series of Bayesian hierarchical mixed-effects logistic regression analyses that accounted for subject-level variability via random intercepts and slopes. We therefore partitioned decisions into three subsets: (1) all cross-context comparisons, (2) pairs with equal absolute values, and (3) pairs with equal relative values (based on context-specific normalized values; see methods on mixed-effects Bayesian logistic regressions for details). These analyses were carried out separately for our two newly collected task classes (probabilistic and Gaussian) as well as for both studies from Bavard and colleagues[16,18]. Across the pooled datasets and individual studies, 95% posterior credible intervals (CIs) for slopes were

**Fig. 4 | Mixed effects Bayesian logistic regressions to model the effect of choice frequency during learning on preference in transfer.** We here analyzed three pooled choice sets ("all pairs", "equal absolute value", "equal relative value") across all new datasets and published studies to quantify whether preference in the first transfer trial is related to relative choice frequency during learning. Rows distinguish group-level Intercept vs. Slope parameters. Points are posterior means; red intervals show parameter estimates whose 95% CI excludes 0 (i.e., credible frequency (repetition) effects), gray intervals represent intervals overlapping with 0. Intervals for the Slope entirely to the right show a positive frequency/repetition preference relation, i.e., that increased relative choice frequency for an option during learning is associated with that option being chosen at transfer. For control analyses, e.g., without partial feedback tasks or with estimated Q value differences as an additional regressor, see Supplementary Information (Supplementary Figs. 1 and 18).

consistently positive in the majority of cases. A significant positive slope indicates that higher relative choice frequency during learning was associated with a greater probability of choosing that option during transfer. Bayes factor comparisons against an intercept-only model strongly favored the inclusion of a frequency slope term. Specifically, we found robust positive slope estimates (95% CIs not overlapping zero) in the pooled dataset, our probabilistic tasks (BF = 83), and in the Bavard and Palminteri 2023[18] dataset for both the equal and absolute value subsets (BFs > 1000). Similarly, the pooled data, Bavard and colleagues 2018[16] and Bavard and Palminteri 2023[18] showed positive slope estimates for equal relative value comparisons (BFs > 1000). In contrast, the 95% CIs overlapped with zero in the Gaussian tasks (equal absolute value subset; BF = 0.97) and in the probabilistic tasks (equal relative value subset; BF = 1.24), likely reflecting limited statistical power due to the small number of relevant comparisons. In these subsets, only two specific transfer decisions contributed data ($A_{LC}$ vs. $D_{HC}$ in both Gaussian tasks and $A_{LG}$ vs. $C_{LG}$ in p4.1 and p4.2), which constrains interpretability. Nevertheless, numerical trends in these subsets were consistent with a frequency-based bias in preference formation. These findings provide converging evidence across modeling approaches, value-matched comparisons, and datasets, that choice frequency during learning is associated with subsequent preference, even in the absence of objective value differences (see Fig. 4 for an overview and Table 4 for all parameter estimates and Bayes factors for model comparison). Please note that this regression analysis does not control for potential value distortions, such as different forms of normalization or asymmetric learning (for control analyses, including learned values, please see Supplementary Figs. 15–18).

### Frequency and valuation

If a higher relative choice frequency of an option causes preference for that option during the transfer phase, one would also expect this to be reflected in participants' valuation of options. Specifically, we computed relative choice frequencies at the participant level and assessed whether differences in relative choice frequency between two options were associated with differences in their post-task valuation (see methods on frequency and valuation for details). To control for the obvious influence of reward, we focused our

analysis as above to pairwise comparisons of options with equal absolute and relative values. In tasks with such pairs (p2, p3, g1, g2), we found that differences in relative choice frequency were robustly correlated with differences in post-task valuation. This correlation was robust whether relative choice frequency was computed from the combined learning and transfer phase data ($r = 0.43$ CI: [0.31,0.54]; $BF_{10}$: >1000; blue squares in Fig. 3D or from the learning phase data alone ($r = 0.41$ CI: [0.29,0.52]; $BF_{10}$: >1000). We next focused on options with equal relative value (tasks p4.1 and p4.2; $C_{HG}(0.76; +0.1$ relative value) and $A_{LG}(0.6; +0.1$ relative value)) and likewise computed the correlation between differences in relative choice frequency and differences in post-task valuation. Again, we found strong evidence for a positive correlation when using combined learning and transfer phase data ($r = 0.49$, CI: [0.28,0.67]; $BF_{10}$: 686; red squares in Fig. 3D) and learning phase data only ($r = 0.45$ CI: [0.24,0.63]; $BF_{10}$: 169). These findings indicate that participants' relative choice frequency is linked to their post-task valuation of options. When directly compared, options chosen more frequently were overvalued compared to those chosen less frequently. Notably, absolute Q value estimates at the end of learning did not differ between these options (see Supplementary Figs. 15 and 16).

### Frequency and uncertainty

Similarly, we analyzed whether relative choice frequency is related to uncertainty about the value of options (post-task valuation rating). In the probabilistic learning tasks the spread of the belief distribution (standard deviation of ratings) was negatively correlated with relative choice frequency (p2, p3, p4.1 and p4.2; $\rho = -0.56$, 95% CI: [$-0.83, -0.16$]; $BF_{10}$: 9; Fig. 5G) For Gaussian tasks this was also negative, but evidence was only anecdotal (g1 and g2; $\rho = -0.44$, 95% CI: [$-0.80, 0.09$]; $BF_{10}$: 1.7; Fig. 5F). In addition, on the individual participant level higher relative choice frequency was related to lower distance from the true expected value (probabilistic tasks) or from the mean of the Gaussian reward magnitudes (Gaussian tasks) when correlated across all participants and tasks ($r = -0.11$; 95% CI[$-0.19, -0.03$]; $BF_{10}$:2.0).

### Computational modeling

To better understand the mechanisms driving the correlation between choice frequency during learning and choice preference during transfer, we analyzed choice data from all tasks using a computational model and compared its performance against all prominent alternative models[15,19,23]. The proposed model integrates two key components: (i) standard RL and (ii) reward-independent learning of a repetition bias[10,12,29,30,33]. For the first component, we assumed participants learn Q-values for each option in an unbiased manner, as would be assumed in standard RL[44]. We here tested two different model variants. For the first, we used different decaying learning rates for each context[44], and for the second, following previous studies[16,18,45], different learning rates for chosen and unchosen options (see methods on RL models above). For the second component, we let choices increase or decrease the repetition bias of an option:

$$REPbias_{a=chosen,t}^{(c_t)} = REPbias_{a=chosen,t-1}^{(c_t)} + \eta_{repetition} * \left( 1 - REPbias_{a=chosen,t-1}^{(c_t)} \right)$$
(19)

$$REPbias_{a=unchosen,t}^{(c_t)} = REPbias_{a=unchosen,t-1}^{(c_t)} + \eta_{repetition} * \left( 0 - REPbias_{a=unchosen,t-1}^{(c_t)} \right)$$
(20)

$$p\left( chosen_t = a | c_t \right) = \frac{\exp\left( \beta * \left( Q_{a,t}^{(c_t)} + REPbias_{a,t}^{(c_t)} \right) \right)}{\sum_{a'} \exp\left( \beta * \left( Q_{a',t}^{(c_t)} + REPbias_{a',t}^{(c_t)} \right) \right)}$$
(21)

where the repetition-bias (REPbias) for each option of the current context is updated using a separate learning rate parameter $\eta_{repetition}$. The REPbias for an option increases on a trial-wise basis when an option is chosen (Eq. 19)

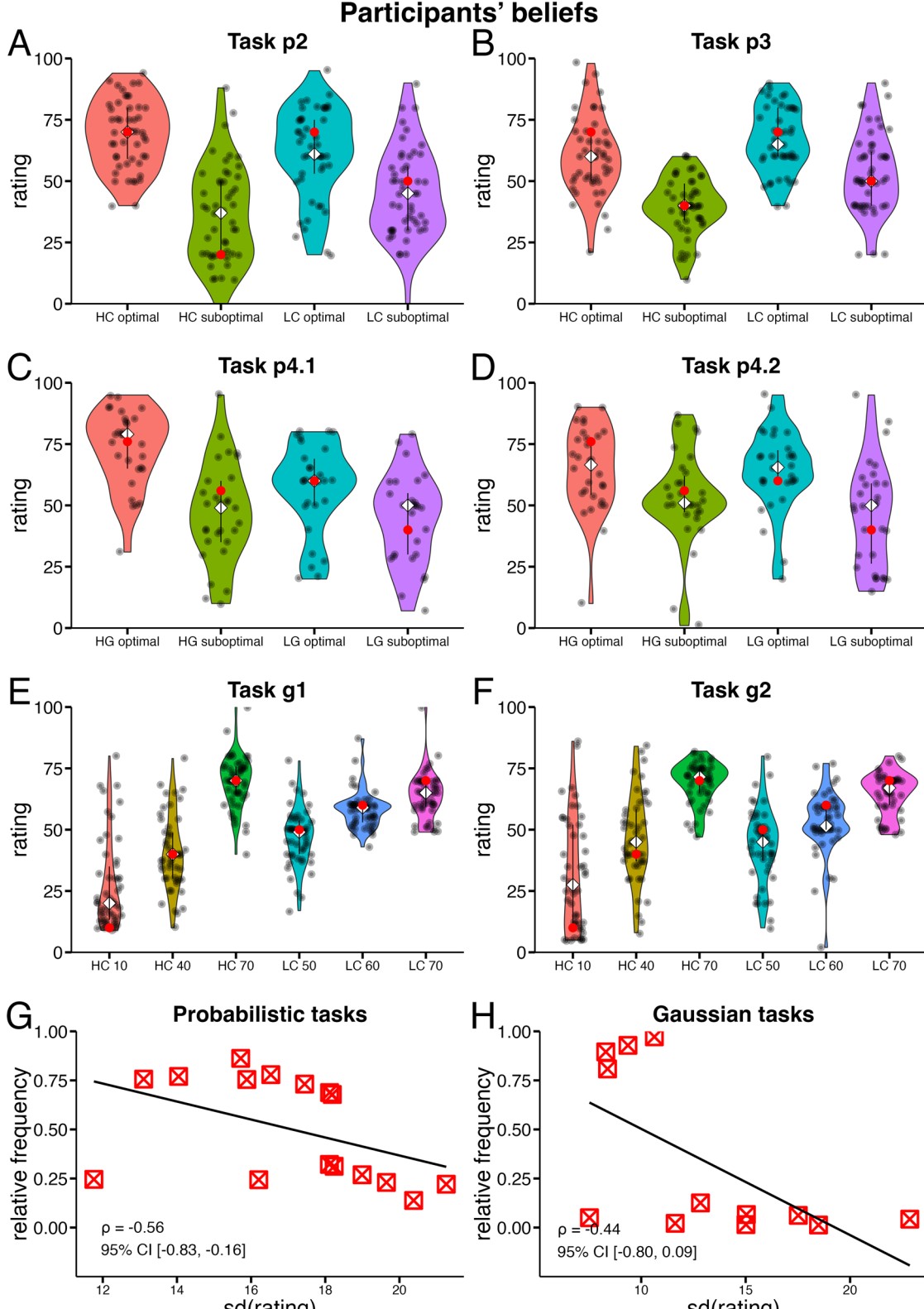

**Fig. 5 | Post-task valuation ratings and relationship between uncertainty and choice frequency (data was not collected for task p1).** A–F Violin plots for post-task valuation ratings (subjective beliefs) with true reward probabilities or reward magnitudes of each option (red dots), mean of participants' beliefs about these task parameters (white diamonds), and individual participants' ratings of each option (black dots). **A** Valuation ratings for task p2 ($n = 49$). **B** Valuation ratings for task p3 ($n = 49$), **C** Valuation ratings for task p4.1 ($n = 29$). **D** Valuation ratings for task p4.2 ($n = 30$). **E** Valuation ratings for task g1 ($n = 50$). **F** Valuation ratings for task g2 ($n = 49$). **G** Relationship between participants' (group level) uncertainty about an option and normalized choice frequency in the learning phase, for all probabilistic tasks except p1. The standard deviation (SD) of participants' individual valuation ratings as a measure of belief uncertainty is plotted on the $x$ axis. Normalized choice frequency of each option during the learning phase is plotted on the y axis. We found substantial evidence for negative correlation ($\rho = -0.56$, 95% CI: [$-0.83$, $-0.16$]; $BF_{10}$: 9). **H** Equivalent to (**G**) but for Gaussian tasks (g1 and g2) with anecdotal evidence for a negative correlation ($\rho = -0.44$, 95% CI: [$-0.80$, 0.09]; $BF_{10}$: 1.7).

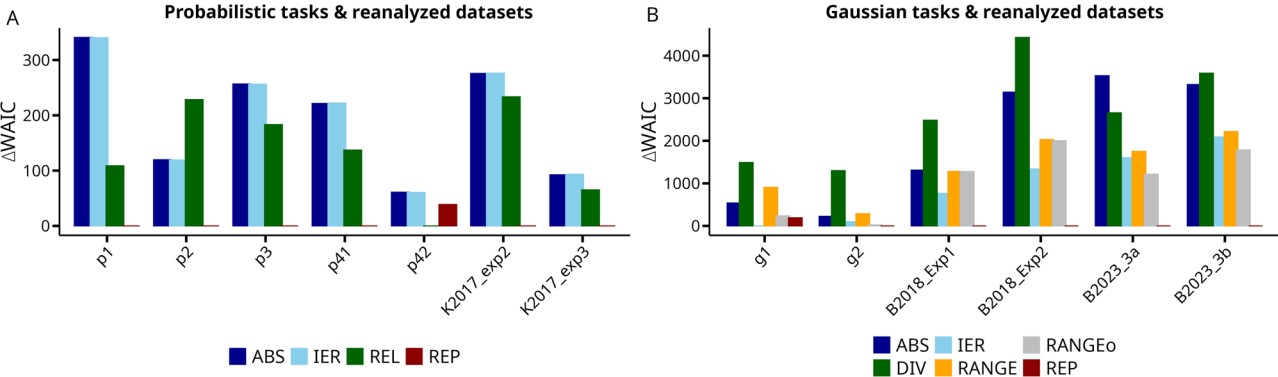

**Fig. 6 | Group level model comparisons for probabilistic and Gaussian tasks introduced in this study and the reanalyzed datasets from Klein et al.[15], Bavard et al.[16] and Bavard and Palminteri 2023. A** Model comparison for probabilistic tasks (p1, p2, p3, p4.1, p4.2) and Experiments 2 and 3 from Klein et al.[15] based on WAIC values, where a lower DIC indicates a better fit. The REP model (red) consistently demonstrates the best fit across all probabilistic tasks except for task 4.2, where it was the second-best model. Specifically, the REP model outperforms all other models in six out of seven tasks or eight out of nine when considering tasks p1.1–p1.3 separately (see Supplementary Table 3). In contrast, other models such as REL (green), IER (blue), and ABS (dark blue) generally perform worse, except for the

REL model in task p4.2. For individual-level results, please see Supplementary Figs. 2 and 3. **B** model comparison for Gaussian tasks (g1, g2) and four datasets from Bavard et al.[16,18] Here, the REP model (red) again is the best model in all cases except for task g1. Note that for tasks g1 and g2, all models perform approximately equally, likely due to the relative ease of these tasks, where even the ABS model shows nearly similar performance. Other models, including RANGE (yellow), DIV (green), and IER (blue), are generally outperformed by REP, especially in more complex tasks such as those in Bavard et al.[16,18], where participants encountered four different choice contexts during learning and 28 and 66 distinct comparisons during transfer.

and decreases when the option is not chosen (Eq. 20). Q-values and REPbiases were then added additively within the softmax function (Eq. 21). Using hierarchical Bayesian modeling and quantitative model comparison, we tested this model (called in the following the repetition (REP) model) against all recently proposed computational mechanisms to explain biased choice preferences in the transfer phase. From the literature, we identified six different models. These are relative value learning (REL model[15,16]), divisive normalization (DIV model[23,39]), variations of range normalization (RANGE & RANGEω models[18]), the intrinsic enhancement model (IER model[19]) proposed by Molinaro and Collins[19] and a fully unbiased base model where Q-values are learned on an absolute scale (ABS model). For datasets based on probabilistic tasks with constant reward magnitude (tasks p1 ($n = 94$), p2 ($n = 49$), p3 ($n = 49$), p4.1 ($n = 30$), p4.2 ($n = 29$) and two datasets from Klein and colleagues[15] we compared the REP, ABS, REL and IER models (subset 1; see methods on RL models for details on model selection). For Gaussian tasks and probabilistic tasks with a broader range of reward magnitudes (g1 ($n = 49$), g2 ($n = 48$), Exp 1 ($n = 20$) and Exp 2 ($n = 40$) from Bavard et al.[16] and Exp3a ($n = 100$) and 3b ($n = 100$) from Bavard and Palminteri[18]) we compared the REP, ABS, DIV, RANGE, RANGEω and IER models (subset 2). In the following, we focus on model comparisons using context-specific decaying learning rates. We do report group-level WAIC values corresponding to the hierarchical Bayesian model (group level) and % of participants best fit by each model to show heterogeneity across participants. Results for models with constant but separated learning rates for chosen and unchosen options were mostly similar and are reported in the Supplementary Information (see Supplementary Table 3). For group-level hyperparameter distributions, see Supplementary Table 5 and Supplementary Figs. 12–14.

**Probabilistic tasks (new datasets)**

First, we analyzed task p1 (full feedback) and task p2 (partial feedback), which show the well-established preference for an option of equal value (see Fig. 2A, B, H, I) when it has been learned in a high contrast (HC) context, compared to an alternative learned in an LC context[15,16]. As expected, the ABS model cannot qualitatively predict these findings. Theoretically, the REL model should predict this outcome because the best option in the HC context (0.5 value difference; see task parametrizations in Figs. 1 and 2) has a higher relative value (+0.25) compared to the best option in the LC context (0.2 value difference; +0.1 relative value). The IER model should explain this preference due to less counterfactual feedback, which should increase the

intrinsic enhanced signal for the best HC option[19]. The REP model should also account for this preference, as the best HC option was chosen more frequently, thereby increasing the REP bias for this option. For task p1 (pooled data for p1.1–p1.3; for separate model fits see Supplementary Table 3), we find that quantitative model comparison using the WAIC favored the REP model (WAIC: 4048.8, lower values indicate better fit; 43% of participants were best fit by this model; for a full overview see Supplementary Figs. 2 and 3) compared to the REL (WAIC: 4157.6, 44%), ABS (WAIC: 4389.7, 7%) and IER (WAIC: 4389.1, 6%) models. For task p2 (partial feedback) the results are qualitatively similar, i.e. participants showed a consistent preference for the best option learned in the HC context. Again, the REP model (WAIC: 2651.8; 39%) performed better than the ABS (WAIC: 2771.5; 18%), REL (WAIC: 2880.2; 27%) or IER (WAIC: 2771.0; 16%) models (see Fig. 6A).

Task p3 was explicitly designed to test whether a preference during transfer for one of the two highly rewarded options ($A_{LC}(0.7)$ or $C_{HC}(0.7)$) was driven either by reward centering (as in the REL model), by an intrinsic signal (as predicted by the IER model), or by choice frequency, i.e. repetition (as considered by the REP model). To test this, we increased the number of LC context repetitions relative to the number of HC context repetitions (see Fig. 2C; 50 times vs 30 times). In theory, both the REL and IER models will predict a preference for the best HC option ($C_{HC}$) because there was less counterfactual feedback, or it had a higher relative value. In contrast, the REP model predicts a preference for the best LC option because this option was chosen more often during the learning phase due to the increased number of LC trials. The behavior shown during the transfer phase supported the REP model's prediction as participants showed a strong preference for the best LC option (see Fig. 2H). Congruent with observed behavior during transfer, model comparison shows that the REP model had the lowest WAIC (3375.2; 43%), outperforming the IER (WAIC: 3631.4; 18%), REL (WAIC: 3558.4; 35%), and ABS (WAIC 3631.8; 7%) models (see Fig. 6A and Supplementary Table 3 and 6).

In tasks p4.1 and p4.2, both contexts in each task featured options with an equal expected value difference of 0.2, but they differed in their absolute expected values, resulting in a low gain (LG) and high gain (HG) context (LG context: $A_{LG}$ (0.60) vs. $B_{LG}$ (0.40) and HG context: $C_{HG}$ (0.76) vs. $D_{HG}$ (0.56); for task parametrizations and experimental results see Fig. 2D, E, K, L. Here, we focused on a potential choice preference between option $A_{LG}$ (0.60) and option $C_{HG}$ (0.76) during transfer. Further in task p4.1 we implemented that participants receive counterfactual feedback for the best

option in the HG context ($C_{HG}[0.76]$) during early trials; in p4.2 this counterfactual information was absent during the beginning of the task. Because counterfactual feedback slows accumulation of positive evidence, the best HG option ($C_{HG} = 0.76$) is repeated less often in p4.1 than in p4.2, thereby decreasing repetition of this option and allowing us to investigate its influence on transfer preferences. In theory, the REL model would predict indifference for either $A_{LG}$ or $C_{HG}$ because both options are associated with equal relative value. We found that the dataset for task p4.1 was again best explained by the REP model (WAIC: 1562.0; 57%); REL (WAIC: 1724.28; 27%); ABS (WAIC: 1783.4; 7%); IER (WAIC: 1784.1; 10%). Task data for p4.2 was best explained by the REL model (WAIC: 1316.3; 69%), with the REP model being second (WAIC: 1,355.1; 10%). The other two models had a higher WAIC: IER (WAIC: 1,376.8; 10%) and ABS (WAIC: 1377.2; 10%); see Fig. 6A) Note that simulations (see Supplementary Fig. 10 and Supplementary Fig. 11C) speak against the REL model, because participants showed a clear transfer preference and no evidence of indifference (see Fig. 2E). ABS and IER models do only differ minimally because counterfactual feedback (reward for the worse option while the best option was unrewarded) was very rare in all probabilistic tasks.

### Gaussian tasks (new data)
Next, we present the results of our Gaussian tasks g1 and g2, where we tested the REP model against several normalization models: divisive normalization (DIV), range normalization (RANGE), weighted range normalization (RANGEω), and the IER model. According to the theory underlying range normalization, participants should be indifferent between the $A_{LC}(70)$ and $D_{HC}(70)$ options because both options would be normalized equally when considering the context-specific ranges (LC context: A(70)-B(60)-C(50) and HC context: D(70)-E(40)-F(10)). Contrary to this prediction, participants showed a clear preference for the $D_{HC}(70)$ option over the $A_{LC}(70)$ option in both the full and partial feedback tasks (g1 and g2; see Fig. 2M, N). For task g1, the IER model provided the best fit (WAIC = 1948.3; 30%), followed by the REP model (WAIC = 2139.6; 18%) and the RANGEω model (WAIC = 2185.9; 18%). The other models performed worse, with WAIC values of 2487.6; 28% (ABS), 2854.7; 4% (RANGE) and 3438.6; 2% (DIV). For task g2, the REP model provided the best fit (WAIC = 4175.5; 31%), followed by the RANGEω model (WAIC = 4199.0; 35%) and the IER model (WAIC = 4274.9; 8%). The other models performed worse with WAIC values of 4400.8; 10% (ABS), 4463.2; 10% (RANGE) and 5474.8; 4% (DIV).

### Previously published datasets
To further assess the generalizability of the REP model, we conducted a comparative model analysis using data from previously published studies that introduced alternative models. First, Klein et al.[15], whose datasets are in principle equally parametrized as our tasks p1 and p2, found evidence in favor of the REL model (when compared to the ABS model). Our reanalysis, as in the analyses of the newly collected datasets p1 and p2, confirmed the original finding that the REL model (WAIC k_exp2: 1724.6; 17%; k_exp3: 1470.4; 30%) explained decision biases in the transfer phase better than the unbiased model (WAIC(ABS) exp2: 1766.8; 3% exp3: 1497.7; 7%). However, when including other models in the comparison, the REP model had the lowest WAIC (WAIC(REP) exp2: 1491.0; 73% exp3: 1405.2; 50%), outperforming both the REL and IER models (WAIC(IER) exp2: 1767.2; 7% exp3: 1498.7; 13%). The Bavard and colleagues[16] dataset allowed us to test the REP model in a task that included both gain and loss contexts, different magnitudes, and mixed partial and full reward feedback. The REP model performed better than the other models, with WAIC(REP) exp1: 4742.0; 90% and exp2: 9438.1; 90%, while the IER model ranked second with WAIC(IER) exp1: 5506.4; 10% and exp2: 10,777.5; 8%, see Fig. 6B and Supplementary Table 3 and 6 for a complete overview of model comparisons). Further, we analyzed two datasets from Bavard and Palminteri 2023[18], which use a Gaussian reward schedule comparable to our tasks g1 and g2, but with the added complexity of a larger context-space (four instead of two choice contexts during learning) and blocking the best option, i.e., making it unavailable, in two out of four choice contexts in a subset of trials.

The authors of the original paper concluded that participants employ scaled range normalization rather than divisive normalization. Our analysis shows that the REP model (WAIC(REP) Exp3a: 27,030.1; 55%; exp3b: 28,262.9; 57%) substantially outperformed the authors' best model (WAIC(RANGEω) exp3a: 28,244.7; 16%; exp3b: 30,049.1; 11%) as well as the recently proposed IER model (WAIC(IER) Exp3a: 28,633.4; 9%; exp3b: 30,353.5; 7%; (see Fig. 6A–D for an overview and Supplementary Figs. 2 and 3 for subject-level results and Supplementary Table 3 and Supplementary Table 6 for group level fits). Interestingly, in this dataset[18], the model fit of the REP model further improved substantially when the repetition bias was not applied to trials in which participants were constrained to select the second-best option due to the best option being blocked. Future research could further explore this effect.

In summary, our findings show that the proposed repetition-bias (REP) model is the best explanation across almost all analyzed datasets ($n = 701$ participants). Moreover, it is the only tested model that implements a mechanism for the correlation between choice frequency during learning and choice preference during transfer, observed across all datasets. Additionally, our results show that the relative choice frequency of an option is linked to both the perceived value of this option (post-task valuation, not $Q$ values after learning) and the certainty about that value. Participants showed greater certainty about options they had selected more frequently, even when full feedback was available and tended to overvalue these frequently chosen options in their post-task valuation compared to those selected less often. We further discuss the implications of our findings below.

### Parameter and model recovery
We performed full model confusion for a range of representative tasks that differ in their reward-generating process (probabilistic or Gaussian), model predictions and task complexity. For each task (p3, p4.1, g1, Exp2 from Bavard et al.[16]) and model (ABS, IER, REL, REP, DIV, RANGE, RANGEω; 20 combinations). Across 20 replicates for each analyzed task, the REP model yielded a 100% recovery rate. However, confusion was not perfect for all models across tasks (for full details please see the section on model recovery and full NxN confusion matrices in the Supplementary Information; Supplementary Figs. 6–9). We performed parameter recovery from the full posterior distribution for the REP model for tasks p3 and Exp2 from Bavard and colleagues[16]. Recovery for all parameters was moderate to good (see Supplementary Figs. 4 and 5).

### Discussion
We tested mechanisms driving biased choice preferences, which emerge when options learned in stable decision-making contexts are encountered in novel situations, here, mixed contexts. To date, it is unclear whether state-dependent normalization[17,18,23,39], or the consideration of internal reward signals[19] is sufficient to account for such biases. First, using standard analysis, across a range of value-based decision-making tasks (including gains, losses and different types of reward feedback) and 15 datasets, we found evidence that there is a link between repeating choices within context during learning and choice preferences across context in the transfer phase. Even when controlling for reward effects, the relative frequency with which participants chose an option during learning was linked to cross-context choice preferences, post-task valuations, and value uncertainty. Second, using hierarchical Bayesian modeling, we found that the proposed REP model (unbiased RL combined with a repetition bias) outperformed all alternative models in 12 out of 15 datasets and was the second-best model in the remaining three datasets. These results suggest that repetition is a key mechanism in preference formation.

The finding that making a choice increases the likelihood of repeating that choice in the future aligns well with both historical and contemporary theories of decision-making. Over a century ago, Thorndike's[26] *Law of Exercise* highlighted the importance of repeated actions in shaping behavior. Consistent with these findings, we observed that higher choice frequency during the learning phase is strongly correlated with biased preferences during the transfer phase, even when expected values were equal on both

absolute and relative scales (Fig. 3A, B). Notably, if state-dependent normalization were the primary process underlying such biases, we would not expect to observe such strong correlations. Strikingly, we found choice frequency to be related not only to preferences but also to explicit post-task valuations, as participants tended to assign higher values to more frequently chosen stimuli. This observation aligns well with Brehm´s[36] theory of post-decision changes in valuation. The free-choice paradigm[36,38,48] has been used extensively to investigate the phenomenon that chosen options are over-valued when contrasted to unchosen ones. Critics of this paradigm have argued that the preference for chosen options may be linked to certainty, i.e., participants might prefer a chosen option because they are less uncertain about its true value[49]. Our results support both perspectives: increased choice repetition was associated with greater certainty about the underlying value (Fig. 4G, H). Further, more frequently chosen options, while controlling for the effect of reward, were also consistently overvalued in post-task valuations when compared to those chosen less frequently (Fig. 3D; see ref. 38 for a recent meta-analysis on the free choice paradigm).

Our computational modeling approach pursued the three main goals of (i) replication, (ii) testing predictions, and (iii) reanalysis: First, with task variants p1 and p2 we replicated phenomena reported by Klein and colleagues[15] and compared models that can in principle explain a biased preference toward options learned in contexts with higher contrast (i.e., for options with context-specific larger relative or normalized value). Using model comparison, we found that the proposed REP model showed the lowest WAIC values across all alternative models. This finding of the relevance of a repetition bias is supported by previous related results: For example, studies have shown that incorporating choice history into RL models enhances the prediction of future preferences in both monkeys[27] and humans[12,28,34,35,50]. Our results further align with two recent preprints. Eckstein and colleagues[51] show that hybrid ANN-RL models can uncover both classical reward-based learning and additional reward-independent mechanisms (e.g., choice kernels). They find that action repetition effects captured by choice kernels improve model fits and conclude that non-value-based repetition biases must be modeled to better capture human choice tendencies. Likewise, Collins[52] shows that humans often rely on non-RL mechanisms, especially in structured environments and find that reward blind associative processes, i.e., repeating actions, are needed to better explain behavior. Relatedly, Klein et al.[15] observed trend-level evidence for a frequency effect in a subset of their data using logistic regression. In our re-analyses, this effect became reliable primarily when pooling across datasets. In another study, Bavard et al.[17] incorporated a habitual (choice trace) component inspired by Miller et al.[31] into one of their models. Although this model nearly matched the performance of their proposed range adaptation model, they found that a habitual component alone was insufficient to account for the large variations in option values in their dataset. One practical consideration is scaling: in their implementation, $Q$ values spanned roughly 0–7.5, whereas the choice-trace term was bounded in [0,1]. Crucially, their approach, while theoretically related to ours, also differs in implementation. In our understanding, they only update the habitual strength of the chosen option[31]. In contrast, in the model proposed here, the repetition biases for both chosen and unchosen options within a context can increase or decrease as previously proposed[30,33]. The usefulness of this updating rule for the unchosen option hints at context-specific, inter-connected updating rules of repetition biases, similar to mechanisms observed in reward-driven valuation[53].

Second, to disambiguate between models, we designed several tasks to explicitly test unique predictions of the REP model. In task p3 the context with a lower value difference (LC context) was repeated more frequently than the context with a higher value difference (HC context; Fig. 2C). For this task, state-dependent normalization[15] and the IER model[19] would both predict a preference for the best HC option due to its higher relative value or reduced counterfactual feedback (note the latter depends on the exact task dynamics). Contrary to these predictions, participants exhibited a strong preference for the best LC option ($A_{LC} = 0.7$), which we found is only explained by the REP model. In tasks p4.1 and p4.2, we designed the relative

values (contrast of expected values) within each context to be equal (see Fig. 1). Under these conditions, normalization models would predict indifference between options, while the REP model would explain choices by the task-dependent combination of unbiased learning and a repetition bias. In task p4.1, the REP model was favored by quantitative model comparison. This aligns with the finding of our standard analysis that a rather equal choice frequency during learning leads to indifference during transfer (see Figs. 2D, K and 3A). Surprisingly, task p4.2 was best explained by the REL model utilizing decaying learning rates, while the REP model was the second-best model. This was despite the REL model's theoretical prediction of indifference, which was clearly not the case for participants' behavior (see Fig. 2L). These results can be explained by two factors. The first factor is that the task was relatively easy to learn (see Fig. 2L and methods for details). Therefore, in principle, model comparison will favor models with less complexity. We note that the REL model is the simplest of the models compared, i.e., it has the fewest parameters, relying only on the learning mechanism and a context-dependent difference value ($Q_{difference}$). The second factor is the REL model's parameter estimates (variant with learning decay). We found that the RL model inferred larger Q-value differences between options $C_{HG}$ (0.76) and $D_{HG}$ (0.56) compared to options $A_{LG}$ (0.60) and $B_{LG}$ (0.40). Consequently, the REL model accounts for the observed preference for option $C_{HG}$ over option $A_{LG}$ during the transfer phase, despite its theoretical prediction of indifference. This is further supported by a much higher WAIC value of the REL model when employing a constant learning rate (see Supplementary Table 3). These counterintuitive but explainable findings highlight the importance of evaluating models across different learning mechanisms and a diverse range of tasks via simulations (see Supplementary Figs. 10 and 11).

Third, we reanalyzed more complex existing datasets featuring a higher number of choice contexts, specifically four different contexts (with eight and twelve different options during learning and 28 and 66 transfer comparisons in contrast to two (probabilistic tasks) or nine (Gaussian tasks) transfer comparisons in our datasets). In these tasks, participants encountered scenarios with (i) varying probabilistic gains and losses[16] and (ii) Gaussian tasks with higher complexity, i.e., blocked best options in a subset of trials[18]. In both of these datasets, the REP model performed substantially better than all alternative models (expressed as larger absolute and relative WAIC differences when compared to model comparisons on other datasets, see Fig. 6B and Supplementary Table 3 and 6). In the study of Bavard and colleagues 2018[16], the mechanism of the REP model explains the findings of the standard analysis that losses (small or big) or relative losses (small gains) are preferred to larger gains when the options associated with these lower values or losses were selected more frequently during learning (see Fig. 3C). This demonstrates a relationship between loss preference and relative choice frequency, which is not implemented in any other model. Similarly, we find by modeling both datasets from Bavard and Palminteri[18] that middle-valued options that had been repeated more often, due to the blocking of the best option, were preferred more frequently during the transfer phase over their equally valued counterparts (see upper right green rectangle in Fig. 3A).

Given these findings, the question arises: why does repetition play such a prominent role in decision-making at all? One explanation is that humans and other animals tend to repeat choices independently of reward to minimize policy complexity and therefore reduce cognitive costs[1,32]. A recent study demonstrated that policy compression, i.e., repetition of choices, increases when memory load is high[35]. Similarly, we find that implementing a repetition bias appears to have larger effects on model performance in more complex tasks, i.e., in tasks with more contexts, more transfer comparisons, and more variations in reward magnitude than in simpler tasks where cognitive constraints likely play a lesser role (e.g., g1 and g2; see Fig. 5B). This may not only explain why the REP model was not the best model in tasks g1 but also suggests that the brain may increasingly rely on specific strategies to manage the heightened cognitive load associated with greater task complexity. One such strategy may be reducing complexity by chunking information into distinct contexts[2]. By organizing information into contexts, contextual cues in principle enable the brain to rapidly infer

the relevant (or more certain) context and activate context-relevant memories or action plans[11,13,14]. Said differently, when options learned in different contexts are recombined, participants likely infer the latent context associated with each option and retrieve the corresponding action policy. Because repetition during learning increases certainty for that option's context-specific policy, the option with the stronger repetition history enjoys a cross-context advantage at transfer.

We therefore speculate that contextual inference may be the basis for a context-specific repetition bias, as recently shown in a modeling study[10] and may result in enhanced efficiency of retrieving relevant actions, i.e., effectively speeding up information processing within familiar contexts or when more certain contextual cues are present[12,29,54–56].

One important advantage of the REP model is that it performs consistently across a range of diverse task parametrizations with varying reward structures, number of trials per context, and independent of whether partial or full feedback is provided. Crucially, in the REP model, action selection is based solely on variables that are always accessible to the agent, as value-free repetition explicitly does not depend on the outcomes of other options. In contrast, the IER[19] and normalization models DIV or RANGE[17,18,39,42], are at a potential disadvantage here as they work best with full knowledge of the outcome structure (or unconstrained memory systems), which may or may not be available to participants in both experimental tasks and, importantly, in real-world interactions.

Note, we do not state that mechanisms like normalization or intrinsic goal signals are absent in value-based decision-making. Our intention was to explain choice preference by context-dependent repetition, a factor not implemented in most alternative models of context-dependent decision biases. However, it is well possible that multiple mechanisms interact. For example, the IER model proposed by Molinaro and Colins[19] effectively explains switching or exploratory behavior following counterfactual rewards during learning. This is enabled by its explicit mechanism that generates signals based on potentially available counterfactual or non-optimal feedback. Furthermore, other mechanisms are known to influence decision-making under risk[57] or when alternative options interact with goals, for instance, anchoring effects in purchasing decisions[58–62]. Future research should examine in more detail if and how repetition and other mechanisms interact, i.e., in multi-attribute decisions in more complex scenarios.

## Limitations

Several limitations need to be addressed: First, all tasks were conducted in stationary environments with stable reward contingencies. Repetition effects may differ under conditions of environmental volatility, i.e., one could imagine that repetition may exert weaker effects and repeating actions is less pronounced as environmental volatility increases. Moreover, although our results replicate across datasets, all tasks share a relatively similar structure with binary or trinary choices and fixed learning phases. Future studies should examine how such biases interact with changing reward structures and more complex environments (but see, ref. 34). Importantly, some analyses, particularly cross-task correlations for frequency preference relationships, are based on a small number of datapoints (i.e., pairwise decisions were limited per task) and were therefore computed on the group level. While we supplemented these analyses with hierarchical mixed-effects regressions that incorporate participant-level variability and replicate the key effects, these models still rely partly on data pooled across tasks, which may limit generalizability at the individual level. Further, in this analysis, we isolate the statistical relationship between frequency and preference and do not control for other effects that might cause distortions in value representations and preference (but see additional analyses where we control for estimated Q-values in the Supplement). Finally, while our model recovery analyses confirm that key models such as REP and REL, but also IER, DIV and RANGEo are distinguishable in key tasks, recoverability was lower for certain task-model pairs. This highlights an important challenge: model discriminability is strongly shaped by task design (i.e., reward

magnitudes, the amount of learning and transfer trials or cross-context comparisons). Because our tasks were largely inherited from prior studies and included reanalysis of existing datasets, optimizing for model separability was not prioritized. Future work would benefit from explicitly testing model identifiability during task development to ensure that competing accounts yield sufficiently divergent predictions within the chosen design.

## Conclusion

While there is overwhelming consensus on the role of reward in shaping preference and guiding action, i.e., the *Law of Effect*[26], expected utility theory[63], operant conditioning[64], prospect theory[65], incentive salience[66], the role of repetition in shaping choice preference is less well-established. However, several findings across various domains, i.e., in experimental research on value-based choice[12,27,28,33–35,50], perceptual decision-making[29,55], social psychology[54,56], recent modeling studies on the interaction between habitual and goal-directed behavior[10,31], cognitive control[11] and real-world preferences[67] point to a fundamental role of the *Law of Exercise*[26] in shaping human choice. The present study adds to these findings by demonstrating a clear role of choice repetition in shaping decision biases previously attributed to value normalization.

## Data availability

All newly collected datasets are available on OSF https://osf.io/zj95m/.

## Code availability

All model code used to fit the RL models is available on OSF https://osf.io/zj95m/.

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

## Acknowledgements

We thank Yaning Hu for assistance with task programming and Dario Cuevas-Riva for helpful discussions and advice. Research was funded by the German Research Foundation (DFG, Deutsche Forschungsgemeinschaft), SFB 940 - Project number 178833530 and as part of Germany's Excellence Strategy—EXC 2050/1 - Project number 390696704 - Cluster of Excellence "Center for Tactile Internet with Human-in-the-Loop" (CeTI) TU Dresden. B.J.W.'s position at University Hospital Tübingen is founded by an Alexander von Humboldt Professorship awarded to Peter Dayan. The funders had no role in study design, data collection and analysis, decision to publish or preparation of the manuscript.

## Author contributions

B.J.W. and S.J.K. conceived the idea. B.J.W. designed the study. B.J.W. and H.B.W. acquired the data. B.J.W. contributed analytical tools. B.J.W. and H.B.W. analyzed the data. B.J.W. wrote the paper. S.J.K. provided funding and supervised the project. All authors read and edited versions of the manuscript and approved the final version.

## Funding

## Competing interests

The authors declare no competing interests.
