## [Transparent Peer Review file · Communications Psychology]

Action repetition biases choice in context-dependent decision-making

Corresponding Author: Dr Ben Wagner

Version 0:

Decision Letter:

Dear Dr Wagner,

Thank you for your patience during the peer-review process. Your manuscript titled "Explaining decision biases through context-dependent repetition" has now been seen by 3 reviewers, and I include their comments at the end of this message. They find your work of interest but raised some important points. We are interested in the possibility of publishing your study in Communications Psychology, but would like to consider your responses to these concerns and assess a revised manuscript before we make a final decision on publication.

We therefore invite you to revise and resubmit your manuscript, along with a point-by-point response to the reviewers. Please highlight all changes in the manuscript text file.

Editorially, we consider it essential that the revised manuscript directly addresses the concerns about the interpretation of repetition effects, including clarifying whether these effects could be driven by changes in experienced value. Multiple reviewers noted the absence of individual-level modeling results and emphasized that group-level fits alone are insufficient. All reviewers requested methodological clarifications, additional model diagnostics (including posterior predictive checks, and parameter and model recovery analyses) and more transparency about the experimental design and analysis procedures. While some statistical issues may require substantial reanalysis, we believe they are addressable and that doing so will ultimately strengthen the manuscript.

I am attaching an Editorial Requests Table that details critical reporting requirements for the revised manuscript. Please attend to each item and ensure your manuscript is fully compliant. If your revised manuscript is not aligned with these requests on major issues, such as those concerning statistics, it may be returned to you for further revisions without re-review.

Please submit the following items:

- Revised manuscript
- Point-by-point response to the referees' comments
- Cover letter (as a separate document)
- <https://www.nature.com/documents/nr-reporting-summary.zip>>Nature Research Reporting Summary
- <https://www.nature.com/documents/nr-editorial-policy-checklist.pdf>>Editorial Policy Checklist
- Completed Editorial Request Table (attached).

via this link: Link Redacted .

Additional guidance is available in our style and formatting guide Communications Psychology formatting guide.

Best regards,

Jesse Rissman

Jesse Rissman, PhD
Editorial Board Member
Communications Psychology
orcid.org/0000-0001-8889-5539

REVIEWER EXPERTISE:

Reviewer #1: computational modeling of human cognition
Reviewer #2: computational modeling of human cognition
Reviewer #3: computational modeling of human cognition

REVIEWER REPORTS:

Reviewer #1 (Remarks to the Author):

In this paper, the authors investigate the role of context-dependent choice repetition in decision making, in contrast to value-based approaches. To do so, they reanalyze 6 published data-sets of reward-based learning and 2 novel ones (with multiple minor variants). They report behavioral analyses showing a role for repetition in choices when controlling for objective absolute and relative value; and they report quantitative model comparisons supporting a model with absolute value + repetition effects over published competing models.

I think the conclusions are probably accurate (partly because at least two other preprints draw the same conclusions in different settings with different analyses [1,2]), and I think this study is worthwhile and timely. Unfortunately, I do not think the conclusions follow from the analyses presented here, as I see serious methodological/statistical/modeling issues in this version of the paper. In addition to methodological issues, another major issue is with the interpretation of the findings. While the paper is overall well written, there are a few places where information is hard to find, and clarification would be helpful.

Major issues.

1. Interpretation. To me the most substantive issue in the paper is in the interpretation of the role of repetitions. There are at least two ways in which repetitions could impact choice: directly [the proposed model and implicit interpretation], and crucially, indirectly [by impacting the estimation of the experienced value]. This is not discussed or controlled for statistically anywhere in this article.
2. For example, P7: the authors claim that they control for value by selecting "equally valued" pairs. However, the equal value is assumed by design (i.e. the experimenter designed this as a $p=.7$ pair), but not guaranteed in terms of the participant's experience, given the probabilistic nature of feedback (and despite the good randomization controls reported in methods) and given participants' probabilistic action selection. A participant who chose the high value option once, got unlucky, received 0, and never selected that option again, has a subjective estimate closer to 0 than to .7. Thus, the analysis does not actually control for (experienced) value (or relative values). [3] demonstrates how taking this experienced value into account (rather than objective) can change interpretation in probabilistic tasks. It is crucial that the statistical analyses be deepened to actually control for experienced value in parallel to repetitions (e.g. in multiple regression frameworks).

3. The same statistical methodological issues confounds the correlation analyses - frequency correlation between learning and transfer needs to be corrected for relative experience outcomes.
4. Experiment 4.1 reports a null result that is not very informative in the absence of either formal null hypothesis testing, or interaction analysis with exp 4.2. Note that the 0.16 probability difference between the transfer options is extremely small and would not be expected to be strongly noticeable, in particular with the low N=30 used.
5. The statistical analyses appear to consistently ignore random effects, treating the whole group as a single participant, which is highly unusual and likely to artificially inflate statistical outcomes. This is true in the fig. 2 analyses, as well as in the correlation analyses (see fig. 3) where the correlation is measured not across participants of a task, but across conditions and tasks, averaged over participants. This raises the concern that results could be driven by a few unlucky participants learning the wrong associations in LC conditions, selecting it more often, and having incorrect estimates for the other option, a completely different interpretation.
6. Modeling results rely completely on quantitative model fit comparisons at the group level (DIC); however, this is known to often be inadequate [4] for testing competing theories. Instead, differential predictions should be qualitatively shown and compared [5]. The methods mention posterior predictive checks, but corresponding results are not reported (that I could find).
7. P15 – competing model predictions are derived in theory by reasoning through the model logic, but not in practice with simulations. I do not think that the authors' reasoning is always valid or parameter independent; for example for the reasons highlighted in point 2 (i.e. hysteresis effects of choices leading to different sampling of the environment leading to differences in how much has been learned for the values). This is a further argument for the need for simulations with fit parameters and qualitative comparison of model predictions.

Minor

- Please report other parameter fits (not just repetition learning rate).
- The difference between 4.1. and 4.2 was difficult to figure out, but seems an important piece of information.
- More generally, some information from the methods could productively be introduced earlier in the results to help with results interpretation.
- Modeling results should report on participant heterogeneity, as should behavioral analyses, rather than being only at the group level.

[1] Eckstein, M.K., Summerfield, C., Daw, N.D. and Miller, K.J., 2023. Predictive and interpretable: Combining artificial neural networks and classic cognitive models to understand human learning and decision making. *BioRxiv*, pp.2023-05.

[2] Collins, Anne GE. "RL or not RL? Parsing the processes that support human reward-based learning." *PsyArXiv Preprints* (2024).

[3] Schutte, Iris, et al. "Stimulus discriminability may bias value-based probabilistic learning." *PLoS one* 12.5 (2017): e0176205.

[4] Palminteri, Stefano, Valentin Wyart, and Etienne Koechlin. "The importance of falsification in computational cognitive modeling." *Trends in cognitive sciences* 21.6 (2017): 425-433.

[5] Wilson, R.C. and Collins, A.G., 2019. Ten simple rules for the computational modeling of behavioral data. *Elife*, 8, p.e49547.

Reviewer #2 (Remarks to the Author):

Wagner, Wolf, and Kiebel report a study on context-dependent repetition effects including computational modeling of behavioral data of 351 participants spread over nine studies using one version of an experimental paradigm each. They examine in how far simple repetition of choices contributes to the explanation of instrumental behavior in these binary choice tasks. They further use six previously published datasets to examine the explanatory power of their computational models for these data. They find evidence for an effect of choice repetition in almost all datasets and discuss their findings in the light of choice preference, instrumental behavior, and habit learning.

This manuscript is timely and relevant, well written, clear, and structured. The research questions and hypotheses are well informed by the literature, and the methods are sound and well-chosen to answer the question at hand. Moreover, openly sharing the data and analysis code (on publication) is commendable (though making these already accessible for us reviewers would have been even better). I have read the paper with great enthusiasm. However, I would like to raise a few issues I would like the authors to address.

1. How were the sample sizes of the studies of the new dataset determined? Did you make a priori analyses of statistical power or were other variables determining sample size? Please include a sample size justification (on the level of analysis, i.e., for each task and not over the total number of participants) in the Methods section.
2. While reading the Results section, it was often unclear to me on how many participants/data points the analyses were based on. Please report the degrees of freedom wherever possible to help with this issue. In addition, I think Supplemental Table 4 should be included in the Methods section instead of the Supplement to make information about the number of participants and trials for each data set more easily accessible.
3. The number of training and transfer trials in the new data sets was relatively low. What was the rationale for choosing this number of training and transfer trials?
4. Some of the correlations reported in the manuscript (e.g., displayed in Figures 3B, 4G and H) are based on rather few datapoints and possibly underpowered (see Schönbrodt & Perugini, 2013; Kretzschmar & Gignac, 2019). The high uncertainty around correlation coefficients based on few data points should at least be made explicit and their limits in robustness and interpretability discussed. Better yet if all underpowered results were to be removed from the manuscript.

5. Why was the assessment of preferences in the transfer phase based only on the first transfer trial? Using only the first transfer trials instead of aggregating over all relevant transfer trials decreases the reliability of this measure and might induce spurious results. Please justify the use of this statistical procedure.
6. Were there any model diagnostics performed? Please report the results of parameter recovery and model recovery analyses for all the computational models used in the manuscript.
7. The variance of the repetition hyperparameter is relatively large across tasks (Supplemental Figure 1 and Supplemental Table 6). Are there task design features correlating with the parameter distribution across tasks? Can you derive any tentative insight into which task features might promote or weaken the influence of behavioral repetition from this?
8. Figure 2 is a very good representation of the task set-up and some important descriptive statistics. Yet I think the text of the legends, captions, and axis titles is too small and the figure caption is in part unnecessarily repetitive.

References

- Kretzschmar, A., & Gignac, G. E. (2019). At what sample size do latent variable correlations stabilize? *Journal of Research in Personality*, 80, 17–22. <https://doi.org/10.1016/j.jrp.2019.03.007>
- Schönbrodt, F. D., & Perugini, M. (2013). At what sample size do correlations stabilize? *Journal of Research in Personality*, 47(5), 609–612. <https://doi.org/10.1016/j.jrp.2013.05.009>

Reviewer #3 (Remarks to the Author):

Summary

The paper examines the role of simple repetition of choices in decision biases, suggesting that decision makers may use their own choice history to simplify comparisons by focusing on previously selected options. Using data from both a new set of experiments and previous studies, the authors test correlations between choice frequency and later preference, finding stronger preference for options which were selected more often even when expected values were matched. This is further supported using a model comparison in which a reinforcement learning model incorporating a repetition bias generally outperforms existing competitors using reward normalisation methods. Based on these results, the authors argue for inclusion of a value-free repetition bias in theories of decision making.

I found this to be a relatively strong paper: the authors propose a simple method to account for observed biases in contextual learning, and perform a set of experiments to test this suggestion from multiple angles, as well as reanalysing existing data to check their results are not restricted to their own designs. Their findings are also further supported by a model comparison offering a theoretical grounding for their conclusions. I do however have some methodological confusions which I think need to be clarified (particularly on the design of the transfer trials), as well as some questions on the empirical analyses and modelling, though these can mostly be solved with edits to the text. I would therefore recommend this paper for publication following minor revisions. Below I offer some more specific comments that I feel should be answered. Thank you for the opportunity to read your work.

Major Comments

- The set-up of the transfer trials is not made sufficiently clear in the text in my opinion: there is no explicit statement on which comparisons were included, or how often each comparison was given. There is some indication of this in Figures 1 and 2, as well as the supplemental materials, but even then some aspects are left unstated or even seemingly contradictory: for example, Figure 1 highlights 3 transfer comparisons in the Gaussian conditions, but Figure 2 only shows 2 (according to the black arrows), while Supplemental Table 4 mentions 9 – so which is accurate? I also found it unhelpful to refer to transfer trials as 'per context' when from my reading these are specifically cross-context. I suggest the authors add more detail on this key aspect of their design, and address any inconsistencies.

- Following on from the previous point, I was confused about the use of feedback in the transfer trials: first, while it is explicitly stated that no feedback was given for the Gaussian tasks, the only statement on feedback for the probabilistic tasks I could find is: 'For the first transfer trial reward feedback was equal for the best HC and LC options' (page 27) – so feedback was given, but what was the procedure? Were rewards for the remaining trials pseudo-randomised as in the training phase, or fully random? And was this full feedback, even when training used partial feedback? More pressingly, why give feedback in this phase at all when this will influence subsequent choices? I'm assuming this is the reason for the restriction to only the first two transfer trials in most of the analyses (though this is not explicitly stated), but then why include feedback if it means dropping most transfer trials from consideration? Or was this restriction a post-hoc decision? There is no mention of a pre-registration for the studies so I cannot verify this, but I wonder if this issue was discovered during analysis and subsequently addressed in the Gaussian tasks. This does not invalidate the reported findings, but it does seemingly force a focus on group-level statistics in the analyses, reducing the statistical power of the study. The authors should acknowledge this limitation of their design, or provide justification for this choice if available.

- In the model comparison, were the normalisation models adjusting by all rewards seen in that context across trials, or specifically the rewards seen within that trial? Given that these models weren't applied to the probabilistic tasks, I'm assuming the former (as normalisation of binary rewards makes little difference), but I think this might differ for Divisive Normalisation if operating at the trial level: if both options are rewarded in a trial, then their individual relative rewards could be reduced (i.e., 0.5 rather than 1), leading to a lower overall subjective value – and since this joint outcome is more common in the low contrast context, it will affect option A more than option C, potentially predicting the observed preferences. This may not be in line with previous uses of this model, of course, and may not account for all observed effects, but seems worth considering. In any case, I would appreciate clarification on the normalisation procedure.

- I appreciate the use of hierarchical Bayesian modelling, but I am somewhat surprised that the paper only reports group-level fits – surely a benefit of this method is to produce individual-level results? For example, does the performance of the REP model hold across all participants, or are there subsets using alternate methods?

- While the REP model does perform best in most of the tasks, it is interesting that in some cases other models offer better fits despite being theoretically unable to predict the observed empirical effects. The authors do provide some explanation for this in the Discussion, but I'm left wondering how often the models did actually follow their theoretical predictions: it seems at least the REL model deviated from this in task p4.2, but was this a common finding? It would be helpful to explore the behaviour of the models in more detail to verify they are acting as expected (even if only in the supplemental), for example by illustrating their transfer preferences in a similar manner to Figure 2. More broadly, if theoretical predictions can indeed be violated by differences in learning mechanism or task design, the authors should be more cautious in falling back on these theoretical predictions when interpreting their analyses (e.g., 'Note that standard analyses speak against the REL model', page 16), and in fact should place more weight on simulation results as concrete demonstrations of model behaviour.

Minor Comments

- A minor question on the order of the training trials: the text details the set-up within a context, but how were the two contexts arranged? Were these blocked or interleaved? If blocked, was there a separation between the contexts like that between training and transfer?

- Figure 1 suggests the Gaussian tasks included a visual illustration of context as coloured backgrounds, but I see no mention of this in the text – is this correct? And if so, why was this added for these tasks if this wasn't used in the probabilistic conditions? Is there something about the Gaussian tasks that necessitated this, or was this simply a progression of the design?

- The assumption that options in the p4 conditions have equal relative value is dependent on the use of range normalisation – if normalising by their sum, this equality no longer holds.

- I had some confusion with the relative choice frequencies used in the correlational analyses: first, following on from my previous comment on the set-up of the transfer phase, I'm unclear exactly which options and comparisons this was calculated for. This particularly applies to the correlation between frequencies and valuations which is stated to use normalised scores 'relative to all other options encountered in the task' (page 29) – so is this across both training and transfer trials? Second, the switching between individual and group-level scores could be better explained: I'm assuming these were calculated at the group level when assessing choices in the transfer phase given the low trial count per participant (i.e., only the first 2), but if not, please clarify the rationale for this.

- The results of the binomial tests are difficult to pull from the text – I would advise putting these in a table for readability.

- There is little to no discussion on comparisons of the worse options from each context (i.e., B vs. D), despite being shown for all conditions in the bottom row of Figure 2 – I realise comparisons of the better options are the key contrast, but is there anything to draw from these? If not, the authors may wish to remove these since this is already a crowded figure.

- I would appreciate some clarification on the number of data points going in to each correlation between training and transfer choice rates: since these measures are stated to be calculated at the group level, I assume this uses 2 points per task (or 1 when restricted to equal expected values), hence the need to pool data across tasks – is this accurate? If so, this presents another instance where restrictions on the number of training trials able to be considered cut into the statistical power of the study, preventing within-task correlations and so any comparisons of these correlations according to factors like partial feedback.

- Why is Task p1 not included in Figure 4? Did this not request valuations? I see no mention of this in the Methods.

- When introducing the REP model, it is initially unclear how the RL and repetition bias components are combined until the 'Action Selection' section, where they are shown to be additive – I would suggest clarifying this earlier, and removing Equation 20.

- The models are stated to deal with partial feedback by retrieving the most recent observation for an option when actual outcomes are unknown, but what about the first trial? Is there some prior value here?

- Since the key measure for model fit is difference in DIC, I would suggest adjusting the plots in Figure 5 to show this on the y-axes, or at least use separate axes between tasks so that smaller differences are not obscured by being plotted next to larger absolute scores (as in the g1 panel).

Typos

Page 2, Significance statement: 'repeating decisions can shapes distort'

Page 3, paragraph 1: 'first phase of such experiment's participants learn...'

Page 17, Figure 5 caption: 'Vertical red lines' should be horizontal

Communications Psychology is committed to improving transparency in authorship. As part of our efforts in this direction, we are now requesting that all authors identified as 'corresponding author' create and link their Open Researcher and Contributor Identifier (ORCID) with their account on the Manuscript Tracking System prior to acceptance. ORCID helps the scientific community achieve unambiguous attribution of all scholarly contributions. You can create and link your ORCID from the home page of the Manuscript Tracking System by clicking on 'Modify my Springer Nature account' and following the instructions in the link below. Please also inform all co-authors that they can add their ORCIDs to their accounts and that they must do so prior to acceptance.
<https://www.springernature.com/gp/researchers/orcid/orcid-for-nature-research>

Version 1:

Decision Letter:

Dear Dr Wagner,

Thank you for your patience during the peer-review process. Your manuscript titled "Explaining decision biases through context-dependent repetition" has now been reconsidered by two of the original reviewers, and I include their comments at the end of this message. They continue to find your work of interest but were not fully satisfied by your revisions and raised some important points. We are interested in the possibility of publishing your study in Communications Psychology, but would like to consider your responses to these lingering concerns and assess a revised manuscript before we make a final decision on publication.

We therefore invite you to again revise and resubmit your manuscript, along with a point-by-point response to the reviewers. Please highlight all changes in the manuscript text file.

Editorially, we consider it essential that you more compellingly address the possibility that the observed repetition effect may be confounded by experienced value. The assumption that full-feedback ensures equivalence between expected and experienced value is not sufficiently supported and should be explicitly tested and controlled for in your analyses. In addition, we encourage a clearer and more complete model recovery procedure, along with improved reporting of correlation statistics and greater integration of individual-level modeling results.

I am attaching an Editorial Requests Table that details critical reporting requirements for the revised manuscript. Please attend to each item and ensure your manuscript is fully compliant. If your revised manuscript is not aligned with these requests on major issues, such as those concerning statistics, it may be returned to you for further revisions without re-review.

Please submit the following items:

- Revised manuscript
- Point-by-point response to the referees' comments
- Cover letter (as a separate document)
- <https://www.nature.com/documents/nr-reporting-summary.pdf> Nature Research Reporting Summary
- Completed Editorial Request Table (attached).

via this link: Link Redacted .

Additional guidance is available in our style and formatting guide Communications Psychology formatting guide.

Best regards,

Jesse Rissman

Jesse Rissman, PhD
Editorial Board Member
Communications Psychology
orcid.org/0000-0001-8889-5539

REVIEWER REPORTS:

Reviewer #1 (Remarks to the Author):

I thank the authors for clarifying multiple points raised in my previous review (reviewer #1). Multiple issues remain not adequately addressed for publication, and should be revised.

Major issues

1. The answer to my previous major point 1 and point 2, regarding experienced value being a confound for repetitions, is not satisfactory. The authors' answer (and corresponding edits) hinge on the "full feedback" condition, arguing that this ensures that the experimenter defined expected value is the true experienced value. However, a large literature shows that participants do not integrate feedback from unselected options in the same way that they do for selected options (see e.g. Cockburn et al 2016, but also all the models from the papers cited that include different chosen and unchosen learning rates). As such, while full feedback helps to some degree, it simply does not guarantee that the experienced value is the true expected value. This is an assumption that the authors are making, and it should be a) made explicit; b) tested (which can easily be done via within participant, single trial logistic regression analyses). As it is likely not fully valid, it should then be controlled for. I appreciate the new mixed effect modeling which shows consistent but weaker findings, as expected; however this model should also include relevant covariates such as experienced absolute value or relative value, rather than limiting to specific subsets of the data. Should the repetition effect survive such corrections, then the statistical conclusions will be valid.
2. All three reviewers pointed in some way to the fact that the number of points entering in the correlations was unclear/low (my previous point 5), but the authors have not corrected the paper accordingly. The reviewers' confusion is largely because it is so unusual to do correlations across experiments, as is the core here in figure 3 and 5GH. Again, this would be much better tested within participants or at least hierarchically, using multiple regression approaches, better controlling for other covariates that might influence the findings. At the very minimum, the main text should make it abundantly clear how underpowered those are. For example, the result of Fig. 5H: $\rho = -.82$ reads impressive to an inattentive reader in the context of "15 tasks and 300's participants", but $\rho(11) = -.82$ puts this better in context by defining the degrees of freedom. Please include explicitly all degrees of freedom in the correlation stats if those correlations remain in the revised version.

Minor and Typos:

- P3: participant's  participants'
- P14: $r = -.43$  .43
- I remain curious how the findings relate to similar recent findings (previous citations[1,2]).

Reviewer #3 (Remarks to the Author):

Summary

The authors have done well in answering my comments from the previous draft, adding greater elaboration on their methodology and analyses, as well as additional tests seeking to capture participant-level effects in both the empirical data and the modelling. While some of my previous criticisms on the design of these experiments (specifically the use of feedback in transfer trials) still stand, I understand now that these were inherited from previous work and retained to support aggregation across tasks. This being said, I do still have some issues with the manuscript, particularly with the new model recovery exercise, which are detailed below. As such, while my impressions of the paper are still largely positive, I suggest another revision to remedy these issues before publication.

Issues

- While I appreciate the addition of a model recovery exercise, I'm afraid the present version is not sufficient to fully support the authors' conclusions: based on the Supplemental materials, simulated data is seemingly only generated for the REP model, so while this model may be recoverable, it remains uncertain whether the REP model also captures the behaviour of other generating models. The proper test would be to simulate and fit data from all models in a full $N \times N$ matrix – this is of course a more intensive procedure so I understand some hesitancy, but this is necessary to properly test the discriminability of the models. This is not to say I expect the REP model to do this, but without this complete comparison, this possibility cannot be eliminated.

- I also did not follow the procedure of the recovery analysis: at first, it seemed this only used 10 simulations, which seemed low, but from the Methods it seems this may have been 10 subsets (of unspecified size) out of 10,000 total simulations, though I'm still not certain. Adding to the confusion, Supplemental Figure 6A seems to suggest the number of simulations was 15 (according to the x-axis), which disagrees with all the other numbers. Part of the issue here may be the ambiguous use of the term 'dataset', which could refer to data from a single subject, data from multiple subjects or a whole task. More clarity on the recovery procedure is needed.

- Following on from this, if model recovery was indeed split into subsets, I don't understand the purpose of this: why fit 10 separate sets rather than one total set? My speculation is that this is intended as a compromise between aggregate and individual-level fits, giving a single best-fitting model in each subset but an overall recovery rate across subsets - this may be what is meant when referring to variations in best fit preventing perfect recovery on page 13 of the Supplemental. If this is the case, I must disagree with this procedure: the purpose of model recovery is to measure how often two models are confused precisely because of similarities in predicted behaviour to allow interpretation of model fitting. This then casts doubt on the perfect recovery reported in the main text, which may be more divided than this suggests. I advise the authors to consider their recovery results in more detail, and potentially add more nuance to their conclusions depending on their findings.

- The addition of the mixed-effects regressions is very welcome to make better use of the data, though I believe these are still restricted to one or two points per person so some of the previous criticisms on statistical power may still apply (though admittedly reduced). As a minor comment here: is there a particular reason to include the intercept estimates? From my understanding, the slopes are the key test of the frequency effect, while intercepts presumably reflect a general bias towards/away from the target choice (there is a suggestion at the end of the caption for Figure 4 that both slope and intercept contribute to the effect, though I do not see how). If so, the authors may wish to remove intercepts to focus on the key slope values, though if I'm mistaken, I would ask for some explanation in the text.

- While Bayes factors are calculated for the mixed-effects regressions, these are all relegated to the supplemental – it would be helpful to put these in the main text somewhere to quantify these effects rather than relying on solely Figure 4.

- I understand the WAIC measure was added in response to reviewer comments, but I would add that using two model criteria is unnecessary and distracting, particularly when their results seem so consistent – I suggest the authors pick one of these and use only that for model comparison.

- I'm grateful for the addition of participant-level model fits, though I'm disappointed this is only given in the Supplemental – some mention of these results in the main text would be beneficial to better reflect the individual variation in model performance.

- A very minor question: what is the difference between Figure 4 and Supplemental Figure 1?

Typos

Page 6, Figure 1 caption: 'For all new probabilistic tasks tested two cross-context...'

Page 12, Figure 3D caption: 'r = -.43'

Page 14, paragraph 1: 'r = -.43'

Supplemental, page 8: 'Supplemental Figure 3 and 4' should be 'Supplemental Figure 2 and 3'?

as a supplementary peer review file. However, on author request, confidential information and data can be removed from the published reviewer reports and rebuttal letters prior to publication. If your manuscript has been previously reviewed at another journal, those Reviewers' comments would not form part of the published peer review file.

Communications Psychology is committed to improving transparency in authorship. As part of our efforts in this direction, we are now requesting that all authors identified as 'corresponding author' create and link their Open Researcher and Contributor Identifier (ORCID) with their account on the Manuscript Tracking System prior to acceptance. ORCID helps the scientific community achieve unambiguous attribution of all scholarly contributions. You can create and link your ORCID from the home page of the Manuscript Tracking System by clicking on 'Modify my Springer Nature account' and following the instructions in the link below. Please also inform all co-authors that they can add their ORCID to their accounts and that they must do so prior to acceptance.

Version 2:

Decision Letter:

Dear Dr Wagner,

Your manuscript titled "Explaining decision biases through context-dependent repetition" has now been seen by our reviewers, whose comments appear below. In light of their advice I am delighted to say that we are happy, in principle, to publish a suitably revised version in Communications Psychology.

We therefore invite you to revise your paper one last time to address the remaining concerns of our reviewers and a list of editorial requests. At the same time we ask that you edit your manuscript to comply with our format requirements and to maximise the accessibility and therefore the impact of your work.

EDITORIAL REQUESTS:

SUBMISSION INFORMATION:

OPEN ACCESS:

* **CODE AVAILABILITY:** All Communications Psychology manuscripts must include a section titled "Code Availability" at the

end of the methods section. We require that the custom analysis code supporting your conclusions is made available in a publicly accessible repository at this stage; please choose a repository that generates a digital object identifier (DOI) for the code; the link to the repository and the DOI must be included in the Code Availability statement. Publication as Supplementary Information will not suffice.

*** DATA AVAILABILITY:**

Link Redacted

Best regards,

Troy Lui, on behalf of

Jesse Rissman

Troy Lui, PhD
Associate Editor
Communications Psychology

Jesse Rissman, PhD
Editorial Board Member
Communications Psychology
orcid.org/0000-0001-8889-5539

REVIEWERS' COMMENTS:

Reviewer #1 (Remarks to the Author):

I thank the authors for thoroughly engaging with the reviewers' comments - I believe the paper is correspondingly strengthened, and will be a solid contribution to the literature.

Reviewer #2 (Remarks to the Author):

Wagner, Wolf, and Kiebel have made extensive edits to the manuscript after the first reviews, which I appreciate greatly. As a disclaimer, I have to point out that I did not see the rebuttal and revisions of the first round of reviews and could not provide feedback on how the authors addressed my previous concerns yet. However, from what I see in the current version of the manuscript and supplemental information, I am satisfied with the authors' revisions and have only very minor comments I would like to draw their attention to.

1. In Results, Assessment of Preferences, you give p-values and credible intervals (CIs), but it's not the CIs of the p-values but of the probability value associated with the binomial test. All other CIs in the manuscript relate to the value/test statistic that was presented directly in front of the CI. Please specify once to which kind of variable the CIs in that section belong so as not to confuse your readers.
2. I would recommend having the "Sample Size" section in the Methods directly after the section "Participants" and before "Tasks".
3. Whenever you refer to the Method section throughout the manuscript, please refer to the specific section of the Methods instead of just writing "see methods". The Methods section is too long to search for where that specific detail is located whenever being referred to that section in the manuscript.
4. You define WAIC to be the "Widely Applicable Information Criterion" in the Methods section and the "wataike information criterion" in the Results section.
5. I still think some of the text/legends/axis captions in Figure 2 are a bit too small to be easily legible.

Reviewer #3 (Remarks to the Author):

Summary

The authors have again done well in answering my previous comments on the paper, and I especially appreciate the additional simulations performed for the expanded model recovery exercise given the mentioned computational cost. While there are still some smaller issues that could be addressed, I do not believe these should prevent the work from being published, so I offer them only as suggestions to consider. Thank you for your thoughtful responses to my previous queries, and I wish you the best for your work going forward.

Minor Comments

- I'm grateful for the further expansion of the model recovery exercise, which I think was worth the extra effort to solidify the discriminability of the models. This being said, I'm afraid I still do not completely follow the procedure here, or at least the logic of it: why simulate 10,000 datasets if only 30 were actually examined? These do not seem to be required for the MCMC sampling process, so this appears to be wasted effort (even if not the main computational cost of the exercise), unless there is some rationale here I'm missing. Alternatively, if all 10,000 were fit, what do the 30 samples refer to? Adding to the confusion, the Supplemental states 'We then draw 20 random samples and refitted the whole task simulated datasets with all models 20 times' (page 14, line 157), suggesting each dataset was fit multiple times, which seems misaligned with the description in the main text. It is possible the precise details of this exercise are not hugely consequential to the actual results if recovery rates are high regardless, but I would add that clarity on procedure is important to properly interpret these results, and lay out the assumptions involved in these analyses.

- Following on from this, I'm also slightly concerned that declaring a single best-fitting model in each simulated dataset (i.e. task) could amplify model discriminability: collecting measures across simulated "participants" may increase the margins between models, making them appear more distinct. This seems particularly relevant given the apparent heterogeneity in model fit in the empirical data: is this truly reflective of individual differences in model use, or could this be produced by a single common model? It may be worth examining the recovery results at the individual level to see whether recovery remains high even for a single subject.

- Thank you for explaining the value of the intercept estimates in your regression – I see the purpose of these now, and agree that the alternate analysis removing the intercepts is unnecessary.

Typos

Page 3, paragraph 1: participants' -> participants

Response to the Editor and Reviewers for Wagner et al., “Explaining decision biases through context-dependent repetition” in *Communications Psychology*

Manuscript ID: COMMSPSYCHOL-25-0091-T

General remarks:

We thank the Editor and the three Reviewers for their careful evaluation of our manuscript and for the constructive suggestions. This feedback enabled us to improve our analyses, increase transparency, and clarify the interpretation of context-specific repetition effects. Below we provide (1) a concise overview of the major changes; (2) a point-by-point reply to the three reviewers' comments and (3) precise pointers (page numbers) to each revision. In line with the Editor's request for greater clarity regarding repetition effects, experienced value, individual-level modelling, diagnostic checks, and methodological detail, we have made the following key revisions:

1. Mixed-effects analyses: All behavioural group level correlations are now supplemented by study-specific mixed-effects logistic regressions with participant-level random intercepts and slopes (Fig. 4 and Supplemental Fig. 1, Supplemental Table 1). Repetition during the learning phase remains a significant predictor of transfer preference throughout.
2. Individual-level model diagnostics: We now report WAIC and further model diagnostics for every participant (Supplemental Fig. 2 and 3).
3. Qualitative model checks: For three tasks in which the relative-learning and repetition accounts make divergent predictions, we provide forward simulations and compare them to the empirical Q-values and choice values (Supplemental Fig. 8).
4. Parameter- and model-recovery: We now report recovery analyses for two key datasets to demonstrate that the REP model is identifiable (Supplemental Fig. 6 and Fig 7) and its parameters are recoverable (Supplemental Fig. 4 and Fig. 5).
5. Further inspection of the model behaviour: We observed that the baseline learning rates (learning at $t=1$) in the decay models consistently converged to values close to 0.99. This pattern is theoretically highly plausible, as participants are likely to be highly explorative at the beginning of the experiment and thus integrate new information almost fully. Based on this observation, we decided to fix the learning rate at 1 in the decay-based model variants to reduce parameter redundancy and improve model identifiability. We then refitted all datasets. Model variants with separate learning rates for chosen and unchosen options were unchanged.

6. Identification of inconsistent implementation of the REP model in one study: Both datasets from Bavard and Palminteri (2023) differ from the others in that they include a “blocking” manipulation: in certain trials, the highest-value option within a context is made unavailable, thereby increasing the frequency of choosing the second-best option. For example, in context 3, the best option is blocked in a subset of trials, forcing participants to choose among the remaining alternatives. In our initial implementation of the REP model, the repetition bias was unintentionally not applied to the second-best stimulus during these blocked trials. When correcting this, i.e., when applying the REP bias consistently across all trials, including blocked ones, this results in an increase for the repetition bias for the second-best option and a reduction in the repetition bias for the best (but unavailable) option. This, in turn, led to a worse overall fit of the model. That is, reinforcing the second-best option in these forced-choice trials diluted the model’s ability to capture the strong preference participants showed for the best option once it became available again. In contrast, omitting the REP bias in these trials appears to better reflect participants’ behavior, possibly because such choices might be less driven by context-specific habits/repetition (which would favor the best option). Despite this finding, the REP model still substantially outperformed all alternative models ($\Delta\text{DIC}/\text{WAIC} > 1000$). For transparency and consistency across datasets, we now report results for the version in which the REP bias is applied across all choices in the main manuscript (like in all other tasks), including blocked trials. This observation is briefly noted in the results section on page p22:

“Interestingly, in this dataset, model fit of the REP model further improved substantially when the repetition bias was not applied to trials in which participants were constrained to select the second-best option due to the best option being blocked. Future research could further explore this effect.”

7. All typographical and stylistic errors mentioned by the reviewers have been corrected.

Below we address each point in the order raised. Reviewer comments appear in italics, followed by our responses in plain text. We believe these comprehensive revisions satisfy the concerns raised and substantially strengthen the manuscript. We look forward to the Editor’s and Reviewers’ further feedback.

Reviewer #1 (Remarks to the Author):

...I think the conclusions are probably accurate (partly because at least two other preprints draw the same conclusions in different settings with different analyses [1,2]), and I think this study is worthwhile and timely. Unfortunately, I do not think the conclusions follow from the analyses presented here, as I see serious methodological/statistical/modeling issues in this version of the paper. In addition to methodological issues, another major issue is with the interpretation of the findings. While the paper is overall well written, there are a few places where information is hard to find, and clarification would be helpful.

Response:

We thank the reviewer for finding our study worthwhile and timely and giving us the opportunity to clarify. Guided by your comments, we have substantially expanded the methodological detail, added new analyses to address each statistical and modelling concern, and clarified the presentation of key results. We hope these revisions resolve these issues and strongly believe that they better substantiate our conclusions.

Major:

Reviewer #1):

1. Interpretation. To me the most substantive issue in the paper is in the interpretation of the role of repetitions. There are at least two ways in which repetitions could impact choice: directly [the proposed model and implicit interpretation], and crucially, indirectly [by impacting the estimation of the experienced value]. This is not discussed or controlled for statistically anywhere in this article. For example, P7: the authors claim that they control for value by selecting “equally valued” pairs. However, the equal value is assumed by design (i.e. the experimenter designed this as a $p=.7$ pair), but not guaranteed in terms of the participant’s experience, given the probabilistic nature of feedback (and despite the good randomization controls reported in methods) and given participants’ probabilistic action selection. A participant who chose the high value option once, got unlucky, received 0, and never selected that option again, has a subjective estimate closer to 0 than to .7. Thus, the analysis does not actually control for (experienced) value (or relative values). [3] demonstrates how taking this experienced value into account (rather than objective) can change interpretation in probabilistic tasks. It is crucial that the statistical analyses be deepened to actually control for experienced value in parallel to repetitions (e.g. in multiple regression frameworks).

Response: We thank the reviewer for the opportunity to clarify this important point. There are three reasons why we are relatively certain we control for an experienced value effect. First and most importantly, the experiments were intentionally designed to minimize value-estimation confounds. Specifically, reward sequences were generated so that every stated probability, including the 0.7 options, were realized within every block of ten to twelve trials and across the entire learning phase. Second, since participants received full feedback in 13 of the 15 tasks, they observed every outcome associated with each option. Thus, even if a participant was momentarily unlucky with the 0.7 option and switched away, they still saw subsequent outcomes of the 0.7

option on the very next trial. Only two of the datasets used in the frequency or frequency-valuation correlation (p2 and g2) involved partial feedback and only one of them was probabilistic. Accordingly, in the overwhelming majority of cases the experienced value is very well aligned with the true absolute or relative value, making our equal-value comparisons appropriate when decision-makers have full knowledge of the environment. To address the reviewers concern we have also repeated our new analyses (mixed effects regression with subject level random effects) after excluding the partial-feedback tasks (see Supplemental Figure 1). Third and finally, note that our analyses focus on the transition from the end of the learning phase to the transfer phase, further alleviating any potential residual impact of experienced value during initial trials.

Reviewer #1: 2. *The same statistical methodological issues confounds the correlation analyses - frequency correlation between learning and transfer needs to be corrected for relative experience outcomes.*

Response: In light of our hierarchical regression analyses, which include subject-level random effects, and the exclusion of tasks with partial feedback, we believe that confounding with subjective value estimates is highly unlikely. Specifically, as the vast majority of tasks provided full feedback and participants observed the outcomes of all options on every trial, regardless of their choices. This implies that their experienced value should correspond to the average observed outcome for each option, irrespective of individual choice patterns. While one could, in principle, use model-derived subjective Q-values to quantify value estimates, such estimates would however depend on model structure and parameter combinations, introducing circularity if used as control variables. We appreciate the importance of communicating this issue clearly to readers and have therefore added the following note to **p.11** of the manuscript:

“Note that due to full feedback in most tasks, all participants observed all rewards, making it unlikely that the observed frequency effects are confounded by differences in experienced value.”

Reviewer #1: 3. *Experiment 4.1 reports a null result that is not very informative in the absence of either formal null hypothesis testing, or interaction analysis with exp 4.2. Note that the 0.16 probability difference between the transfer options is extremely small and would not be expected to be strongly noticeable, in particular with the low N=30 used.*

Response: We thank the reviewer for pointing that out and changed our wording in the manuscript. We now explicitly write **(p7)**:

“In task 4.1, where participants chose options, during learning, in approximately the same relative choice frequency (green and orange lines in Figure 2D), we do not find evidence for clear choice preference during transfer.”

Reviewer #1: 5. *The statistical analyses appear to consistently ignore random effects, treating the whole group as a single participant, which is highly unusual and likely to artificially inflate statistical outcomes. This is true in the fig. 2 analyses, as well as in the correlation analyses (see*

fig. 3) where the correlation is measured not across participants of a task, but across conditions and tasks, averaged over participants. This raises the concern that results could be driven by a few unlucky participants learning the wrong associations in LC conditions, selecting it more often, and having incorrect estimates for the other option, a completely different interpretation.

Response: We value this comment and agree that relying solely on group-level aggregates can mask individual variability. In the original submission we collapsed data across participants for Fig. 2 and the frequency-preference correlations because each task delivers only two to four critical transfer trials per option pair (a design inherited from Klein et al. 2017 and Bavard et al. 2018). Pooling therefore allowed us to visualize the effect on the group level and show correlations with acceptable statistical power. To alleviate these concerns and to rule out the possibility that a handful of participants drove the result, we have carried out a full set of Bayesian generalized linear mixed-effects models using the package *brms* which interfaces to the probabilistic programming software Stan (2024; Stan Development Team). For every dataset we now model transfer preference in the first cross-context comparison as a Bernoulli outcome with a logit link (see Eq 4 below), (a) on the complete set of pairwise comparisons pooled across the seven probabilistic tasks (including Klein et al. 2017), (b) on the two Gaussian tasks, (c) on the two Bavard et al. 2018 datasets, and (d) on the two Bavard et al. 2023 datasets. We repeated the analysis for (1) all comparisons, (2) only equal-absolute-value pairs, and (3) only equal-relative-value pairs. To guard against experienced-value confounds, we also reran all models after removing the partial-feedback tasks (p2, g2). The key result is that across all but one task pool for each equal absolute and equal-relative value pairs, the slope β_1 remains credibly positive ($\text{CrI} > 0$) and Bayes factors favour the full (random intercept, random slope) over the null model ($\text{BF}_{10} \geq 1000$), confirming that learning-phase choice frequency predicts transfer preference after accounting for participant-level random effects (see new Figure 4 and Supplemental Figure 1).

We now report these additional analyses in the results section on p11 and p12 and added a detailed description to the method section on p34.

p11:

“To further quantify the relationship of choice frequency during learning on transfer preferences, we conducted a series of Bayesian hierarchical mixed-effects logistic regression analyses that accounted for subject-level variability via random intercepts and slopes. As in previous analyses, we partitioned decisions into three subsets: (1) all cross-context comparisons, (2) pairs with equal absolute values, and (3) pairs with equal relative values (based on context-normalized values; see Methods for details). These analyses were carried out separately for our two newly collected task classes (probabilistic and Gaussian) as well as for all previously published studies. Across the pooled dataset and individual studies, 95% credible intervals (CIs) for intercepts were consistently negative, and for slopes consistently positive in the majority of conditions. A significant positive slope indicates that higher relative choice frequency during learning was associated with a greater probability of choosing that option during transfer. Specifically, we found robust positive slope estimates (95% CIs not overlapping zero) in the pooled dataset, the probabilistic tasks, and in the Bavard and Palminteri (2023) dataset for both the equal absolute subsets. Similarly, the pooled data, Bavard et al. (2018) and Bavard and Palminteri

(2023) showed positive slope estimates for equal relative value comparisons. In contrast, the 95% CIs overlapped with zero in the Gaussian tasks (equal absolute value subset) and in the probabilistic tasks (equal relative value subset), likely reflecting limited statistical power due to the small number of relevant comparisons. In these subsets, only two specific transfer decisions contributed data (ALC vs. DHC in both Gaussian tasks and ALG vs. CLG in p4.1 and p4.2), which constrains interpretability. Nevertheless, numerical trends in these subsets were consistent with a frequency-based bias in preference formation. These findings provide converging evidence across modeling approaches, value-matched comparisons, and datasets, that choice frequency during learning systematically biases subsequent preference, even in the absence of objective value differences (see Figure 4 for an overview and Supplemental Table 1 for all parameter estimates and Bayes factors for model comparison).”

p34:

“Mixed-effects Bayesian logistic regressions

To test whether an effect of choice frequency once individual variability is taken into account, we fitted a Bayesian generalized linear mixed model with a Bernoulli likelihood and logit link:

$$Choice_{i,j}^{HC/HG} = \text{Bernoulli} \left(\text{logit}^{-1}(\beta_0 + \beta_1 \text{FreqLearn}_{i,j} + u_{0,j} + u_{1,j} \text{FreqLearn}_{i,j}) \right) \text{ (Eq 4)}$$

In this formulation, i indexes first-transfer trials, j indexes participants, $\text{FreqLearn}_{i,j}$ is the centered relative frequency with which one option was chosen during the learning phase, β_0 and β_1 are group-level intercept and slope capturing the average baseline preference and the mean effect of frequency (repetition), and $u_{0,j}$ and $u_{1,j}$ are participant-specific random intercepts and slopes (assumed jointly normal with zero mean and an unstructured covariance) that allow each individual to deviate from the population values. A posterior estimate of β_1 that is credibly positive would demonstrate that greater learning-phase repetition increases the likelihood that a specific option is selected in the transfer phase. All models were fitted via the brms package (v2.20) which interfaces Stan (2024, Stan Development Team) for full-Bayesian inference. Weakly informative priors were used (Normal (0, 1) for fixed effects; Exponential (1) for standard-deviations). Models were fitted per four chains \times 4,000 iterations (2,000 warm-up). For every data set we fitted the full model above, and a null model containing only group and random level intercepts. The relative evidences of these models were quantified with Bayes factors computed via marginal-likelihood estimation in brms. Values > 10 are considered strong evidence for the frequency effect.”

Figure 4 Mixed effects Bayesian logistic regressions to model the effect of choice frequency during learning on preference in transfer. We here analyzed three pooled choice sets (“all pairs”, “equal absolute value”, “equal relative value”) across all new datasets and published studies to quantify whether preference in the first transfer trial is related to normalized choice frequency during learning. Rows distinguish group-level intercept vs. slope parameters. Points are posterior means; red intervals show parameter estimates whose 95 % CI excludes 0 (i.e. credible frequency (repetition) effects), grey intervals represent intervals overlapping with 0. The dashed vertical line at 0 aids visual inspection; intervals for the slope entirely to the right show a positive frequency/repetition preference relation. A negative intercept combined with a positive slope means that increased repeated choice of an option increases its likelihood of being chosen at transfer, i.e. a repetition bias.

Supplemental Figure 1 Mixed effects Bayesian logistic regressions without partial feedback tasks. We here analyzed three pooled choice sets (“all pairs”, “equal absolute value”, “equal relative value”) across all new datasets and published studies to quantify whether preference in the first transfer trial is related to normalized choice frequency during learning. Rows distinguish group level intercept vs. slope parameters. Points are posterior means; red intervals show parameter estimates whose 95 % CI excludes 0 (i.e. credible frequency (repetition) effects), grey intervals represent intervals overlapping with 0. The dashed vertical line at 0 aids visual inspection; intervals for the Slope entirely to the right show a positive frequency/repetition preference relation. A negative intercept combined with a positive slope means that increased repeated choice of an option increases its likelihood of being chosen at transfer, i.e. a repetition bias. Note to control for experienced value both partial-feedback tasks (p2 and g2) are omitted, so all estimates are based exclusively on full-feedback datasets where all rewards were observable.

Supplemental Table 1 Hierarchical (multi-level) logistic regressions linking learning-phase choice frequency (*FreqLearn*) to transfer-phase choice preference (Choice^{HC/HG}). Posterior summaries are shown for the fixed-effects intercept (b_Intercept) and the slope of *FreqLearn* (the normalized frequency during learning). Estimate: Posterior mean of the regression coefficient (on the log-odds scale); 2.5 % and 97.5 % = lower and upper bounds of the central 95 % credible interval (CI). BF₁₀ = Bayes factor favouring the model that contains the random slope and intercept over a null model without these; values “ > 1000” indicate BF₁₀ ≥ 1000. ✓ in the Sig95 column marks coefficients whose 95 % CrI does not include zero. Choice set indicates the subset of cross-context stimulus pairs that was analyzed (“all pairs”, “equal absolute value”, “equal relative value”). Study/Task names the experiments: “Pooled data” refers to the joint analysis across all datasets.

Choice set	Study/Tasks	Term	Estimate	2.5%	97.5%	BF ₁₀	Sig95
all pairs	Pooled data	b_Intercept	-1.609	-1.727	-1.492	> 1000	✓
all pairs	Pooled data	b_Value1L	3.875	3.642	4.109	> 1000	✓
all pairs	Bavard et al. 2018	b_Intercept	-1.863	-2.282	-1.430	> 1000	✓
all pairs	Bavard et al. 2018	b_Value1L	5.090	4.296	5.844	> 1000	✓
all pairs	Bavard et al. 2023	b_Intercept	-1.506	-1.630	-1.383	> 1000	✓
all pairs	Bavard et al. 2023	b_Value1L	3.683	3.442	3.918	> 1000	✓
all pairs	Gaussian tasks	b_Intercept	-4.556	-5.134	-3.986	> 1000	✓
all pairs	Gaussian tasks	b_Value1L	8.228	7.353	9.106	> 1000	✓
all pairs	Probabilistic tasks	b_Intercept	-1.298	-1.748	-0.845	> 1000	✓
all pairs	Probabilistic tasks	b_Value1L	2.143	1.354	2.934	> 1000	✓
equal abs value	Pooled data	b_Intercept	-1.876	-2.111	-1.655	> 1000	✓
equal abs value	Pooled data	b_Value1L	2.314	1.901	2.736	> 1000	✓
equal abs value	Bavard et al. 2023	b_Intercept	-1.850	-2.095	-1.619	> 1000	✓
equal abs value	Bavard et al. 2023	b_Value1L	2.186	1.774	2.607	> 1000	✓
equal abs value	Gaussian tasks	b_Intercept	-1.443	-2.548	-0.361	1.23	✓
equal abs value	Gaussian tasks	b_Value1L	0.674	-1.294	2.612	1.23	✓
equal abs value	Probabilistic tasks	b_Intercept	-1.711	-2.440	-0.993	81.1	✓
equal abs value	Probabilistic tasks	b_Value1L	2.377	0.925	3.830	81.1	✓
equal rel value	Pooled data	b_Intercept	-1.264	-1.510	-1.018	> 1000	✓
equal rel value	Pooled data	b_Value1L	1.957	1.494	2.414	> 1000	✓
equal rel value	Bavard et al. 2018	b_Intercept	-0.691	-1.286	-0.085	> 1000	✓
equal rel value	Bavard et al. 2018	b_Value1L	2.779	1.635	3.931	> 1000	✓
equal rel value	Bavard et al. 2023	b_Intercept	-1.361	-1.636	-1.083	> 1000	✓

Choice set	Study/Tasks	Term	Estimate	2.5%	97.5%	BF ₁₀	Sig95
equal rel value	Bavard et al. 2023	b_Value1L	1.557	1.057	2.054	> 1000	✓
equal rel value	Probabilistic tasks	b_Intercept	-1.157	-2.363	-0.033	1.82	✓
equal rel value	Probabilistic tasks	b_Value1L	0.732	-1.155	2.595	1.82	

Reviewer #1: 6. *Modeling results rely completely on quantitative model fit comparisons at the group level (DIC); however, this is known to often be inadequate [4] for testing competing theories. Instead, differential predictions should be qualitatively shown and compared [5]. The methods mention posterior predictive checks, but corresponding results are not reported (that I could find).*

Response: We thank the reviewer for pointing that out. We now verify that the model ranking is not driven by a subset of extreme participants. To do this, we now compute participant-specific WAIC (Vehtari et al. 2017) for every model and dataset and repost these in the Supplementary Information. Please also note that we showed the results of posterior predictive checks (proportions of correctly predicted binary choices) in Figure 6C-D on page 17. Here, we simulated choices from each model and compared which model was best in predicting participants. Averaged across 10,000 simulation the REP model on average predicted the highest number of choices across all datasets. We believe that showing individual level fits and % of best fit participants by all models gives a detailed picture, thus we removed the aggregated PPC plot and point the reader to the Supplemental Figure 2 and 3 below.

“Participant level model diagnostics

For each hierarchical fit we extracted the stored point-wise log-likelihood matrix, sliced the appropriate columns for every participant’s trials, and computed the participant level WAIC (Vehtari et al., 2017). We then (1) summarized the participant distribution (first column in Supplemental Figure 2 and 3), (2) reported the percentage of participants best-fit by each model (second column in Fig. 2 and 3), and (3) repeated the classical group comparison with Δ WAIC (third column in Supplemental Figure 2 and 3). The results corroborate the group-level DIC pattern: the repetition-bias (REP) model yields the lowest WAIC in most participants across most tasks and remains the global optimum at the group level except for tasks p4.2 and g2. While the advantage of the REP model is expressed at the individual level one can also see that in a subset of tasks the REL model also performs well at explaining participant behavior. However, we explicitly tested this in our task design and also show using simulations that the REL model digresses from its own predictions (see Supplemental Figure 8). Note that when computing the participant level WAIC from group level fits, these values are obtained after shrinkage, i.e. borrowing information through the group level parameters. However, we used uninformative priors on the group level parameters, and any possible shrinkage affects all models equally, so differences in WAIC per participant should remain meaningful (Vehtari et al. 2017).”

Supplemental Figure 2 Dataset-wise model comparison at the participant-level using WAIC. Each task is represented by a three-panel row (left to right). A, G, D, J, M, P, S: Individual WAIC: every thin line connects the WAIC values of one participant across the fitted models; semi-transparent dots are the raw scores; the box-and-whisker plot summarises the distribution and the red diamond marks the sample mean. B, E, H, K, N, Q, T: Proportion best: bars show the percentage of participants for whom a given model achieved the lowest WAIC (ties count for all tied models; numbers above the bars give the exact percentages). C, F, I, L, O, R, U: Δ WAIC: bar height equals the difference between the summed WAIC of a model and the overall best (lower is better; the

optimum therefore appears at 0). WAIC was computed from the point-wise log-likelihood columns stored in the hierarchical MCMC output; for every participant we sliced the columns belonging to their trials and applied WAIC via the loo R package (Vehtari et al. 2017).

Supplemental Figure 3 Dataset-wise model comparison at the participant-level using WAIC. Each task is represented by a three-panel row (left to right). A, G, D, J, M, P: Individual WAIC, every thin line connects the WAIC values of one participant across the fitted models; semi-transparent dots are the raw scores; the box-and-whisker plot summarizes the distribution and the red diamond marks the sample mean. B, E, H, K, N, Q: Proportion best, bars show the percentage of participants for whom a given model achieved the lowest WAIC (ties count for all tied models; numbers above the bars give the exact percentages). C, F, I, L, O, R: Δ WAIC, bar height equals the difference between the summed WAIC of a model and the overall best (lower is better; the optimum therefore appears at 0). WAIC was computed from the point-wise log-likelihood columns stored in the hierarchical MCMC output; for every participant we sliced the columns belonging to their trials and applied WAIC via the loo R package (Vehtari et al. 2017).

Reviewer #1: 7. P15 – competing model predictions are derived in theory by reasoning through the model logic, but not in practice with simulations. I do not think that the authors’ reasoning is always valid or parameter independent; for example for the reasons highlighted in point 2 (i.e. hysteresis effects of choices leading to different sampling of the environment leading to differences in how much has been learned for the values). This is a further argument for the need for simulations with fit parameters and qualitative comparison of model predictions.

Response: We agree with the reviewer that model simulations will help to improve the manuscript and now report these in the Supplementary Information of the paper, as follows:

“Model simulations

Given that for most tasks the majority of models do predict biased preference in the same direction, we benchmarked forward predictions with reasonable parameters against fitted latent trajectories (simulated from the fitted parameter estimates) for the three newly collected probabilistic tasks in which the relative-value (REL) and repetition (REP) accounts diverge most strongly (p3, p4.1 and p4.2). Concretely, for every candidate mechanism (ABS, REL, REP and IER) we first simulated 50 synthetic participants with reasonable parameters (decay-rate of 0.8, baseline learning rate of 1, softmax β of 5 and η repetition of 0.4). Each synthetic participant’s choices were then propagated through the model’s own update rules to yield trial-by-trial latent values, plain Q-values for ABS, composite C-values for REP (Q + choice/repetition values) and Q-differences for REL, which are plotted in the first, second and fourth columns of Supplemental Figure 8. Next, we re-fitted the REP and REL models to the behavioural data and simultaneously simulated latent Q- and choice values from the posterior parameter estimates, inferred from the data, for every trial. These fitted trajectories appear in the third and fifth columns of Supplemental Figure 8. It is insightful to visually compare the models’ unconstrained forward predictions with these fits to provide a test of qualitative adequacy. Across all three tasks only the REP model simultaneously recovers the learning-phase dynamics that can explain the observed transfer preferences. In contrast, the REL model predicts indifference and can account for the observed preferences only post-hoc by learning context-specific gaps that contradict its own theoretical prediction (see Supplemental Figure 8).”

Supplemental Figure 8 Each row depicts one task in which the candidate mechanisms diverge qualitatively: p3 (top), p4.1 (middle) and p4.2 (bottom). Columns show, from left to right: ABS simulation, REP simulation, REP fitted to experimental data, REL simulation, and REL fitted to experimental data. Dots are trial-wise group means. Here only the REP model simultaneously reproduces both learning-phase dynamics that are in line with the observed transfer choices in all three tasks (see. Figure 2 in the main manuscript). In task p3 (row one) the REP simulation correctly predicts higher average combined values for stimulus A. That is this stimulus, despite its lower relative value in the LC context, on average overtakes stimulus C during learning, which we also see in the averaged fitted Q- and choice values (see top row, column two and three). In contrast, the REL simulation learns higher relative values for the best stimulus C in context 2 and would therefore predict preference for this stimulus (see column four in the first row). In task p4.1 the REP simulation (and fit) assigns less divergent combined values to stimulus A (red) and C (blue) than in task p4.2, where the REP simulation captures the larger between-context gap that drives a strong preference for the best HG stimulus. The REL model predicts indifference for simulations in both of these tasks and can only mimic a preference C (blue) by learning a bigger value difference in context 2 (HG) than in context 1 (LG; see column five in the bottom row). This is an ad-hoc departure from its own theoretical logic, which would predict indifference otherwise. Thus, across these tasks, the repetition-bias mechanism is the only mechanism whose neutral forward predictions align with the fitted latent values and with participants' transfer preferences (see Figure 2 in the main manuscript).

Minor:

Reviewer #1: 8. Please report other parameter fits (not just repetition learning rate).

Response: We now report the 95% HDI for hyperparameter distributions for learning decay, and SoftMax β and ω in IER models) in the supplement (see Supplemental Figure 10, 11 and 12)

Reviewer #1: 9. The difference between 4.1. and 4.2 was difficult to figure out, but seems an important piece of information. More generally, some information from the methods could productively be introduced earlier in the results to help with results interpretation.

Response: We thank the reviewer for pointing that out. We now highlight this difference more prominently when reporting the results on p19:

“Further in task p4.1 we implemented that participants receive counterfactual feedback for the best option in the HG context ($C_{HG}[0.76]$) during early trials; in p4.2 this counterfactual information was absent during the beginning of the task. Because counterfactual feedback slows accumulation of positive evidence, the best HG option ($C_{HG} = 0.76$) is repeated less often in p4.1 than in p4.2, thereby decreasing repetition of this option and allows us to investigate its influence on transfer preferences.”

Reviewer #1: 10. Modeling results should report on participant heterogeneity, as should behavioral analyses, rather than being only at the group level.

Response: We thank the reviewer for reminding us and agree that documenting individual variability is essential. In the revised manuscript we now (1) present Bayesian mixed-effects regressions for all key behavioural tests, including random intercepts and slopes for all participants (2) provide participant-wise model diagnostics for every competing model; and (3) summarise the percentages of participants best described by each model (see our response to comment #6 above). Together we believe these additions make the degree of heterogeneity across observers fully transparent.

Reviewer #2 (remark to the authors):

This manuscript is timely and relevant, well written, clear, and structured. The research questions and hypotheses are well informed by the literature, and the methods are sound and well-chosen to answer the question at hand. Moreover, openly sharing the data and analysis code (on publication) is commendable (though making these already accessible for us reviewers would have been even better). I have read the paper with great enthusiasm. However, I would like to raise a few issues I would like the authors to address.

Response: We thank the reviewer for their positive assessment. All material will be made publicly available on OSF (<https://osf.io/zj95m/>) within the next days. In this revision we have addressed every point of the reviewer, providing fuller methodological detail, additional analyses, and clearer explanations throughout. We hope these changes fully resolve the reviewer's concerns.

Major:

Reviewer #2:

1. How were the sample sizes of the studies of the new dataset determined? Did you make a priori analyses of statistical power or were other variables determining sample size? Please include a sample size justification (on the level of analysis, i.e., for each task and not over the total number of participants) in the Methods section.

Response: For the nine new experiments we did not run a formal prior power analysis. Instead, we decided on a minimum sample size by benchmarking against the two most closely-related studies available when we designed the work, Klein et al. (2017) and Bavard et al. (2018). Both papers used similar probabilistic transfer-learning paradigms and reported robust context effects with a minimum of $n=21-40$ participants per experiment. To ensure at least comparable precision, and anticipating a slightly larger parameter space in some of our variants (tasks with partial feedback or three stimuli per context), we set a target of 30–50 participants per task.

We now transparently include our determination of sample size in the methods section on p28, where we write:

“Sample size

We did not run a formal prior power analysis. Instead, we decided on a minimum sample size by benchmarking against the two most closely-related studies available when we designed the work: Klein et al. (2017) and Bavard et al. (2018) reported robust effects of their normalization models in comparison to standard RL with $N = 21-40$ participants per experiment.”

Reviewer #2: 2. While reading the Results section, it was often unclear to me on how many participants/data points the analyses were based on. Please report the degrees of freedom wherever possible to help with this issue. In addition, I think Supplemental Table 4 should be included in the Methods section instead of the Supplement to make information about the number of participants and trials for each data set more easily accessible.

Response: We thank the reviewer for this comment and now substantially increased mentioning of the relevant sample size in the main manuscript. We further moved Supplemental Table 4 to the methods section of the paper (see Table 3 in the manuscript).

Reviewer #2: 3. The number of training and transfer trials in the new data sets was relatively low. What was the rationale for choosing this number of training and transfer trials?

Response: We used mostly 30 and in some cases 40/50 learning trials per context and two cross-context transfer trials (per comparison) in the new probabilistic tasks, and four probes per comparison (9 comparisons) in the Gaussian tasks. The seminal demonstrations of robust context effects by Klein et al. (2017, 30 learning trials), Bavard et al. (2018, 40), and Bavard & Palminteri (2023, 45), all reached robust effects well before 50 trials and analyzed no more than four cross-context probes in their main analyses. Klein et al. in fact based their main tests for the context effects on just the first two transfer trials. Since we likewise delivered outcome feedback during the transfer phase of our probabilistic tasks, analyzing all ten transfer choices would have conflated the original bias with new learning. Limiting the window to two trials (where the most interesting comparison of the best stimuli where designed feedback neutral) avoids this problem without adding extra learning-rate parameters for the transfer phase. By contrast, our Gaussian tasks followed Bavard et al.'s protocol and withheld feedback during transfer, allowing four unbiased probes. Altogether, these trial counts reproduce the scope of prior work while maintaining internal validity. To clarify this rationale, we now added a paragraph in the methods section on p30 of the paper:

“Transfer-phase

Design of the transfer-phase followed prior studies. For probabilistic tasks we followed Klein et al. (2017) focusing on the first (binomial tests and logistic regression) or first two transfer trials for (computational modeling). Within these tasks reward feedback was provided during transfer. However, tasks were designed in a way that there was equal feedback within the first cross-context comparison between the best stimuli to minimize differential value-updating (here reward was delivered to both options). All subsequent transfer trials (starting with the second) followed the same pseudo-random schedule used during learning. Full feedback in the transfer phase was shown in tasks that had full feedback during learning; partial-feedback tasks continued to reveal only the chosen option's outcome. For the Gaussian tasks we followed the design of Bavard et al. (2018) and Bavard and Palminteri (2023). Here no feedback during transfer was provided. For standard analysis we again focused on the first cross-context comparison of the high-reward options, while for computational modeling we included all trials and cross-context comparisons. For an overview of task dynamics during learning and transfer see Table 3.”

Reviewer #2: 4. *Some of the correlations reported in the manuscript (e.g., displayed in Figures 3B, 4G and H) are based on rather few datapoints and possibly underpowered (see Schönbrodt & Perugini, 2013; Kretzschmar & Gignac, 2019). The high uncertainty around correlation coefficients based on few data points should at least be made explicit and their limits in robustness and interpretability discussed. Better yet if all underpowered results were to be removed from the manuscript.*

Response: We thank the reviewer for this valuable feedback. We implemented two additional analyses that we hope will resolve these concerns. First, we now compute linear effects models with subject-level random effects to complement the correlation results. We report these analyses in the main manuscript on p11 and p12 and in Figure 4. Further, for our main correlational results in Figure 3 and Figure 4 we now compute these via Bayesian regressions and report the 95% CI to show uncertainty around correlations coefficients (see changes on p11, p12 and p13). We also recreated these Figures so that these include the CIs for each correlation.

Reviewer #2: 5. *Why was the assessment of preferences in the transfer phase based only on the first transfer trial? Using only the first transfer trials instead of aggregating over all relevant transfer trials decreases the reliability of this measure and might induce spurious results. Please justify the use of this statistical procedure.*

Response: We apologize that this wasn't stated clearly. We now state explicitly why only the first (or first two) transfer trials are analyzed for our new probabilistic tasks. As all landmark studies using similar paradigms limit themselves to 2-4 transfer trials (Klein 2017: 2 in their main analysis; Bavard 2018, Bavard & Palminteri 2023: 2/4), our design followed these precedents. Further, in our own probabilistic tasks feedback continues during transfer: trial 1 gives identical outcomes to the two high-value options, but from trial 3 onward differential rewards may overwrite the context effects (as found by Klein et al. 2017 in their extended analysis). Restricting the window to the first two trials therefore maximizes internal validity; in the Gaussian tasks (g1, g2), where no transfer feedback is given (following Bavard et al. 2018 and Bavard et al. 2023), all four probes are retained. Published datasets are analyzed exactly as in the original papers. Running every analysis with all transfer trials leaves every behavioural effect, correlation, and model ranking unchanged (not reported). We now clarify this in the methods section (p30 *Transfer phase*, see also our response to comment 3 above).

Reviewer #2: 6. *Were there any model diagnostics performed? Please report the results of parameter recovery and model recovery analyses for all the computational models used in the manuscript.*

Response: We thank the reviewer for this comment and performed additional analyses (parameter recovery and model recovery) for two key tasks (task p3 and Bavard et al. 2018, Exp 2) and now report these results in the supplement of the paper. Further we added an additional section to the methods section of the paper on p46:

“Model and parameter recovery

To test identifiability, we conducted full model- and parameter-recovery analysis on two representative datasets (Task p3 and Bavard et al. 2018, Exp 2). For each task we simulated 10.000 new datasets directly from the full joint posterior of the winning REP model, thereby preserving the covariance structure among all parameters (β -weights, context-specific decay rates, repetition learning-rate η). From this pool we randomly sampled ten independent datasets (subject \times trial matrices). Each replicate was then re-analysed with the entire model battery (ABS, REP, IER, REL for task p3 and ABS, REP, IER, NORM, RAN, RAN- ω for exp 2 from Bavard et al. 2018) using equal priors, sampler settings and convergence criteria of the main fits. Model recovery was evaluated with DIC and WAIC; the model with the lowest score in a replicate was deemed its winner. To assess parameter recovery, we computed correlations between the known parameter values used to generate synthetic datasets and the posterior means obtained from fitting the model to these datasets.”

We also link to the supplemental results section in main manuscript on p22 where we write:

“Parameter and model recovery

We performed full model- and parameter-recovery analysis on two representative datasets (Task p3 and Bavard et al. 2018, Exp 2). Across all ten replicates in both tasks the REP model was selected as the best-fitting model, yielding a 100% recovery rate within these datasets (for further details please see the section on model recovery in the Supplemental; Supplemental Figure 6 and 7 and Supplemental Table 5 and 6). Parameter recovery from the full posterior distribution for these tasks was moderate (see Supplemental Figure 4 and 5).”

Supplemental Figure 4 Scatter plots of true generating values (x-axis) against posterior means after refitting (y-axis) for each recoverable parameter of the REP model. Each panel corresponds to one parameter; points represent individual participants across 10 simulated datasets. The dashed diagonal marks perfect recovery. Parameters showed moderate to good recovery, while notably parameter recovery worsens in the tail of the context specific learning decays. decay_rate_c1 : $r = 0.64$, decay_rate_c2 : $r = 0.61$, decay_rate_c3 : $r = 0.78$, decay_rate_c4 : $r = 0.59$ softmax β : $r = 0.57$, η repetition: $r = 0.67$. Note that because simulating directly from the posterior we preserve the whole variance-covariance that is parameter correlations and do not fix any parameter.

Parameter recovery for Generating Model: rep

Task p3: data simulated and fit by rep (10 sims)

Supplemental Figure 5 Scatter plots of true generating values (x-axis) against posterior means after refitting (y-axis) for each recoverable parameter of the REP model. Each panel corresponds to one parameter; points represent individual participants across 10 simulated datasets. The dashed diagonal marks perfect recovery. Parameters showed moderate to good recovery, while notably parameter recovery worsens in the tail of the context specific learning decays. decay_rate_c1 : $r = 0.31$; decay_rate_c2 : $r = 0.53$; $\text{softmax } \beta$: $r = 0.80$, η repetition: $r = 0.58$.

We further added a section on model recovery within the Supplementary Information:

“Model recovery

We ran full model- and parameter-recovery exercises on two representative data sets: our core repetition-manipulation task p3 and the more complex Exp.2 of Bavard et al. (2018), which contains many cross-context comparisons in the transfer phase. In both cases, synthetic data generated by the REP model were re-analyzed with the entire model battery, and REP was correctly identified as the best-fitting model in 100% of the 10 replicates (see Supplemental Figure 6 and 7 below). This demonstrates that the REP model is fully recoverable when most participants are best explained by this model. Note perfect model recovery is not expected for every dataset: when competing models make nearly similar predictions, as for example in the Gaussian task G2 and the group Δ WAIC/ Δ DIC is much smaller the winner can vary. Hence the most practical indicator of recoverability is the share of participants uniquely best-fit by each model plus the size of the group-level information-criterion gap (see Supplemental Figure 2 and 3 and Supplemental Table 8 and 9). Overall, across the entire battery the REP model was most often the best model across the majority of participants and had the largest average WAIC/DIC advantage. We believe this confirms the REP model as the most plausible mechanism tested here, even though, complete recoverability is not guaranteed in all datasets.”

Supplemental Figure 6 REP model recovery for task p3. A: Bar plots showing how often each candidate model was selected as best-fitting across the 10 replicates (criterion: lowest WAIC/DIC). The REP model is recovered in 10/10 simulations (100%), while competing models are never preferred (bars not shown), confirming that the recovery procedure is well powered to discriminate the generative mechanism in this dataset from Bavard et al. 2018. B: Distribution of DIC differences relative to the REP model. C: Distribution of WAIC differences relative to REP model.

Supplemental Table 5 Overview of model recovery results. We simulated 10,000 new datasets from the full posterior distribution of the rep model and refitted 10 random draws of these datasets via all models. Columns show alternative model and averaged WAIC/DIC differences across all refits. Further for WAIC we show the average SE across all 10 refits. The REP model was the best model in 100% of all cases.

Task p3		
Alternative model	average WAIC difference (10 runs)	average SE
ABS	+2051.08	± 135.89
IER	+891.21	± 145.82
DIV	+3754.31	± 115.69
RANGE	+1546.03	± 142.49
RANGE ω	+1537.70	± 142.36
Alternative model	average DIC difference (10 runs)	average SE
ABS	+2072.97	-
IER	+896.99	-
DIV	+3921.75	-
range	+1540.02	-
range omega	+1542.05	-

Supplemental Figure 7 Bar plots showing how often each candidate model was selected as best-fitting across the 10 posterior-predictive replicates (criterion: lowest WAIC; DIC/LOOIC give identical counts). The REP model is recovered in 10/10 simulations for both tasks (100 %), while competing models are never preferred (bars not shown), confirming that the recovery procedure is well powered to discriminate the generative mechanism. B: Distribution of DIC differences relative to the REP model. C: Distribution of WAIC differences relative to REP model.

Supplemental Table 6 Overview of model recovery results. We simulated 10,000 new datasets from the full posterior distribution of the rep model and refitted 10 random draws of these datasets via all models. Columns show alternative model and averaged WAIC/DIC differences across all refits. Further for WAIC we show the average SE across all 10 refits. The rep model was the best model in 100% of all cases.

Bavard et al. 2018 Exp. 2		
WAIC/DIC difference vs rep model		
Alternative model	average WAIC difference (10 runs)	average SE
ABS	+2051.08	±135.89
IER	+891.21	±145.82
DIV	+3754.31	±115.69
RANGE	+1546.03	±142.49
RANGE ω	+1537.70	±142.36
Alternative model	average DIC difference (10 runs)	average SE
ABS	+2072.97	-
IER	+896.99	-
DIV	+3921.75	-
range	+1540.02	-
range omega	+1542.05	-

Reviewer #2: 7. *The variance of the repetition hyperparameter is relatively large across tasks (Supplemental Figure 1 and Supplemental Table 6). Are there task design features correlating with the parameter distribution across tasks? Can you derive any tentative insight into which task features might promote or weaken the influence of behavioral repetition from this?*

Response: Yes, certainly. We believe the repetition hyper-parameter varies across tasks because its magnitude is partially a function of task complexity and task dynamics. Although a full analysis requires in our opinion additional experimental tests and is beyond the scope of the present paper, our current working intuition is as follows. When within-context uncertainty is low and option values are easily discriminable, the repetition term mainly reflects a slow, cumulative effect and therefore remains smaller. Conversely, under higher uncertainty participants lean more heavily on recent outcomes, and the repetition term grows as it captures that short-term reliance. Said differently, it may be that the repetition bias reflects a context-specific increased probability for a specific action, but it also interacts with value uncertainty, i.e. when one is uncertain, one sticks with what worked out most often before. However, this explanation is necessarily speculative and we would refrain from discussing this but will undertake targeted experiments to test this explicitly.

Reviewer #2: 8. *Figure 2 is a very good representation of the task set-up and some important descriptive statistics. Yet I think the text of the legends, captions, and axis titles is too small and the figure caption is in part unnecessarily repetitive.*

Response: We thank the reviewer for their suggestion and improved visibility and clarity in *Figure 2*.

Reviewer #3 (Remarks to the Author):

I found this to be a relatively strong paper: the authors propose a simple method to account for observed biases in contextual learning, and perform a set of experiments to test this suggestion from multiple angles, as well as reanalysing existing data to check their results are not restricted to their own designs. Their findings are also further supported by a model comparison offering a theoretical grounding for their conclusions. I do however have some methodological confusions which I think need to be clarified (particularly on the design of the transfer trials), as well as some questions on the empirical analyses and modelling, though these can mostly be solved with edits to the text. I would therefore recommend this paper for publication following minor revisions. Below I offer some more specific comments that I feel should be answered. Thank you for the opportunity to read your work.

Response: We thank the reviewer for the positive evaluation and for pinpointing the areas that needed clearer exposition. We have expanded the description of the transfer-trial design, added step-by-step detail on the empirical analyses, and clarified the modelling framework to resolve the methodological points the reviewer flagged. We believe that these targeted edits address all the reviewer's comments.

Major:

Reviewer #3: *1. The set-up of the transfer trials is not made sufficiently clear in the text in my opinion: there is no explicit statement on which comparisons were included, or how often each comparison was given. There is some indication of this in Figures 1 and 2, as well as the supplemental materials, but even then some aspects are left unstated or even seemingly contradictory: for example, Figure 1 highlights 3 transfer comparisons in the Gaussian conditions, but Figure 2 only shows 2 (according to the black arrows), while Supplemental Table 4 mentions 9 – so which is accurate? I also found it unhelpful to refer to transfer trials as ‘per context’ when from my reading these are specifically cross-context. I suggest the authors add more detail on this key aspect of their design, and address any inconsistencies.*

Response: We thank the reviewer for this comment and agree that the specifics of the transfer trials were not explained sufficiently. We therefore have expanded the description of the transfer-trial design (p.30), added further detail in the legend of Figure 1 (explicitly stating the number of cross-context comparisons during transfer) and Supplemental Table 4, that we now moved to the methods section of the paper (Table 3). In addition, we now explicitly label transfer comparisons as “cross-context”.

Reviewer #3: *2. Following on from the previous point, I was confused about the use of feedback in the transfer trials: first, while it is explicitly stated that no feedback was given for the Gaussian tasks, the only statement on feedback for the probabilistic tasks I could find is: ‘For the first transfer trial reward feedback was equal for the best HC and LC options’ (page 27) – so feedback was given, but what was the procedure? Were rewards for the remaining trials pseudo-randomised as in the training phase, or fully random? And was this full feedback, even when training used*

partial feedback? More pressingly, why give feedback in this phase at all when this will influence subsequent choices? I'm assuming this is the reason for the restriction to only the first two transfer trials in most of the analyses (though this is not explicitly stated), but then why include feedback if it means dropping most transfer trials from consideration? Or was this restriction a post-hoc decision? There is no mention of a pre-registration for the studies so I cannot verify this, but I wonder if this issue was discovered during analysis and subsequently addressed in the Gaussian tasks. This does not invalidate the reported findings, but it does seemingly force a focus on group-level statistics in the analyses, reducing the statistical power of the study. The authors should acknowledge this limitation of their design, or provide justification for this choice if available.

Response: We thank the reviewer for bringing up these points apologize for having not being clear enough. We agree with the reviewer that it likely has advantages to not show feedback during the transfer phase. Our main motivation was the constraint to be coherent with previous experiments to increase comparability. We now clarify the procedure in the revised Methods under “**Transfer phase**” (p. 30). In brief, in probabilistic tasks, to be consistent with the design of Klein et al. (2017), we delivered outcome feedback in the transfer phase. Nevertheless, we also designed the probabilistic tasks in a way that there was equal feedback within the first cross-context comparison between the best stimuli to minimize differential value-updating, i.e. reward was delivered to both options. All subsequent transfer trials (starting with the second) followed the same pseudo-random schedule used during learning. Full feedback in the transfer phase was shown in tasks that had full feedback during learning; partial-feedback tasks continued to reveal only the chosen option's outcome. In Gaussian tasks, and in line with Bavard and Palminteri (2023), no feedback was given at transfer; here the design decisions followed Bavard & Palminteri (2023).

Reviewer #3: *3. In the model comparison, were the normalisation models adjusting by all rewards seen in that context across trials, or specifically the rewards seen within that trial? Given that these models weren't applied to the probabilistic tasks, I'm assuming the former (as normalisation of binary rewards makes little difference), but I think this might differ for Divisive Normalisation if operating at the trial level: if both options are rewarded in a trial, then their individual relative rewards could be reduced (i.e., 0.5 rather than 1), leading to a lower overall subjective value – and since this joint outcome is more common in the low contrast context, it will affect option A more than option C, potentially predicting the observed preferences. This may not be in line with previous uses of this model, of course, and may not account for all observed effects, but seems worth considering. In any case, I would appreciate clarification on the normalisation procedure.*

Response: We are happy to clarify this important point: The normalization models were adjusted by all rewards seen within a specific trial. We now explicitly state this when we report on the normalization procedure in the methods section, see changes on p40. We tested the differences due to different normalization on task p3 and agree that the low contrast context (because of the reasons the reviewer noted) would be more affected by the use of divisive normalization and that doing so would predict preference for the best HC context stimulus C during transfer. However, this is what the REL model does (and only the REP model does not) so we believe that qualitatively there would not be a difference for task p3.

Reviewer #3: 4. I appreciate the use of hierarchical Bayesian modelling, but I am somewhat surprised that the paper only reports group-level fits – surely a benefit of this method is to produce individual-level results? For example, does the performance of the REP model hold across all participants, or are there subsets using alternate methods?

Response: Yes, this is an important point. We now also report participant level fits from the hierarchical model and provide an overview of how many participants were best fit by any of the applied models (see Supplemental on Participant level model diagnostics and Supplemental Fig. 2 and Supplemental Fig. 3).

“Participant level model diagnostics

For each hierarchical fit we extracted the stored point-wise log-likelihood matrix, sliced the appropriate columns for every participant’s trials, and computed the participant level WAIC (Vehtari et al., 2017). We then (1) summarized the participant distribution (first column in Supplemental Figure 2 and 3), (2) reported the percentage of participants best-fit by each model (second column in Fig. 2 and 3), and (3) repeated the classical group comparison with Δ WAIC (third column in Supplemental Figure 2 and 3). The results corroborate the group-level DIC pattern: the repetition-bias (REP) model yields the lowest WAIC in most participants across most tasks and remains the global optimum at the group level except for tasks p4.2 and g2. While the advantage of the REP model is expressed at the individual level one can also see that in a subset of tasks the REL model also performs well at explaining participant behavior. However, we explicitly tested this in our task design and also show using simulations that the REL model digresses from its own predictions (see Supplemental Figure 6). Note that when computing the participant level WAIC from group level fits, these values are obtained after shrinkage, i.e. borrowing information through the group level parameters. However, we used uninformative priors on the group level parameters, and any possible shrinkage affects all models equally, so differences in WAIC per participant should remain meaningful (Vehtari et al. 2017).”

Reviewer #3: 5. While the REP model does perform best in most of the tasks, it is interesting that in some cases other models offer better fits despite being theoretically unable to predict the observed empirical effects. The authors do provide some explanation for this in the Discussion, but I’m left wondering how often the models did actually follow their theoretical predictions: it seems at least the REL model deviated from this in task p4.2, but was this a common finding? It would be helpful to explore the behaviour of the models in more detail to verify they are acting as expected (even if only in the supplemental), for example by illustrating their transfer preferences in a similar manner to Figure 2. More broadly, if theoretical predictions can indeed be violated by differences in learning mechanism or task design, the authors should be more cautious in falling back on these theoretical predictions when interpreting their analyses (e.g., ‘Note that standard analyses speak against the REL model’, page 16), and in fact should place more weight on simulation results as concrete demonstrations of model behaviour.

Response: We thank the reviewer for making this point. From our point of view, it is very well the case that all models have some flexibility in fitting the data which can be problematic when interpreting the results and this applies to all previously published studies in the field. In most of our tasks this is not critical, because we examine *quantitative* differences in fit: all candidate models can account for the direction of the preference bias, and our comparisons focus on how well they do so. Some key exceptions are tasks p3, p4.1, and p4.2, where the REL model can diverge from the correct learning contrast to explain transfer preference. This can happen either through the decay parameter, which differentially weights early versus late observations, or (as in prior studies) when using separate learning rates for chosen and unchosen options. We now do report model simulations for these tasks in the Supplementary Information of the paper, as follows:

“Model simulations

Given that for most tasks the majority of models do predict biased preference in the same direction, we benchmarked forward predictions with reasonable parameters against fitted latent trajectories (simulated from the fitted parameter estimates) for the three newly collected probabilistic tasks in which the relative-value (REL) and repetition (REP) accounts diverge most strongly (p3, p4.1 and p4.2). Concretely, for every candidate mechanism (ABS, REL, REP and IER) we first simulated 50 synthetic participants with reasonable parameters (decay-rate of 0.8, baseline learning rate of 1, softmax β of 5 and η repetition of 0.4). Each synthetic participant’s choices were then propagated through the model’s own update rules to yield trial-by-trial latent values, plain Q-values for ABS, composite C-values for REP (Q + choice/repetition values) and Q-differences for REL, which are plotted in the first, second and fourth columns of Supplemental Figure 8. Next, we re-fitted the REP and REL models to the behavioural data and simultaneously simulated latent Q- and choice values from the posterior parameter estimates, inferred from the data, for every trial. These fitted trajectories appear in the third and fifth columns of Supplemental Figure 8. It is insightful to visually compare the models’ unconstrained forward predictions with these fits to provide a test of qualitative adequacy. Across all three tasks only the REP model simultaneously recovers the learning-phase dynamics that can explain the observed transfer preferences. In contrast, the REL model predicts indifference and can account for the observed preferences only post-hoc by learning context-specific gaps that contradict its own theoretical prediction (see Supplemental Figure 8).”

Minor:

Reviewer #3: 6. A minor question on the order of the training trials: the text details the set-up within a context, but how were the two contexts arranged? Were these blocked or interleaved? If blocked, was there a separation between the contexts like that between training and transfer?

Response: The two contexts were fully interleaved following a pseudorandomized sequence ensuring equal exposure over the whole learning (training) phase and preventing more than two consecutive trials from the same context further we made sure that within in the last trials of the learning to context was shown twice. We have added a clarifying sentence to Methods on p29:

“In the learning phase, presentation of contexts was fully interleaved following a pseudorandomized sequence.”

Reviewer #3: 7. Figure 1 suggests the Gaussian tasks included a visual illustration of context as coloured backgrounds, but I see no mention of this in the text – is this correct? And if so, why was this added for these tasks if this wasn't used in the probabilistic conditions? Is there something about the Gaussian tasks that necessitated this, or was this simply a progression of the design?

Response: Yes, the Gaussian tasks (g1 & g2) featured context cues (background colour plus tree images). From our perspective nothing in the procedure required these as prior tasks have not implemented such cues. Our intuition was that it might make the data less noisy, while we lack direct evidence that the cues indeed lowered noise. This was also the reason we implemented the waiting period between learning and transfer, which prior studies had not. We now report the intention for our design decision in the methods section on p31 where we write:

“During learning and transfer both contexts yielded different background colors and unique tree icons in the corners. This was an exploratory design decision with the intention to boost participants awareness for the different contexts and cross-context decision during the transfer-phase.”

Reviewer #3: 8. The assumption that options in the p4 conditions have equal relative value is dependent on the use of range normalisation – if normalising by their sum, this equality no longer holds.

Response: Yes, this is correct. In tasks p4.1 and p4.2 we defined relative value as the centered value with respect to the contexts mean, which, unlike the normalizing by sum yields identical relative values of +0.10 for A_{LG} (0.60) and C_{HG} (0.76). We have clarified this in the methods section of the paper on p29:

“For example, according to state-dependent normalization (mean centering), participants should show indifference between these options. Note that this equality in relative value would not hold if we would normalize by the sum.”

Reviewer #3: 9. I had some confusion with the relative choice frequencies used in the correlational analyses: first, following on from my previous comment on the set-up of the transfer phase, I'm unclear exactly which options and comparisons this was calculated for. This particularly applies to the correlation between frequencies and valuations which is stated to use normalised scores 'relative to all other options encountered in the task' (page 29) – so is this across both training and transfer trials? Second, the switching between individual and group-level scores could be better explained: I'm assuming these were calculated at the group level when assessing choices in the transfer phase given the low trial count per participant (i.e., only the first 2), but if not, please clarify the rationale for this.

Response: We agree that the rationale must be clarified. The relative frequency scores for the correlations between frequencies and valuations were computed on the individual participant level because these were computed either across the whole learning (training) phase or across both phases (we included both approaches in the main manuscript). Because of this and the fact that valuation ratings were on a continuous scale (instead of 0 or 1 as in case of the first transfer choice) we had enough datapoints for correlational analysis on the individual level. We then correlated the difference values of these normalized frequency (difference in normalized frequency for stimulus A vs C) scores with the differences in their subjective valuations. We now updated our standard analysis section on p33 for clarity.

Frequency and valuation

For the correlation analysis between normalized choice frequency during learning or learning and transfer and valuation ratings we determined the relative choice frequency for each option at the individual participant level. For each option, we calculated a normalized frequency score, representing how often it was chosen relative to all other options encountered during learning (e.g. for tasks p1: how often did a participant choose option A in relation to options B, C and D). We then determined the differences in these normalized choice frequencies between pairs of options and correlated these differences with the differences in their individual valuation ratings. To control for reward effects, we focused on options with equal expected values (absolute and relative), as described previously. To analyze the relationship between choice frequency and uncertainty, we calculated the relative choice frequency for each option and the standard deviation of valuation ratings at the group level. We then tested for a linear relationship between relative choice frequency and the standard deviation of the value distribution. We further examined a correlation of relative

choice frequency and absolute distance from the true reward probability on the participant level (probabilistic tasks) or reward magnitude (Gaussian tasks).

Reviewer #3: 10. The results of the binomial tests are difficult to pull from the text – I would advise putting these in a table for readability.

Response: We thank the reviewer for pointing that out and now show an overview of all binomial tests in Table 1 on p8.

Table 1 Results of two-sided binomial tests on the first transfer trial. For each task we test whether the proportion of participants choosing stimulus A over stimulus C differs from chance ($H_0: p = 0.5$; the two stimuli have identical expected value). A p-value < 0.05 indicates a significant group-level bias in favour of one stimulus.

Task	Comparison	Statistics
p1(p1.1-p.1.3)	A _{LC} vs C _{HC}	CI[0.63-0.82]; $p < 0.0001$; n = 94
p2	A _{LC} vs C _{HC}	CI[0.48-0.77]; $p = 0.08$; n = 49
p3	A _{LC} vs C _{HC}	CI[0.16-0.43]; $p = 0.004$; n = 49
p4.1	A _{LG} vs C _{HG}	CI[0.31-0.6]; $p = 1$; n = 30
p4.2	A _{LG} vs C _{HG}	CI[0.64-0.94]; $p < 0.001$; n = 29
g1	A _{LC} vs D _{HC}	CI[0.59-0.85]; $p < 0.001$; n = 49
g2	A _{LC} vs D _{HC}	CI[0.60- 0.86]; $p < 0.001$; n = 48

Reviewer #3: 11. There is little to no discussion on comparisons of the worse options from each context (i.e., B vs. D), despite being shown for all conditions in the bottom row of Figure 2 – I realise comparisons of the better options are the key contrast, but is there anything to draw from these? If not, the authors may wish to remove these since this is already a crowded figure.

Response: True, we agree that comparison and further investigation of these worse options might yields valuable information. Given prior research and the already extensive model comparison across multiple datasets our intention was to keep the focus on the main effects involving the high value stimuli. Most results shown in Figure 2 follow what one would assume given that participants correctly learn the values of options. However, there are two exceptions to this, which are the first comparison in the partial feedback task p2 and the first comparison in p4.1 where more

participants chose option B than option D, even though option D has a higher absolute value. We believe that the exact mechanism of these findings might be related to a combination of repetition and value uncertainty. While we do not explore these effects further in the current manuscript to maintain a clear focus on our main research questions, we would like to keep these results in the figure to potentially inspire future work by others.

Reviewer #3: *12. I would appreciate some clarification on the number of data points going in to each correlation between training and transfer choice rates: since these measures are stated to be calculated at the group level, I assume this uses 2 points per task (or 1 when restricted to equal expected values), hence the need to pool data across tasks – is this accurate? If so, this presents another instance where restrictions on the number of training trials able to be considered cut into the statistical power of the study, preventing within-task correlations and so any comparisons of these correlations according to factors like partial feedback.*

Response: We agree, some clarification is needed. Given that we had 30 learning trials per context but only one or two *cross-context* comparisons with equal absolute or relative value, per-participant correlations would have been based on just one data point and is therefore uninterpretable. For example, in task p1, we could only compare stimulus ALC(0.7) vs CHC(0.7) (relative choice frequency during learning vs. relative preference during transfer). Thus, we pooled data across all tasks and computed correlations on the group level. We fully understand that this is not optimal but results show a consistent picture across different tasks. We now also complement the pooled correlations with hierarchical, logistic regressions that include subject-level random intercepts and slopes. These regressions confirm that choice frequency during learning, by itself, predicts (1) transfer choices for option pairs matched in absolute or relative value and (2) participants' explicit value ratings (see p11 in the manuscript). Collectively, the correlation and hierarchical regressions and computational modeling provide converging evidence for a robust frequency effect.

Reviewer #3: *13. Why is Task p1 not included in Figure 4? Did this not request valuations? I see no mention of this in the Methods.*

Response: For task p1 we did not request valuation data, we have stated this in the legend of Figure 4 and now also state this in the methods section on p27.

Reviewer #3: *14. When introducing the REP model, it is initially unclear how the RL and repetition bias components are combined until the 'Action Selection' section, where they are shown to be additive – I would suggest clarifying this earlier, and removing Equation 20.*

Response: We thank the reviewer for pointing this out. We now present Eq. 20 earlier as Eq.3 on p17 in the manuscript where we also introduce the equations for the repetition bias.

Reviewer #3: 15. *The models are stated to deal with partial feedback by retrieving the most recent observation for an option when actual outcomes are unknown, but what about the first trial? Is there some prior value here?*

Response: We thank the reviewer for pointing this missing information out. We now explicitly write this in the methods section on modeling partial feedback on p44. In brief, we set initial Q-values to 0.5 and initial unknown rewards to 0.1 in probabilistic tasks and draw from uniform distribution for Gaussian tasks.

Reviewer #3: 16. *Since the key measure for model fit is difference in DIC, I would suggest adjusting the plots in Figure 5 to show this on the y-axes, or at least use separate axes between tasks so that smaller differences are not obscured by being plotted next to larger absolute scores (as in the g1 panel).*

Response: We thank the reviewer and changed the plots in Fig. 5 accordingly. We now do report WAIC and DIC difference in this plot (see Figure 5).

Reviewer #3:

17. *Typos*

Page 2, Significance statement: 'repeating decisions can shapes distort'

Page 3, paragraph 1: 'first phase of such experiment's participants learn...'

Page 17, Figure 5 caption: 'Vertical red lines' should be horizontal

Response: We thank the reviewer for pointing out these typos and corrected them accordingly.

Response to the Editor and Reviewers for Wagner et al., “Explaining decision biases through context-dependent repetition” in *Communications Psychology*

Manuscript ID: COMMSPSYCHOL-25-0091-A

General remarks:

We thank the Editor and the two Reviewers for their further evaluation of our revised manuscript and for their helpful suggestions. We sincerely thank the two reviewers for their constructive feedback, and we have carefully responded to all comments and incorporated revisions, as detailed below.

In addition, and for clarification we also made some slight changes to the wording in our manuscript, especially about the terms ‘experienced’ or ‘objective value’, where we highlighted those changes in the main text. Further, for transparency, and for journal requirements we added a general limitations section where we further discuss limitations highlighted by the Reviewers, i.e. the power of our correlation- and regression analysis and model confusion.

Limitations (p27 & 28)

Several limitations need to be addressed: First, all tasks were conducted in stationary environments with stable reward contingencies. Repetition effects may differ under conditions of environmental volatility, i.e. one could imagine that repetition may exert weaker effects and repeating actions is less pronounced as environmental volatility increases. Moreover, although our results replicate across datasets, all tasks share a relatively similar structure with binary or trinary choices and fixed learning phases. Future studies should examine how such biases interact with changing reward structures and more complex environments (see Legler et al. 2025). Importantly, some analyses, particularly cross-task correlations for frequency preference relationships, are based on a small number of datapoints (i.e., pairwise decisions were limited per task) and were therefore computed on the group level. While we supplemented these analyses with hierarchical mixed-effects regressions that incorporate participant-level variability and replicate the key effects, these models still rely partly on data pooled across tasks, which may limit generalizability at the individual level. Further, in this analysis we isolate the statistical relationship between frequency and preference and do not control for other effects that might cause distortions in value representations and preference (see additional analyses where we control for estimated Q-values in the Supplement). Finally, while our model recovery analyses confirms that key models such as REP and REL but

also IER, DIV and RANGEo are distinguishable in key tasks, recoverability was lower for certain task-model pairs. This highlights an important challenge: model discriminability is strongly shaped by task design (i.e. reward magnitudes, the amount of learning and transfer trials or cross-context comparisons). Because our tasks were largely inherited from prior studies and included reanalysis of existing datasets, optimizing for model separability was not prioritized. Future work would benefit from explicitly testing model identifiability during task development to ensure that competing accounts yield sufficiently divergent predictions within the chosen design.

Furthermore, to support our findings, we added a new Figure 11 to the Supplemental to directly show transfer predictions for our simulations shown in Supplemental Figure 10. The main finding is that only the REP model consistently captures the direction of experimentally observed transfer preferences across those tasks.

Supplemental Figure 11 Transfer phase predictions. Panels A–C show simulated choice proportions from each model (ABS, IER, REL, REP) alongside the observed data (DATA) for A: task p3, B: task p4.1, and C: task p4.2. Each dot represents the proportion of choices favoring one option over another (A vs C in red; B vs D in blue), computed across the full group of participants for one of 15 full-dataset simulations per model. The dashed line at 0.5 indicates no difference. Only the REP model consistently captures the direction and magnitude of observed transfer preferences across those tasks.

We added new funding information on p49:

BJWs position at University Hospital Tübingen is funded by an Alexander von Humboldt Professorship awarded to Peter Dayan.

Reviewer #1:

1. The answer to my previous major point 1 and point 2, regarding experienced value being a confound for repetitions, is not satisfactory. The authors' answer (and corresponding edits) hinge on the "full feedback" condition, arguing that this ensures that the experimenter defined expected value is the true experienced value. However, a large literature shows that participants do not integrate feedback from unselected options in the same way that they do for selected options (see e.g. Cockburn et al 2016, but also all the models from the papers cited that include different chosen and unchosen learning rates). As such, while full feedback helps to some degree, it simply does not guarantee that the experienced value is the true expected value. This is an assumption that the authors are making, and it should be a) made explicit; b) tested (which can easily be done via within participant, single trial logistic regression analyses). As it is likely not fully valid, it should then be controlled for. I appreciate the new mixed effect modeling which shows consistent but weaker findings, as expected; however this model should also include relevant covariates such as experienced absolute value or relative value, rather than limiting to specific subsets of the data. Should the repetition effect survive such corrections, then the statistical conclusions will be valid.

Response:

We thank the reviewer for the thoughtful comment and appreciate the opportunity to clarify our approach and provide further analyses. We agree with the reviewer that the paper requires some checks whether the transfer could be explained by an effect due to differences in experienced values. To do this, we directly tested this by fitting asymmetric- α (chosen/unchosen) RL models (ABS, REL) to the learning phase of 9 representative datasets (all of our newly collected probabilistic and Gaussian datasets, and the data from Klein et al. 2017). These datasets included key decisions with equal absolute and equal relative values. To check whether the resulting experienced values after conclusion of the learning phase explains choices in the transfer phase, we extracted end-of-learning Q-values as experienced value estimates and show their distribution across participants. Crucially, we included ΔQ alongside relative frequency in hierarchical logistic regressions predicting transfer choices. If differences in experienced value drove preferences during the early transfer phase, one would expect their correlation. The key result is that we did not find such correlations: ΔQ was not significantly correlated with the relative frequency of choices during learning, in all these tasks. Further the estimated slope coefficient for ΔQ perfectly

overlaps with 0. We believe that these checks on representative parts of the data provide compelling evidence against experienced value as a driving force of the observed transfer effects.

Our main inference relies on formal model comparison, which directly disambiguates between mechanistic accounts (asymmetric learning, normalization, repetition) by fitting latent learning and transfer dynamics. We view this as more diagnostic than adding covariates in a logistic regression, which collapses learning histories into summary scores and assumes linear additive effects. We also emphasize that formal RL model comparison is not only more mechanistic, but also necessary to relate our findings to the existing literature, since most competing accounts (e.g., asymmetric learning, normalization) were originally proposed in this standard modelling framework. These RL models make nonlinear relations explicit and can be compared directly. Importantly, the additional analyses above (showing that experienced value differences (ΔQ) does not explain variance for preference in the transfer phase) corroborates one of the modelling results of the paper: that the ABS model fails to account for transfer behaviour. We report these additional control analyses in the Supplement, where we write:

Control analyses for relative frequency during learning and transfer preference

To test whether learned values (at the end of the learning phase), in the form of absolute or relative Q-values, that is their differences can account for transfer preference or explain away the frequency effect, we conducted the following control analyses: We first refitted our hierarchical ABS and REL RL models for each newly collected dataset (tasks p1, p2, p3, p4.1, p4.2, g1, g2) and the two datasets from Klein et al. (2017). Specifically, we used a model with different learning rates for chosen and unchosen options (ABS model) or different learning rates for each context (REL model). We then directly simulated Q-values from the joint posterior distribution at the end of the learning phase and (i) show these in side-by-side plots for both options with equal absolute values, i.e., option C in the high-contrast (HC) context and option A in the low-contrast (LC) context (tasks p1, p2, p3, g1, g2, and Exp 2 and Exp 3 from Klein et al., 2017) and for both options with equal relative values - option C in the high gain (HG) context and option A in the low gain (LG) context. Plotting these values shows that our task design (balanced and controlled reward schedule) worked as intended. Q-values at the end of learning show only minor variability (Supplemental Figures 15A, 16A, and 17A). Next, we show associations between Q-value difference and relative

choice frequency (Supplemental Figures 15B,C; 16B,C; and 17B,C). If the association between relative choice frequency and transfer preference stemmed from differences in learned values, there should be a positive association between Q-value differences (option C [D] minus option A) and relative choice frequency (option C [D] versus option A). However, such an association is mostly absent. The only task showing a positive association is task g2 ($t(96) = 3.5987$, $p = 0.001$), driven by an extreme outlier (> 3.5 SD); if this data point is removed ($t(95) = 1.348$, $p = 0.18$), all correlations are non-significant. Likewise, recomputing both hierarchical regressions for equal absolute and equal relative comparisons does not change the results. Please note that these control analyses are not exhaustive and are not intended to be. We do not examine all tested associations (comparisons, datasets, e.g., both datasets from Bavard et al., 2018 and 2023) and we do not use other forms of normalized values to control for effects on transfer preference. Therefore, this analysis should only be interpreted for what it is: evidence of an association with choice frequency, not an exclusive mechanistic explanation.

Supplemental Figure 15 **A**: Posterior-median absolute Q-values for all equal absolute value pairs across our new probabilistic tasks. Option A (0.7 LC) vs. option C (0.7 HC), shown separately for each task (Task p1, Task p2, Task p3, Klein et al. 2017 (Exp 2), Klein et al. 2017 (Exp 3)). Points are absolute Q-values at the end of the learning phase for individual participants estimated from the ABS RL model with different learning rates for chosen and unchosen options. **B**: Relationship between the ABS Q-value difference (x-axis; option C – option A) and the relative choice frequency of choosing C (in contrast to A) during learning (y-axis). The solid line is a linear fit; dashed lines mark the reference values where the difference in Q-values is 0 and relative frequency is 0.5 **C**: Same relationship as in panel **B** but split by task (points colored by task) with separate linear fits per task. We do not find any positive association between relative choice frequency and Q-value difference.

Supplemental Figure 16 **A**: Posterior-median absolute Q-values for all equal absolute value pairs across our new Gaussian tasks. Option A (LC) vs option D (HC), shown separately for Task g1 and Task g2. Points are absolute Q-values at the end of the learning phase for individual participants estimated from the ABS RL model with different learning rates for chosen and unchosen options. **B**: Relationship between the ABS Q-value difference (x-axis; option D – option A) and the relative choice frequency of choosing D (in contrast to A) during learning (y-axis). The solid line is a linear fit; dashed lines mark the reference values where the difference in Q-values is 0 and relative frequency is 0.5 **C**: Same relationship as in panel **B** but split by task (points colored by task) with separate linear fits per task. There is a positive association of relative choice frequency and Q-value difference in task g2. However, this is driven by one extreme outlier (> 4 SD; upper right) and n.s. when this datapoint is excluded.

Supplemental Figure 17 **A**: Posterior-median relative Q-values for Context 1 and Context 2, shown separately for Task p4.1 and Task p4.2. Points are relative Q-values at the end of the learning phase for individual participants estimated from the REL RL model with different learning rates for each context. **B**: Relationship between the REL Q-value difference (x-axis; Context 2 – Context 1) and the relative choice frequency of choosing C (in contrast to A) during learning (y-axis). The solid line is a linear fit; dashed lines mark the reference values where the difference in relative Q-values is 0 and relative frequency is 0.5 **C**: Same relationship as in panel **B** but split by task (points colored by task) with separate linear fits per task. There is no positive association of relative choice frequency and relative Q-value difference.

Frequency and preference (adjusted for Q)

Supplemental Figure 17 Hierarchical logistic regressions. We recomputed our hierarchical logistic regressions for all of our new datasets (and both datasets from Klein et al. 2017) to predict the first transfer choice between options C (HC context/ HG context in tasks p4.1 and p4.2) and option A (LC context / LG context in tasks p4.1 and p4.2) from Q-value difference (ΔQ ; $C(HC) - A(LC)$) and relative choice frequency (Slope) during learning (relative frequency for C in contrast to A), with random intercepts and random slopes for both predictors (ΔQ and relative choice frequency) by participant. Panels (columns) separate datasets into equal absolute value (probabilistic tasks [p1, p2, p3, Klein et al. 2017 exp2, Klein et al. 2017 exp3]; Gaussian tasks [g1, g2]) and equal relative value (probabilistic tasks p4.1 and p4.2). Points are posterior means and bars are 95% credible intervals; the vertical dashed line corresponds to 0. Red points indicate effects whose 95% CI excludes 0. As in our main analysis in the probabilistic, equal absolute value set, the frequency slope is positive and credible, predicting transfer preference after controlling for ΔQ . In the equal-relative value set, the frequency effect is numerically positive but not credible. In the Gaussian, equal absolute set, the frequency effect is likewise numerically positive but not significant. Across all analyses, ΔQ (HC - LC) estimates are centered near zero with wide CIs, indicating that adding

Q-value differences at the end of the learning phase estimated via ABS or REL models do not account for transfer preference.

In addition, we do directly mention assumptions for analysis in the results section and highlight limitations in the limitations section of the paper where we now write:

Results on p11 and p12:

Note that, by design, via controlled trial sequences and, in most tasks, full feedback, those matched pairs had equal absolute- and equal context-normalized values, yielding equal ground-truth expected values (EVs) for both options in each comparison. Therefore, those analyses above assume that participants' internal estimates tracked these EVs. To address potential confounds and subject-level variability, we present hierarchical regressions below and control analyses based on model-derived Q-values in the Supplement (Supplemental Figures. 15–17).

and *Limitations* on p27:

Importantly, some analyses, particularly cross-task correlations for frequency preference relationships, are based on a small number of datapoints (i.e., pairwise decisions which were limited per task) and were therefore computed on the group level. While we supplemented these analyses with hierarchical mixed-effects regressions that incorporate subject-level variability and replicate the key effects, these models still rely partly on data pooled across tasks, which may limit generalizability at the individual level. Further in this analysis we isolate the statistical relationship between frequency and preference and do not control for other effects that might cause distortions in value representations and preference (see additional analyses in the Supplement).

Reviewer #1:

2. All three reviewers pointed in some way to the fact that the number of points entering in the correlations was unclear/low (my previous point 5), but the authors have not corrected the paper accordingly. The reviewers' confusion is largely because it is so unusual to do correlations across experiments, as is the core here in figure 3 and 5GH. Again, this would be much better tested within participants or at least hierarchically, using multiple regression approaches, better controlling for other covariates that might influence the findings. At the very minimum, the main text should make it abundantly clear how underpowered those are. For example, the result of Fig. 5H: $\rho=-.82$ reads impressive to an inattentive reader in the context of "15 tasks and 300's participants", but $\rho(11)=-.82$ puts this better in context by defining the degrees of freedom. Please include explicitly all degrees of freedom in the correlation stats if those correlations remain in the revised version.

Response: We thank the reviewer for raising this point again and agree that misleading or underpowered correlations should be made transparent. Please note that we have already substantially revised the manuscript in response to the earlier concerns. In detail we added complementary hierarchical logistic regression analyses that assess within-subject variability and are more powered and statistically robust. These models do account for participant-level random effects and confirm that the observed frequency effects are not merely a by-product of low-N correlations. In this way, the regression models directly address the reviewer's suggestion to test the frequency-preference link using within-subject or hierarchical approaches. Please note that we did choose hierarchical regressions with subject level random effects over participant level correlations because across tasks there were only one or two transfer choices per participant, which would be too low for participant level correlations. Further we have clarified the number of points entering the correlations across experiments in the methods section as outlined in our previous response. We also do now report 95% credible intervals for each correlation analysis thus every reader can directly see the uncertainty around correlations coefficients. We thought these are more informative to the average reader. We now also report the low number of datapoints in some correlations as a clear limitation in the discussion on p27.

Minor and Typos:

- P3: participant's  participants'

- P14: $r=-.43$  $.43$

- I remain curious how the findings relate to similar recent findings (previous citations[1,2]).

Response: We thank the reviewer for their attention to pointing these out and corrected those typos accordingly. Further we apologize that we overlooked those valuable studies. We now discuss how our findings align with those studies, in the discussion on p24:

Our results further align with two recent preprints. Eckstein et al. (2023) show that hybrid ANN-RL models can uncover both classical reward-based learning and additional reward-independent mechanisms (e.g., choice kernels). They find that action repetition effects captured by choice kernels improve model fits and conclude that non-value-based repetition biases must be modeled to better capture human choice tendencies. Likewise, Collins (2023) shows that humans often rely on non-RL mechanisms, especially in structured environments and find that reward blind associative processes, i.e. repeating actions are inherent in human learning.

Reviewer #3 (Remarks to the Author):

The authors have done well in answering my comments from the previous draft, adding greater elaboration on their methodology and analyses, as well as additional tests seeking to capture participant-level effects in both the empirical data and the modelling. While some of my previous criticisms on the design of these experiments (specifically the use of feedback in transfer trials) still stand, I understand now that these were inherited from previous work and retained to support aggregation across tasks. This being said, I do still have some issues with the manuscript, particularly with the new model recovery exercise, which are detailed below. As such, while my impressions of the paper are still largely positive, I suggest another revision to remedy these issues before publication.

Response:

We are glad that the additional methodological clarifications and new analyses have been helpful, and we appreciate the opportunity to further clarify our results and address the remaining points.

Issues

- While I appreciate the addition of a model recovery exercise, I'm afraid the present version is not sufficient to fully support the authors' conclusions: based on the Supplemental materials, simulated data is seemingly only generated for the REP model, so while this model may be

recoverable, it remains uncertain whether the REP model also captures the behaviour of other generating models. The proper test would be to simulate and fit data from all models in a full NxN matrix – this is of course a more intensive procedure so I understand some hesitancy, but this is necessary to properly test the discriminability of the models. This is not to say I expect the REP model to do this, but without this complete comparison, this possibility cannot be eliminated.

Response: We thank the reviewer for this important point and fully agree that a complete recovery matrix (simulating and fitting all models against all others) represents the gold standard for assessing model discriminability.

As the reviewer noted, a full NxN recovery is computationally demanding. To balance rigor with feasibility, we now provide complete model recovery matrices (confusion plots) for four representative tasks: p3, p41 (probabilistic tasks, diverging model predictions), g1 (Gaussian task) and Exp2 from Bavard et al. 2018 (task with gains and losses and higher overall complexity) that span the experimental space we cover. These analyses are presented in the Supplement. We believe this extended recovery exercise substantially strengthens our study. The new results are reported in the Supplement and briefly described in the main text on p22 and in the methods section on p48.

Model Recovery Confusion Matrix – WAIC (full)

Supplemental Figure 2 Confusion matrix for task p3. Results show a full model recovery analysis using WAIC scores for 20 simulated full datasets per model. Each cell indicates how often a dataset generated by a given model (rows: “True Generating Model”) was best fit by one of the candidate models (columns: “Winning Model”). Both REP and REL models are correctly identified in 100% of cases, suggesting they are distinguishable from competing models given the task structure and number of trials. In contrast, the ABS and IER models show confusion. Data generated by ABS is often misattributed to REL (45%) or IER (10%), while IER simulations are frequently misclassified as REL (35%) or ABS (15%). This pattern suggests that, in this probabilistic task, REP and REL models are identifiable, while ABS and IER may produce qualitatively similar behavior. These results are consistent with our theoretical expectations, as ABS and IER make similar predictions in this probabilistic task, and REL often fits the data well despite violating its own theoretical predictions (see Simulations below).

Model Recovery Confusion Matrix – WAIC (full)

Supplemental Figure 3 Confusion matrix for task p4.1. Results show a full model recovery analysis using WAIC scores for 20 simulated full datasets (*i.e.*, all participants and trials) per model. REP and REL models are again identifiable, each correctly recovered in 100% of simulations. In contrast, both ABS and IER models are consistently misclassified as REL. This likely reflects two factors: First, ABS and IER make highly similar predictions under this task dynamics. Second, the REL model is comparatively simpler and can accommodate the behavior generated by ABS and IER with fewer parameters (see simulations below), leading to higher WAIC. These results suggest that while REP and REL produce clearly distinguishable behavioral patterns in task p4.1, the ABS and IER models lack uniquely identifiable signatures, either with each other or with the more parsimonious REL model.

Model Recovery Confusion Matrix – WAIC (full)

Supplemental Figure 4 Confusion matrix for task g1. Results show a full model recovery analysis using WAIC scores for 20 simulated full datasets (i.e., all participants and trials) per model for ABS, REP, DIV, RANGE, IER, and RANGEo models. All models except RANGE and ABS show high identifiability, with 100% correct recovery. RANGE was recovered in 75% of cases, with the remaining 25% misclassified as RANGEo, a more complex variant. This partial confusability highlights the similarity of predictions between RANGE and RANGEo. Further 15% of ABS datasets were better fit by the REP model, which is likewise a more complex version of the ABS model. Results show that most models are identifiable in this task given the number of trials per learning and transfer phase.

Model Recovery Confusion Matrix – WAIC (full)

Supplemental Figure 5 Confusion matrix for *Exp2* from *Bavard et al. 2018*. Results show a full model recovery analysis using WAIC scores for 20 simulated full datasets (i.e., all participants and trials) per model for ABS, REP, DIV, RANGE, IER, and RANGEo models. All models except RANGE and ABS show high identifiability, with 100% correct recovery. RANGE was recovered in 65% of cases, with the remaining 35% misclassified as RANGEo, a more complex variant. This partial confusability highlights the similarity of predictions between RANGE and RANGEo. Further 90% of ABS datasets were better fit by the REP model, which is likewise a more complex version of the ABS model. Results show that most models are identifiable in this task, though the ABS model and RANGE variants remain more difficult to distinguish given task dynamics.

We further discuss these findings and their limitations in the Discussion section on p28 where we now write:

Finally, while our model recovery analyses confirms that key models such as REP and REL but also IER, DIV and RANGEo are distinguishable in key tasks, recoverability was lower for certain task-model pairs. This highlights an important challenge: model discriminability is strongly shaped by task design (i.e. reward magnitudes, the amount of learning and transfer trials or cross-context comparisons). Because our tasks were largely inherited from prior studies and included reanalysis of existing datasets, optimizing for model separability was not prioritized. Future work would benefit from explicitly testing model identifiability during task development to ensure that competing accounts yield sufficiently divergent predictions within the chosen design.

Reviewer #3:

- I also did not follow the procedure of the recovery analysis: at first, it seemed this only used 10 simulations, which seemed low, but from the Methods it seems this may have been 10 subsets (of unspecified size) out of 10,000 total simulations, though I'm still not certain. Adding to the confusion, Supplemental Figure 6A seems to suggest the number of simulations was 15 (according to the x-axis), which disagrees with all the other numbers. Part of the issue here may be the ambiguous use of the term 'dataset', which could refer to data from a single subject, data from multiple subjects or a whole task. More clarity on the recovery procedure is needed.

Response: Yes, we agree that the recovery section was partially unclear and thank the reviewer for pointing this out. The term “subset” was used ambiguously, and we have now revised the manuscript and supplemental materials to ensure clarity. To clarify: in our model recovery analysis, we simulated 10,000 total datasets and re-fitted the entire dataset comprising all participants for each corresponding task 10 times. That is, each of those simulations represents a full simulation of all participants for that task under the model being tested, followed by full re-fitting of all participants. We have now clarified this in both the methods section, supplement and the Supplemental Figure 4 and 5 captions.

Reviewer #3:

- Following on from this, if model recovery was indeed split into subsets, I don't understand the purpose of this: why fit 10 separate sets rather than one total set? My speculation is that this is intended as a compromise between aggregate and individual-level fits, giving a single best-fitting model in each subset but an overall recovery rate across subsets - this may be what is meant when referring to variations in best fit preventing perfect recovery on page 13 of the Supplemental. If this is the case, I must disagree with this procedure: the purpose of model recovery is to measure how often two models are confused precisely because of similarities in predicted behaviour to allow interpretation of model fitting. This then casts doubt on the perfect recovery reported in the main text, which may be more divided than this suggests. I advise the authors to consider their recovery results in more detail, and potentially add more nuance to their conclusions depending on their findings.

Response: We thank the reviewer for this thoughtful follow-up and clarify that our model recovery procedure was not split into subsets. We had used the term “subset” to describe the 10 random samples (full datasets) drawn from the 10,000 simulated datasets. To avoid confusion, we have revised the terminology throughout and now use clearer wording in the manuscript.

Reviewer #3:

- The addition of the mixed-effects regressions is very welcome to make better use of the data, though I believe these are still restricted to one or two points per person so some of the previous criticisms on statistical power may still apply (though admittedly reduced). As a minor comment here: is there a particular reason to include the intercept estimates? From my understanding, the slopes are the key test of the frequency effect, while intercepts presumably reflect a general bias towards/away from the target choice (there is a suggestion at the end of the caption for Figure 4 that both slope and intercept contribute to the effect, though I do not see how). If so, the authors may wish to remove intercepts to focus on the key slope values, though if I'm mistaken, I would ask for some explanation in the text.

Response: We thank the reviewer for this thoughtful comment. We fully agree that the slope is the key parameter for testing whether relative frequency during learning predicts transfer choices. While our initial model included an intercept, we now also computed versions without the intercept term and reflect on the rationale for either approach below. First, we wish to clarify a technical point: in the earlier version of our regression, we predicted choices for the less frequently chosen stimulus (i.e., where choice = 0) using its corresponding frequency and misinterpreted the intercept. Because all analyses were based on pairwise comparisons, this inversion did not affect slope estimates, which are symmetric under reversal. Nonetheless, we now follow a more intuitive implementation: the predictor reflects the relative frequency of the more frequently chosen option, and the outcome is coded as 1 if that option is chosen in transfer. This revision improves interpretability and aligns with standard practices. The updated figures and Bayes Factors now reflect this implementation. Regarding the intercept term: we agree that when both stimuli in a pair are chosen equally often during learning (i.e., relative frequency = 0.5), the intercept captures the average tendency to prefer one over the other in transfer. In most equal-value comparisons, the intercepts are near zero, suggesting no consistent bias. However, in some datasets, particularly those with equal *relative* values but unequal *absolute* values the intercept can capture meaningful and systematic choice biases. For instance, in the Bavard et al. (2018) datasets, many transfer comparisons involve pairs with the same relative expected value but very different absolute magnitudes for these. In such cases, the intercept helps reveal a general preference for stimuli with higher overall reward, independent of frequency. Including the intercept thus allows us to use a consistent modeling approach across all datasets while preserving interpretability when the

distribution of comparison types varies. It also permits Bayes Factor comparisons against an intercept-only null model. Removing the intercept would change meaning of the slope in some cases and therefore bias results in datasets where other factors (e.g., magnitude effects) are present. For completeness, we also computed our regressions without intercepts (see below), using group-level and subject-level random slopes only. These yielded very similar results for the slope in most datasets. If the Reviewer wishes we are happy to include these in the Supplement of our study.

Figure 1 Mixed effects Bayesian logistic regressions to model the effect of choice frequency (Slope) during learning on preference in transfer (dependent variable). We here analyzed three pooled choice sets (“all pairs”, “equal absolute value”, “equal relative value”) across all new datasets and published studies to quantify whether preference in the first transfer trial is related to relative choice frequency during learning. Points are posterior means; red intervals show parameter estimates whose 95 % CI excludes 0 (i.e. credible frequency effects), grey intervals represent intervals overlapping with 0. The dashed vertical line at 0 aids visual inspection; Intervals for the Slope entirely to the right show a positive frequency/repetition preference relation.

Reviewer #3:

- While Bayes factors are calculated for the mixed-effects regressions, these are all relegated to the supplemental – it would be helpful to put these in the main text somewhere to quantify these effects rather than relying on solely Figure 4.

Response: That's a good point. We now also report directional Bayes factors for all models we show in Figure 4 in the main manuscript on **p12**.

Reviewer #3:

- I understand the WAIC measure was added in response to reviewer comments, but I would add that using two model criteria is unnecessary and distracting, particularly when their results seem so consistent – I suggest the authors pick one of these and use only that for model comparison.

Response: We agree and now only do report WAIC in the main manuscript.

Reviewer #3

- I'm grateful for the addition of participant-level model fits, though I'm disappointed this is only given in the Supplemental – some mention of these results in the main text would be beneficial to better reflect the individual variation in model performance.

Response: That's a valuable point. We now also report participant level fits the main manuscript. In detail we now report the percentage of participants best fit by each model in the results section of the main manuscript and further point the reader to the supplemental figures more prominently.

- A very minor question: what is the difference between Figure 4 and Supplemental Figure 1?

Response: With respect to Reviewer #1s comment that the frequency-preference effect might be distorted through “experienced value” we also computed the regressions while excluding both partial feedback task. In doing so we assured that at least all participants -given they were not distracted-, must have observed all rewards. Those results do not deviate in any meaningful way from results in Figure 4. We now also control for estimated Q-values in an additional analysis.

Minor:

Typos

Page 6, Figure 1 caption: 'For all new probabilistic tasks tested two cross-context...'

Page 12, Figure 3D caption: 'r = -.43'

Page 14, paragraph 1: 'r = -.43'

Supplemental, page 8: 'Supplemental Figure 3 and 4' should be 'Supplemental Figure 2 and 3'?

Response: We thank the reviewer for pointing these out and corrected them.

Response to the Editor and Reviewers for Wagner et al., “Explaining decision biases through context-dependent repetition” in Communications Psychology

Manuscript ID: COMMSPSYCHOL-25-0091-B

General remarks:

We very much thank the two Reviewers for their further evaluation and positive review of our revised manuscript and for their very helpful suggestions. Thank you very much for your feedback.

Reviewer #2:

1. Wagner, Wolf, and Kiebel have made extensive edits to the manuscript after the first reviews, which I appreciate greatly. As a disclaimer, I have to point out that I did not see the rebuttal and revisions of the first round of reviews and could not provide feedback on how the authors addressed my previous concerns yet. However, from what I see in the current version of the manuscript and supplemental information, I am satisfied with the authors' revisions and have only very minor comments I would like to draw their attention to.

Response: We very much thank the reviewer. We were initially confused (I had bad mood for some days) by the first post-revision comments because we believed we had at least addressed the substantial points raised by the reviewer in our first revision. Now that we know that our first rebuttal was not received, this fully explains the discrepancy. In any case, examining end-of-learning Q-value differences was still informative. We are pleased that the reviewer's concerns are now largely resolved and we address the remaining points below.

Reviewer #2: *In Results, Assessment of Preferences, you give p-values and credible intervals (CIs), but it's not the CIs of the p-values but of the probability value associated with the binomial test. All other CIs in the manuscript relate to the value/test statistic that was presented directly in front of the CI. Please specify once to which kind of variable the CIs in that section belong so as not to confuse your readers.*

Response: We thank the reviewer for pointing that out and agree that this is important. We now explicitly specify which kind of variable the CIs belong to. For example on p22 we write:

“To quantify those preferences we report the proportion \hat{p} choosing of choosing the target option (C_{HC} in probabilistic tasks; D_{HC} in Gaussian tasks) in the first transfer trial (TT) together with its 95% exact binomial confidence interval; p-values are from two-sided exact binomial tests of $H_0: p=0$ (Task p1(full feedback): $\hat{p}(C_{HC}) = 0.73$ [0.63-0.82]; $p < 0.0001$; Task p2(partial feedback): $\hat{p}(C_{HC}) = 0.63$ [0.48-0.77]; $p = 0.08$; Task g1(full feedback): $\hat{p}(D_{HC}) = 0.74$ [0.59-0.85]; $p < 0.001$; Task g2 (partial feedback): $\hat{p}(D_{HC}) = 0.74$ [0.60-0.86]; $p < 0.001$; see Figure 2H-I and M-N).”

Further we updated Table 1 for clarity:

Task	Comparison	\hat{p} (stimulus)	95% binomial CI	p (two-sided)	n
p1(p1.1-p1.3)	C_{HC} vs A_{LC}	0.73 (C_{HC})	[0.63-0.82]	$p < 0.0001$	n = 94
p2	C_{HC} vs A_{LC}	0.63 (C_{HC})	[0.48-0.77]	$p = 0.08$	n = 49
p3	C_{HC} vs A_{LC}	0.29 (C_{HC})	[0.16-0.43]	$p = 0.004$	n = 49
p4.1	C_{HG} vs A_{LG}	0.5 (C_{HG})	[0.31-0.6]	$p = 1$	n = 30
p4.2	C_{HG} vs A_{LG}	0.83 (C_{HG})	[0.64-0.94]	$p < 0.001$	n = 29
g1	D_{HC} vs A_{LC}	0.74 (D_{HC})	[0.59-0.85]	$p < 0.001$	n = 50
g2	D_{HC} vs A_{LC}	0.75 (D_{HC})	[0.60-0.86]	$p < 0.001$	n = 48

Table 1 Results of two-sided binomial tests on the first transfer trial. For each task we test whether the proportion of participants choosing stimulus C (tasks p1-p4.2) or stimulus D (tasks g1 and g2) over stimulus A differs from chance ($H_0: p = 0.5$; the two stimuli have either equal absolute values in experiments p1, p2, p3, g1 and g2 or equal relative values in experiments p4.1 and p4.2). For each task we show \hat{p} the observed choice proportions and the 95% binomial confidence interval. A p-value < 0.05 (italic) indicates a significant group-level bias in favour of one stimulus.

Reviewer #2: I would recommend having the “Sample Size” section in the Methods directly after the section “Participants” and before “Tasks”.

Response: We appreciate that suggestion and moved the “Sample Size” section behind the “Participants” section.

Reviewer #2:

Whenever you refer to the Method section throughout the manuscript, please refer to the specific section of the Methods instead of just writing “see methods”. The Methods section is too long to search for where that specific detail is located whenever being referred to that section in the manuscript.

Response: We appreciate this suggestion and agree it makes the manuscript easier to follow. We now point readers directly to the specific methods section.

Reviewer #2: *You define WAIC to be the “Widely Applicable Information Criterion” in the Methods section and the “wataike information criterion” in the Results section.*

Response: We thank the reviewer for pointing this out. It seems as the Widely Applicable Information Criterion (WAIC), also known as the Watanabe-Akaike Information Criterion, was here inadvertently abbreviated with a nonstandard short form of the latter. We now consistently use ‘Widely Applicable Information Criterion (WAIC)’ throughout.

Reviewer #2: *I still think some of the text/legends/axis captions in Figure 2 are a bit too small to be easily legible.*

Response: Due to a cyberattack on TU Dresden’s IT infrastructure, I currently cannot access the script used to regenerate Figure 2 and its quite complex. I further do think the high-DPI version I will upload with the final edits should be sufficient for readers. If not, I can recreate this figure as soon as server access is restored. I very much thank the reviewer for their helpful suggestions.

Reviewer #3:

The authors have again done well in answering my previous comments on the paper, and I especially appreciate the additional simulations performed for the expanded model recovery exercise given the mentioned computational cost. While there are still some smaller issues that could be addressed, I do not believe these should prevent the work from being published, so I offer them only as suggestions to consider. Thank you for your thoughtful responses to my previous queries, and I wish you the best for your work going forward.

Response:

We are glad that the additional methodological clarifications and new analyses have been helpful, and we appreciate the opportunity to further clarify our results and address the remaining points.

Reviewer #3:

- I'm grateful for the further expansion of the model recovery exercise, which I think was worth the extra effort to solidify the discriminability of the models. This being said, I'm afraid I still do not completely follow the procedure here, or at least the logic of it: why simulate 10,000 datasets if only 30 were actually examined? These do not seem to be required for the MCMC sampling process, so this appears to be wasted effort (even if not the main computational cost of the exercise), unless there is some rationale here I'm missing. Alternatively, if all 10,000 were fit, what do the 30 samples refer to? Adding to the confusion, the Supplemental states 'We then draw 20 random samples and refitted the whole task simulated datasets with all models 20 times' (page 14, line 157), suggesting each dataset was fit multiple times, which seems misaligned with the description in the main text. It is possible the precise details of this exercise are not hugely consequential to the actual results if recovery rates are high regardless, but I would add that clarity on procedure is important to properly interpret these results, and lay out the assumptions involved in these analyses.

Response: We very much thank the reviewer for acknowledging the expanded model-recovery exercise and apologize that our description was unclear. In our procedure, we generated posterior-predictive data during MCMC: at each post-warm-up draw we simulated a full dataset from the likelihood conditional on that joint parameter draw. Because our sampler stored 10,000 post-warm-up draws, this yielded 10,000 posterior-predictive datasets for each model. For the recovery analysis, we then selected 20 (per simulated model) of these datasets across the range of the posterior parameters and then re-fit each dataset once with every candidate model using the same MCMC settings as in the main analysis. This approach preserves within-subject and cross-parameter covariance learned from the empirical data and yields recovery results that are most relevant to the empirically plausible parameter regions (while accounting for within-subject parameter correlations and uncertainty). We agree that we could have reduced the number of stored

draws (e.g., to 1,000) with negligible impact on the representativeness of the posterior pool, thereby saving computational resources. We have revised the methods and supplemental information to clarify this procedure and will keep your suggestion in mind to save time in future work.

Reviewer #3:

Following on from this, I'm also slightly concerned that declaring a single best-fitting model in each simulated dataset (i.e. task) could amplify model discriminability: collecting measures across simulated "participants" may increase the margins between models, making them appear more distinct. This seems particularly relevant given the apparent heterogeneity in model fit in the empirical data: is this truly reflective of individual differences in model use, or could this be produced by a single common model? It may be worth examining the recovery results at the individual level to see whether recovery remains high even for a single subject.

Response: We thank the reviewer for this thoughtful point and agree. While group-level fits generally align with the overall participant-level patterns, declaring a single best-fitting model per simulated dataset aggregated across participants can sharpen margins between models and thus inflate discriminability, especially when some individuals are better captured by alternative models. As the reviewer framed this as an optional extension, we will transparently report it as a limitation in the Supplementary Information. We have added the following note (p. 14):

"Note that our model-recovery analysis is conducted at the group level within a hierarchical Bayesian framework. Aggregating across simulated participants can increase separation between models and may overestimate subject-level discriminability. Individual fits can show heterogeneity (see Supplementary Figs. S2–S3), and some participants may be better captured by alternative models. Accordingly, the recovery results should be interpreted as evidence for group-level discriminability rather than a claim that a single model uniformly explains every individual."

Thank you very much.